# Seismic risk scenarios for the residential buildings in the Sabana Centro province in Colombia

Dirsa Feliciano[1,2], Orlando Arroyo[1,2], Tamara Cabrera[3,4], Diana Contreras[5,6], Jairo Andrés Valcárcel Torres[7], and Juan Camilo Gómez Zapata[8,9]

[1] Faculty of Engineering, Universidad de La Sabana, Chía, 250001, Colombia

[2] Research Center on Disasters and Climate Change (CIDEC), Chía, 250001, Colombia

[3] Research Center for Integrated Natural Disaster Management (CIDIGEN), ANID/FONDAP/15110017, 4860, Santiago, Chile

[4] Department of Structural and Geotechnical Engineering, Pontificia Universidad Católica de Chile, 4860, Santiago, Chile

[5] School of Earth and Environmental Sciences, College of Physical Sciences and Engineering, Cardiff University, Park Place, Cardiff CF10 3AT, UK

[6] School of Engineering, Faculty of science agriculture and engineering, Newcastle University, Newcastle upon Tyne, NE1 7RU, UK

[7] Faculty of Engineering, Universidad de La Salle, Bogotá, 11121, Colombia

[8] Seismic Hazard and Risk Dynamics, GFZ German Research Centre for Geosciences, Potsdam, 14473, Germany

[9] Institute for Geosciences, University of Potsdam, Potsdam, 14469, Germany

*Correspondence to*: Dirsa Feliciano (dirsafeag@unisabana.edu.co)

**Abstract.** Colombia is in one of the most active seismic zones on Earth, where the Nazca, Caribbean, and South American plates converge. Approximately 83% of the national population lives in intermediate to high seismic hazard zones, and a significant part of the country's building inventory dates from before the nation's first seismic design code (1984). At present, seismic risk scenarios are available for the major cities of the country, but there is still a need to undertake such studies in other regions. This paper presents a seismic risk scenario for the "Sabana Centro" province, an intermediate hazard zone located close to the country's capital. An exposure model was created combining information from the Global Earthquake Model (GEM) Foundation, surveys, and the national census. Fragility and vulnerability curves were assigned to the building types of the region. A hazard model was developed for the region and eighteen earthquake scenarios with a return period of 475 years were simulated using the OpenQuake (OQ) hazard and risk assessment tool to estimate damage and economic losses. In addition, a social vulnerability index (SVI) based on demographic information was used to assess the direct economic loss in terms of replacement costs. The results show that 10% of all buildings considered in the region would experience collapse, and 7% would suffer severe damage. Losses account for 14% of the total replacement cost of the buildings and represent 21% of the annual Gross Domestic Product (GDP) of the region.

## 1. Introduction

Colombia is in one of the most active seismic zones on Earth, where the Nazca and Caribbean tectonic plates converge against the South American plate (Paris et al., 2000). The seismicity of the country is associated with the activity of the South American subduction zone along the Colombian Pacific, the Bucaramanga Seismic Nest (BSN), and several other active faults (Arcila et al., 2020). According to the Colombian Geological Service (SGC, by its acronym in Spanish), approximately 83% of the national population lives in areas with intermediate to high seismic hazard levels (AIS, 2010; Arcila et al., 2020).

In addition to the hazard levels mentioned above, more than 10 million Colombians live in houses vulnerable to seismic events (Build Change, 2021). This situation stems from non-engineered buildings and informal constructions that account for between 60 to 90% of the country's residential building stock (Bonet et al., 2016; Yepes-Estrada et al., 2017). Due to these conditions, earthquakes have resulted in considerable economic and human losses in recent history. Examples include the $M_w$ 5.5 Popayan earthquake in 1983 (Contreras, 2018) and the Mw 6.2 Armenia earthquake in 1999. In the first case, the earthquake caused 287 deaths, 7248 injuries, and 150 thousand people affected (Cardona et al., 2004; Lomnitz and Hashizume, 1985). This earthquake represented an estimated loss of 0.98% of the gross domestic product (GDP) for that year (Cardona et al., 2004; AIS, 2009). In the second case, this event left 1185 casualties, 8523 injured people (Naciones Unidas | CEPAL, 1999), and 35,000 buildings that collapsed or experienced severe damage (Chávez et al., 2021). The estimated losses from this earthquake amounted to 1.9% of that year's national GDP (AIS, 2009; Cardona et al., 2004). In such cases, field observations showed that the resulting damage was concentrated in old and historical buildings, and in those built from low-quality materials and using inadequate construction techniques (Villar-Vega and Silva, 2017; Cardona et al., 2004; Macdonald et al., 2000; PAHO, 1983).

To help formulate mitigation strategies for earthquakes, risk management agencies and researchers have developed earthquake risk scenarios for different countries at the local, national, and global levels (Chaulagain et al., 2014, 2015; Silva et al., 2014a; Erdik et al., 2003; Nievas et al., 2022). Recently, a seismic risk assessment and a set of earthquake scenarios were developed for the residential building stock of Colombia's three largest metropolitan centers: Bogotá, Medellín, and Cali (Acevedo et al., 2020). In addition, probabilistic seismic risk assessments have been conducted in cities such as Medellín (Salgado et al., 2014) and Manizales (Salgado et al., 2017; Carreño et al., 2017). Despite these efforts, there is still a need to assess the expected consequences of potential earthquake events in other parts of the country. Therefore, this study presents the methodology and results of a seismic risk scenario for the "Sabana Centro" region, a zone made up of 11 municipalities located in the Department of Cundinamarca, north of Bogotá, the capital of the country. Historical earthquakes have occurred and affected this region. In 1644, a Mw 5.5 earthquake mainly affected churches and houses in Bogotá, and in 1743 a Mw 6.2 earthquake caused severe damage to the churches of Cota and Chía, two of the region's municipalities (JICA, 2002; Salcedo and Gómez, 2013), which saw intensities of VII being experienced (Mercalli scale) (SGC, 2021a).

The development of seismic risk scenarios involves three main components: 1) a set of ground motion fields estimated for a given earthquake rupture (seismic hazard model), 2) an exposure model defining the types of buildings in the study zone

and their spatial distribution, and 3) a set of fragility and vulnerability functions that describe the seismic vulnerability of the

buildings. The seismic vulnerability of a structure is a quantity associated with the likelihood of it suffering damage in the event of ground motion of a given level (Calvi et al., 2006). To simulate this vulnerability, fragility curves are associated with the type of construction employed for the buildings in the study area. This association allows the estimation of the probability of a building suffering different damage levels due to earthquake-induced ground motion, i.e., light, moderate, extensive, and collapse.

For the first component (i.e., the hazard), a national probabilistic seismic hazard model developed by the SGC was used to select the events of interest to estimate potential damage and expected losses. In addition, a model developed by the SGC that describes the spatial distribution of $Vs_{30}$ values was considered as a proxy to account for ground motion amplification due to soil conditions (Choi and Stewart, 2005). Information available from the national census was used to create the exposure model. The methodology used in (Yepes-Estrada et al., 2017) was followed to assign the number of buildings per municipality.

Regarding the structural vulnerability of the building stock, a database of fragility functions developed for the residential building stock in South America by Villar-Vega et al. (2017) and those developed for global seismic risk analysis (Martins and Silva, 2021) were taken as a basis. Seismic risk scenarios were simulated using these three components as input for the OpenQuake (OQ) hazard and risk assessment engine (Silva et al., 2014b), from which the number of damaged buildings and associated economic losses was calculated.

One aspect of risk assessment frequently neglected is social vulnerability (SV). Post-disaster assessments have demonstrated that the extent of losses from disasters depends not only on the magnitude and duration of extreme natural events but also on the resilience of the population to build-back their lives, livelihoods, and property (Chen et al., 2013; Schmidtlein et al., 2011; Contreras, 2016). The most vulnerable segments of a population are usually the most severely affected by extreme natural phenomena (Contreras et al., 2020b). Experiences from past earthquakes, such as the 2010 Haiti earthquake which resulted in 200,000 deaths (Boot

et al., 2010) and 1.5 million homeless (Contreras et al., 2020a) have shown that casualties and building damages are higher among people who live in poorly constructed non-engineered buildings (Boot et al., 2010). In some cases, less-favored families may be forced to sell their income-providing assets to fulfill their immediate basic needs, even though they are less able to replace them. Moreover, the impact of natural phenomena may span for generations, as parents may need to withdraw children from schools to help generate family income, thus limiting their future opportunities. Consequently, earthquake preparedness plans should consider that the

consequences of these events have a greater impact on more vulnerable members of a community. According to data from the World Bank (WB), Colombia has a Gini index of 0.517, making it a country with a substantial level of income inequality (World Bank, 2022). In fact, the same study showed that Colombia's economy has the second most uneven distribution of income within Latin America, with only Brazil being higher. In Colombia, inequality goes beyond income level, as it is also present in aspects related to the quality of life, such as social security, access to basic services, education, and so forth (Joumard and

Londoño Vélez, 2013). These differences are visible throughout the country, and the Sabana Centro province is an example of this. This study, therefore, also considers social vulnerability (Cutter et al., 2003), which is represented as an index to adjust the economic losses due to structural damage.

## 2. Description of the study area

Sabana Centro is a region of Cundinamarca, Colombia, to the north of Bogotá, the country's capital. Cundinamarca is one of the four most populated regions of the country and Sabana Centro is one of the provinces that contributes the most population (18%) and the department GDP (32%). The province comprises 11 municipalities (Figure 1) and according to the 2018 National Population and Housing Census (CNPV, by its acronym in Spanish), the number of inhabitants of the region is 539,295 (DANE, 2018). Table 1 presents the area and the number of inhabitants of the 11 municipalities that make up this region.

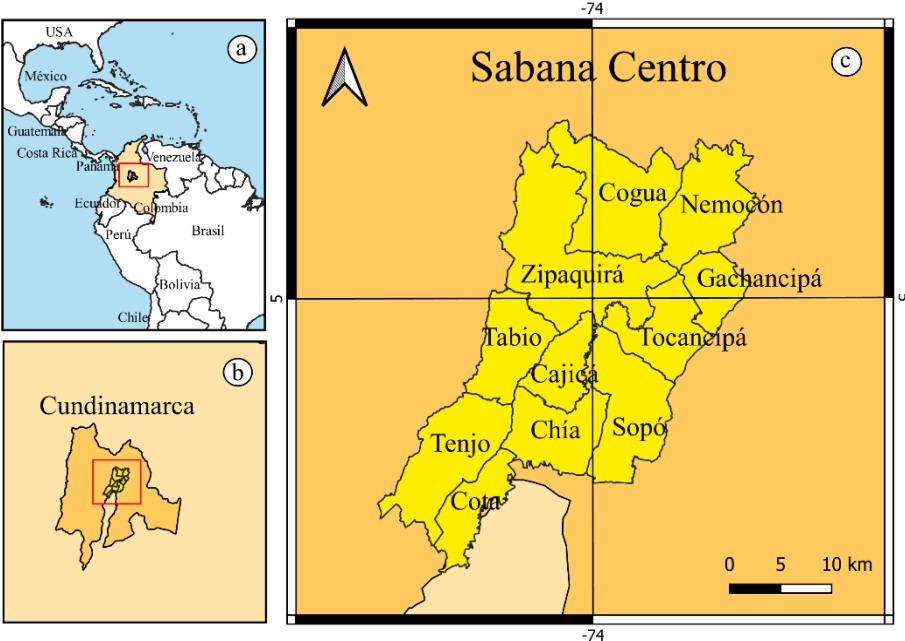

**Figure 1.** a) Location of the study area within Colombia. b) The region within the department of Cundinamarca ("department" is the first administrative division in Colombia). c) The municipalities which make up the Sabana Centro province.

**Table 1.** Area, distribution of population and population density of the eleven municipalities that make up Sabana Centro (DANE, 2018).

| Municipality | Area (km²) | Inhabitants | Population density (Inhabitants/km²) |
|---|---|---|---|
| Cajicá | 51 | 82,244 | 1613 |
| Chía | 79 | 132,181 | 1673 |
| Cogua | 136 | 22,067 | 162 |
| Cota | 55 | 32,691 | 594 |
| Gachancipá | 44 | 17,026 | 387 |
| Nemocón | 94 | 13,171 | 140 |
| Sopó | 111.5 | 25,782 | 231 |

| | | | |
|---|---|---|---|
| Tabio | 74.5 | 21,665 | 291 |
| Tenjo | 108 | 21,935 | 203 |
| Tocancipá | 73.51 | 39,996 | 544 |
| Zipaquirá | 197 | 130,537 | 663 |

In addition to natural population growth in Colombia, the country's capital and several municipalities have experienced a greater increase in population partly due to the constant migration from neighboring Venezuela since 2015. The region of Sabana Centro has not been no stranger to this process, wherein recent years, it has seen a significant demographic change in most municipalities (Sabana Centro Cómo Vamos, 2019). In 2015, the population density was 460 inhabitants per km², and in 2018, it had risen to 527 inhabitants per km². This increase in population density means that there was a growth rate of 14.6%,

higher than the national average of 5.9%. Among the municipalities, Chía, Cajicá and Zipaquirá had the highest population growth with 64% of the region's total population. The number of inhabitants in the region represents 18% of the department of Cundinamarca (67% in urban areas and 33% in rural).

## 3. Description of input parameters

### 3.1. Seismic hazard

The SGC, in collaboration with researchers from the Geological and Mining Institute of Spain and the GEM Foundation, developed a national seismic hazard model (Arcila et al., 2020). Overall, this national seismic hazard model comprises a set of tectonic environments and seismogenic sources. In that study, the seismicity of the Colombian territory was classified into four tectonic environments. Superficial events (cortical) correspond to events in the national territory down to depths limited by the upper crust-mantle boundary. Interplate earthquakes of the Colombian Pacific subduction zone correspond to

earthquakes that occur in the area of contact between the Nazca and South American Plates along the country's Pacific coast. Earthquakes in the Benioff area correspond to earthquakes inside the plate, which is subducting towards the east from the Colombian Pacific towards the country's interior. The Bucaramanga's seismic nest corresponds to an area where earthquakes with moment magnitudes between Mw 4.0 and 5.0 usually occur at depths between 140 and 200 km (Prieto et al., 2012).

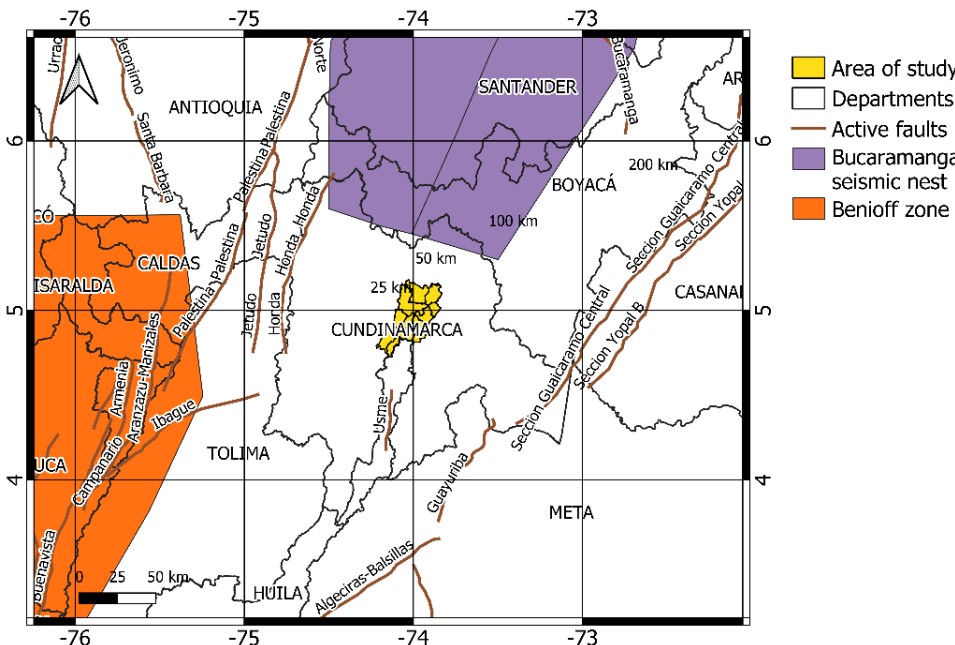

**Figure 2.** Seismic hazard sources close to the Sabana Centro province (Arcila et al., 2020): The active faults are presented in brown lines. The Usme fault is located less than 50 km from the municipality of Tenjo. The Benioff zone and the Bucaramanga seismic nest are located less than 120 km and 150 km approximately from Tenjo.

### 3.1.1. Definition of the earthquake scenarios

The Sabana Centro province is located close to seismic hazard sources of different tectonic regional types, as shown in Figure 2. According to the national seismic hazard model developed by the SGC and the GEM Foundation, the Sabana Centro province is close to active shallow seismic sources (such as the Usme Fault), intraplate events from the Benioff zone, as well as deep events from the Bucaramanga's seismic nest (Arcila et al., 2020).

In this study, earthquake events are defined in terms of the magnitude, location, and geometric characteristics of their ruptures. For the determination of the magnitude and location of the events to be considered in the estimation of damage, events from the unified earthquake catalogue developed by the SGC (SGC, 2021b) within a radius of 200 km were considered. Figure 3 shows the events of the complete catalog, considering those from the seismic nest, as well as those events from a cortical environment. The figure shows events at distances less than 50 km from the center of Tenjo near the surface with depths less than 70 km and moment magnitudes ranging from Mw 4.0 to 5.5. There are also some events at depths between 70 km and 300 km at distances between 50 km and 100 km. These events range between a moment magnitude of Mw 4.0 and 6.5. It should be noted that most of the events in this area are shallow. Some far events at distances greater than 100 km are shallow events that can range between magnitudes of Mw 4.0 and Mw 7.0. It is also noted that there are deep events that can reach a moment magnitude of Mw 7.0.

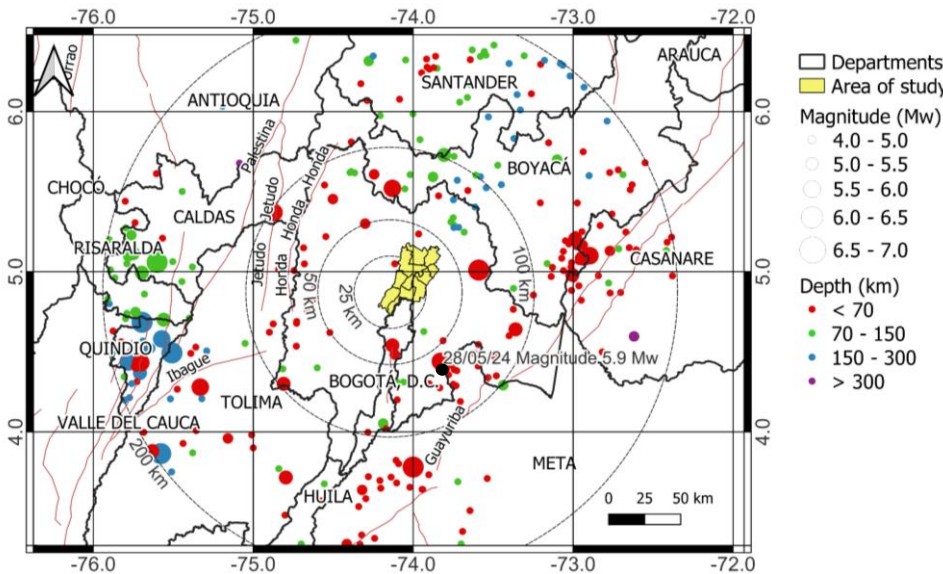

**Figure 3.** Geographic distribution of events occurring within 200 km of the study area selected from the Unified Earthquake Catalogue of the SGC (SGC, 2021b). The size of the circle represents the magnitude of the event, and color indicates depth. The event marked with the black circle corresponds to the focal mechanism of the Quetame earthquake of magnitude Mw 5.9.

To identify the type of events that contributes the most to the seismic hazard of the Sabana Centro province, a hazard
disaggregation analysis (Bazzurro and Cornell, 1999) was conducted using the OQ Engine, considering the national seismic hazard of Colombia (Arcila et al., 2020). Details of the seismic hazard disaggregation procedure are described in Pagani et al. (2014). The disaggregation was developed for a point within the region of analysis, which corresponds to the population centroid of the municipality of Tenjo (longitude: -74.144, latitude: 4.872) considering the Joyner-Boore distance to the projection of the rupture surface. The annual rate is 0.0021 (10% probability of exceedance over 50 years, or 475 years return
period). Regarding the geometry of the earthquake ruptures, in the case of shallow events, the dip, strike, and rake angles were defined using available information from the seismic hazard model (Arcila et al., 2020), as well as the focal mechanism of the Quetame earthquake of magnitude Mw 5.9, which occurred in May 2008 (Páez et al., 2015).

The results obtained given the distance and magnitude of the earthquakes are presented in Figure 4. In the case of the Peak Ground Acceleration (PGA), crustal events make a higher contribution to the seismic hazard, located at distances less
than 35 km, with magnitudes ranging between Mw 5.0 to 7.0. These events correspond to seismic sources of the crustal tectonic region type. A lower contribution is observed from events of the Benioff zone with magnitudes between Mw 6.5 and 7.0 at distances between 125 and 150 km. In the case of spectral acceleration with a period of 1.0 second (Sa (1.0s)), the most significant contribution also comes from crustal events. However, there is an important contribution of events of magnitude greater than Mw 8.0 at distances ranging between 275 and 300 km, whose origins are in the subduction-interplate tectonic
region.

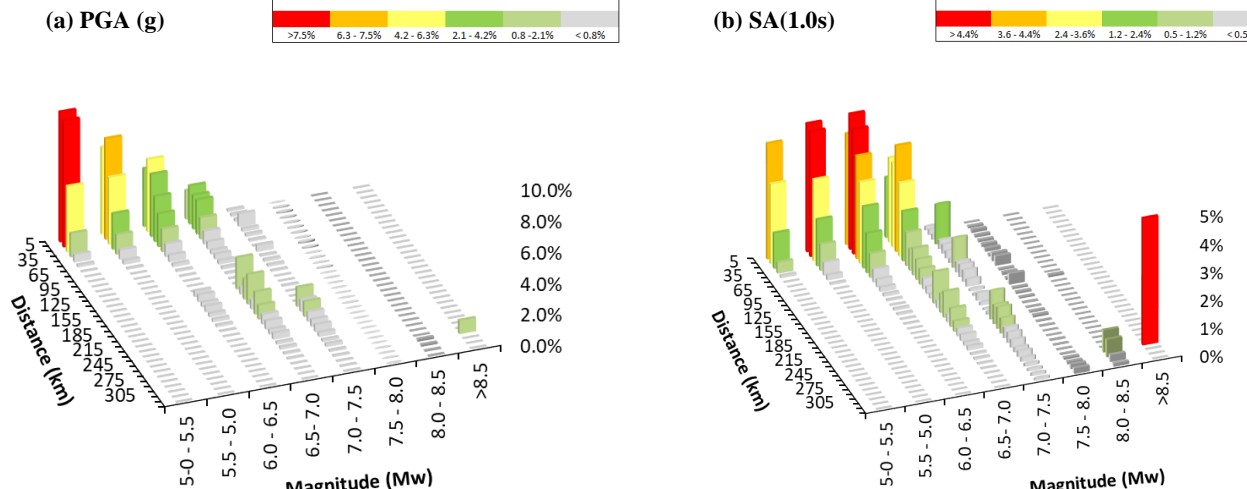

**Figure 4.** Contribution to the seismic hazard of earthquakes by distance and magnitude (a) PGA (g); (b) Sa (1.0 s). The color scale represents the percentage contribution of seismic events to the seismic hazard, with gray representing the lowest contribution and red the highest.

Based on this disaggregation, eighteen crustal events were selected from the probabilistic seismic hazard catalogue to be used in this study to calculate the expected damages and economic losses. The magnitude, location and geometry of ruptures are shown in Table 2. The epicenter of each event is located within the municipality mentioned in the first column of the Table 2 and shown in Figure 5.

**Table 2.** Information describing the seismic events selected as scenarios in this work to estimate potential damage and impact.

| Municipality | Magnitude (Mw) | Depth (km) | Strike (°) | Dip (°) | Rake (°) |
|---|---|---|---|---|---|
| Cajicá | 6.35 | 5 | 0 | 90 | 0 |
| Chía | 5.95 | 5 | 0 | 90 | 0 |
| Cogua | 6.45 | 5.51 | 0 | 90 | 0 |
|  | 6.35 | 5 | 0 | 90 | 0 |
| Cota | 6.95 | 9.27 | 39 | 76 | -6.5 |
|  | 5.55 | 5 | 0 | 90 | 0 |
| Gachancipá | 5.95 | 7.5 | 39 | 76 | -6.5 |
| Nemocón | 6.65 | 6.78 | 0 | 90 | 0 |
|  | 6.25 | 5 | 0 | 90 | 0 |
| Sopó | 6.55 | 6.11 | 0 | 90 | 0 |
|  | 6.25 | 5 | 0 | 90 | 0 |
| Tabio | 6.65 | 6.78 | 0 | 90 | 0 |
|  | 6.25 | 5 | 0 | 90 | 0 |
| Tenjo | 5.95 | 7.5 | 81 | 38 | -76 |
|  | 5.35 | 5 | 0 | 90 | 0 |
| Tocancipá | 6.85 | 25 | 81 | 38 | -76 |

| | 6.15 | 5 | 0 | 90 | 0 |
| --- | --- | --- | --- | --- | --- |
| Zipaquirá | 5.65 | 5 | 0 | 90 | 0 |

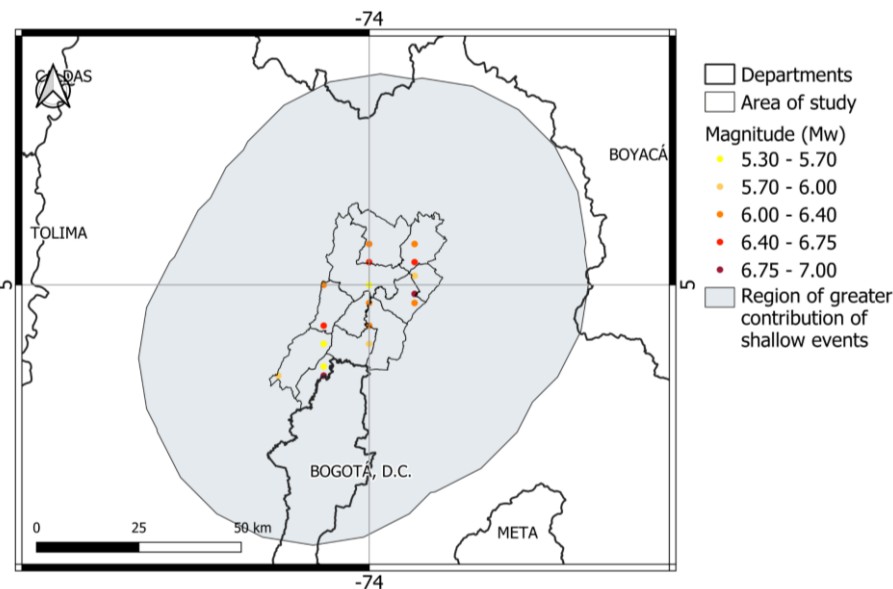

**Figure 5.** Location of the 18 selected events, the color range varies according to the magnitude of the timing of the events. The gray region represents the zone where shallow events have the largest contribution.

### 3.1.2. Soil-site conditions

To the best authors' knowledge, there are no specific studies of the seismic response of soil deposits within the region of Sabana Centro reported in the scientific literature. Therefore, the average shear wave velocity in the top 30 m ($Vs_{30}$) has been
considered a proxy to address the contribution of soil-site conditions on the calculated ground motions at this regional scale (Derras et al., 2017). For designing the foundations of new buildings, the current Colombian seismic design code, NSR-10 (AIS, 2010), classifies soils based on the $Vs_{30}$ values of the site of interest and proposes a set of coefficients to account for soil effects in the calculation of the seismic demand. Therefore, such ranges of Vs30 are considered for the Sabana Centro province. A map of $Vs_{30}$ values within the Sabana Centro province is presented in Figure 6, according to a map developed by Eraso and
Montejo (2020), with a 7.5 arc second resolution (~250 m$^2$) based on digital elevation models. It shows the presence of different conditions, from soft soils with values of $Vs_{30}$ under 200 m/s to stiff soils with $Vs_{30} > 1000$ m/s. The figure also shows that most urban blocks are in sites with $Vs_{30}$ values less than 450 m/s. In particular, the municipalities of Tenjo, Tocancipá, Nemocón, Gachancipá, Cajicá, and Chía are in areas with $Vs_{30}$ less than 180 m/s, corresponding to soft soils.

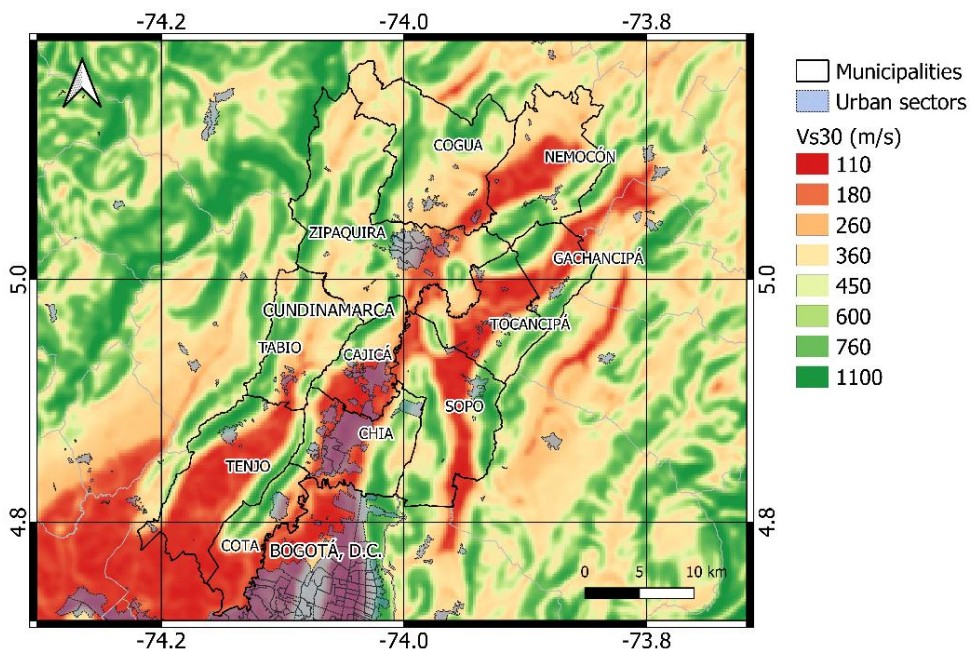

**Figure 6.** Spatial distribution of $Vs_{30}$ values in the Sabana Centro province according to Eraso and Montejo, (2020). The urban areas of the municipalities in the region are also shown.

### 3.1.3. Selection of ground motion prediction equations

Ground motion prediction equations (GMPE) allow to forecast the expected intensity of ground motion at a given site due to an earthquake event in terms of some measure, for example, spectral accelerations (Stewart et al., 2015). Several equations have been proposed worldwide for different tectonic environments, with different functional forms and input parameters. In Colombia, two sets of equations were developed to define the seismic hazard maps of the national building design code (NSR-10) (Gallego Silva, 2000) and the bridge design code (CCP-14) (Bernal Granados, 2014). More recently, Arcila et al. (2020) defined logic trees of GMPEs for the different tectonic regions of the country as a way to address epistemic uncertainty in the selection of other GMPEs, following the criteria proposed by Scherbaum et al. (2005) and Cotton et al. (2006), as shown in Table 3.

As introduced above, this study uses $Vs_{30}$ to account for ground motion amplification due to soil conditions (Choi and Stewart, 2005). The values in this region range between 112 m/s and 1100 m/s. This study considered crustal earthquakes and among the three GMPE proposed by Arcila et al. (2020) for shallow crustal regions in Colombia, the Idriss (2014) GMPE is not defined for $Vs_{30} < 450$ m/s, therefore, it was not considered for the scenarios. The weight assigned to this model (0.399) was distributed proportionally between the Cauzzi et al. (2015) and Abrahamson et al. (2014) GMPE, whose final weights used in this study are 0.65 and 0.35, respectively.

**Table 3.** Logic tree of GMPEs for crustal events as defined by Arcila et al. (2020) for the National Model and the actual weights used for the earthquake's scenarios.

| Tectonic region type | GMPE | Weight | |
|---|---|---|---|
| | | Defined in the national model | Used for the scenarios |
| | Idriss (2014) | 0.399 | 0 |
| Shallow crustal | Cauzzi et al. (2014) | 0.390 | 0.65 |
| | Abrahamson et al. (2014) | 0.211 | 0.35 |

Using the GMPE logic tree shown in Table 3, the mean expected PGA values for the Chía Mw 5.95 scenario of Table 2 may range between 0.12g (Cogua) and 0.49g (Cajicá).

### 3.2. Exposure model for the residential building stock

The building exposure model for the region has information about the building classes, the number of buildings, inhabitants,
and the buildings' replacement costs. To develop this model for Sabana Centro, the methodology used by the South America Risk Assessment (SARA) project to develop exposure models in South America (Yepes-Estrada et al., 2017) was taken as a basis. The source of information to assign the number of buildings was the 2018 national census (DANE, 2018). The census allowed having information on the number of dwellings and typical wall and roof materials, which were used to infer the different classes of buildings by municipality. A total of 156,628 dwellings were calculated; this number differs from that
reported by the national census by 2.8%, since it did not consider dwellings whose wall material is poured concrete. This material was not included since there was no information available to relate it to any type of building class. The set of dwellings were related to the same building classes and same relationships ('mapping schemes') used in Yepes-Estrada et al. (2017). As the census information is reported in terms of dwellings the procedure used in Yepes-Estrada et al. (2017) to calculate the number of buildings was also followed. Then, this data was complemented using information collected during remote surveys
carried out by students from the Universidad de La Sabana in the municipality of Chía. The building replacement cost refers to the cost of structural and non-structural components of a building and it is a value associated with the building's rehabilitation. This study has only considered the structural cost per building calculated based on cadastral information available in the Territorial Statistical Systems (TerriData[1]) of the country. This replacement cost was computer per building, expressed in USD. As the currency in Colombia is in Colombian pesos, the exchange to U.S. dollars was made for an average
exchange rate of 4080 pesos. Figure **7** shows the results of inhabitants, buildings, and their total replacement cost for the region. The bold numbers indicate the percentages for each municipality.

---

[1] https://terridata.dnp.gov.co/index-app.html#/

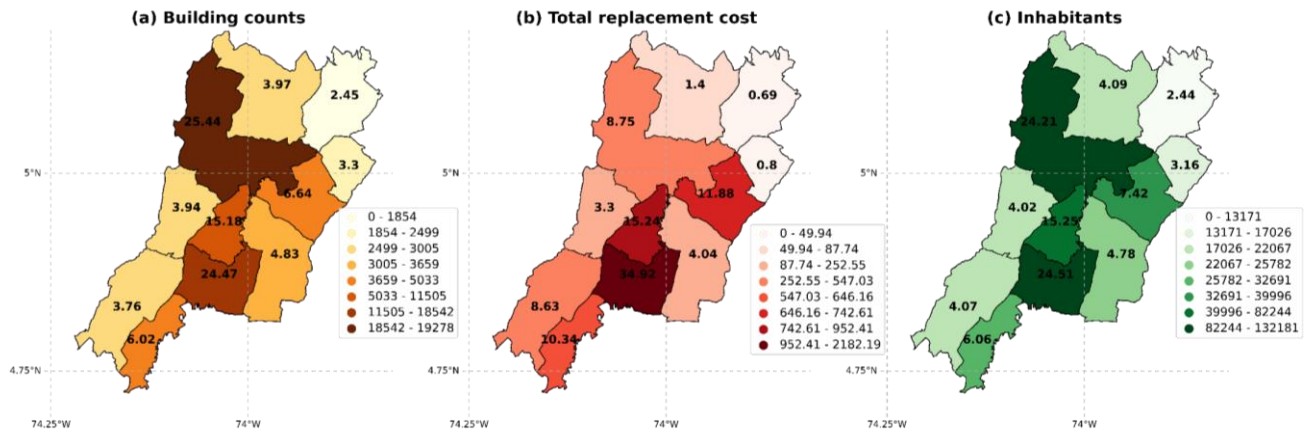

**Figure 7.** Summary of the exposure model for the Sabana Centro province. a) Building counts, b) total replacement cost in million USD, and c) inhabitants per municipality.

A total of 75,778 residential buildings in the region were classified into thirty-three building classes. Table 4 provides a description of these typologies along with the number of buildings within each category and their percentages. Among them, 6249 randomly distributed buildings in Chía were inspected in 2020 by civil engineering students of the Universidad de La Sabana. Their attributes were collected, making use of the Rapid Remote Visual Screening -RRVS web-platform (Haas et al., 2016), which allowed the use of the GEM V.2.0 taxonomy (Brzev et al., 2013) as a checklist while observing the buildings' façades through Google Street View. The resulting dataset is available in Arroyo et al. (2022). Four attributes of the GEM v.2.0 taxonomy were used for classifying these inspected buildings: the main construction material type, material technology, lateral load-resisting system, the expected level of ductility, and the number of stories. During the elaboration of the surveys, instead of "labelling" buildings as certain typologies, the collected attributes were used to classify them in a probabilistic manner. For such a purpose, the method proposed in Pittore et al. (2018) was used to evaluate the level of compatibility between the observed building attributes and each predefined building typology. Details of this process can be consulted in Arroyo et al. (2022). This procedure allowed to compare the percentages of the building classes calculated based on the SARA methodology and discretize the buildings by height.

From Table 4 it is noticeable that 58.60% of the buildings are constructed of non-ductile unreinforced masonry walls, including adobe blocks and dressed and semi-dressed stone. In addition, 20.92% of the buildings are from non-ductile confined masonry and 3.32% are non-ductile reinforced concrete frames, for a total of 82.84% non-ductile buildings. Figure 8 shows the number of buildings for the three types of construction materials identified in the exposure model: concrete, masonry, and wood. This figure shows that the predominant construction material is masonry (65,272 buildings), mainly in Chía and Zipaquirá, with more than ten thousand buildings for each one. Then, there are those buildings made of concrete (6745), with more than one thousand in Chía, Zipaquirá and Cajicá. Last, there are those structures made of wood (3762), especially in Chía with more than 990 units. The number of masonry buildings represents the 86.14% of the total buildings in Sabana Centro,

whereas those of concrete and wood represent 8.90% and 4.96%, respectively. The number of buildings for each of the 33 typologies is depicted in Figure 9. In the case of concrete, the predominant building class is one story non-ductile reinforced concrete moment frames, whilst for the masonry buildings, the non-ductile unreinforced masonry walls class is predominant.

**Table 4.** Summary of the building typologies in the exposure model defined for the study area. The building classes are defined based on the GEM v.2.0.

| Building class | Description | Number of buildings | Proportion (%) | Replacement cost (M. USD) |
|---|---|---|---|---|
| CR/LDUAL/DUC/H:4,7 | Ductile reinforced concrete dual frame-wall system, 4 to 7 stories | 14 | 0.02 | 34.94 |
| CR/LFINF/DUC/H:1 | Non-Ductile reinforced concrete infilled frames, 1, 2 and 3 stories | 855 | 1.13 | 557.05 |
| CR/LFINF/DUC/H:2 | | 1207 | 1.59 | 786.42 |
| CR/LFINF/DUC/H:3 | | 453 | 0.60 | 294.91 |
| CR/LFINF/DUC/H:4,7 | Ductile reinforced concrete infilled frames, 4 to 7 stories | 464 | 0.61 | 1131.06 |
| CR/LFM/DNO/H:1 | Non-Ductile reinforced concrete moment frames, 1, 2 and 3 stories | 855 | 1.13 | 557.05 |
| CR/LFM/DNO/H:2 | | 1207 | 1.59 | 786.42 |
| CR/LFM/DNO/H:3 | | 453 | 0.60 | 294.91 |
| CR/LFM/DUC/H:4,7 | Ductile reinforced concrete moment frames, 4 to 7 stories | 464 | 0.61 | 1131.06 |
| CR/LWAL/DUC/H:4,7 | Ductile reinforced concrete walls, 4 to 7 stories | 292 | 0.39 | 715.24 |
| CR/LWAL/DUC/H:1 | Ductile reinforced concrete walls, 1, 2 and 3 stories | 163 | 0.22 | 170.02 |
| CR/LWAL/DUC/H:2 | | 230 | 0.30 | 240.03 |
| CR/LWAL/DUC/H:3 | | 86 | 0.11 | 90.01 |
| MCF/LWAL/DNO/H:1 | Non-Ductile confined masonry, 1, 2 and 3 stories | 6154 | 8.12 | 1388.94 |
| MCF/LWAL/DNO/H:2 | | 7055 | 9.31 | 1713.00 |
| MCF/LWAL/DNO/H:3 | | 2646 | 3.49 | 642.37 |
| MCF/LWAL/DUC/H:1 | Ductile confined masonry walls, 1, 2 and 3 stories | 1230 | 1.62 | 796.62 |
| MCF/LWAL/DUC/H:2 | | 1736 | 2.29 | 1124.65 |
| MCF/LWAL/DUC/H:3 | | 651 | 0.86 | 421.74 |
| MR/LWAL/DUC/H:1 | Ductile reinforced masonry walls, 1, 2 and 3 stories | 473 | 0.62 | 384.56 |
| MR/LWAL/DUC/H:2 | | 668 | 0.88 | 542.91 |
| MR/LWAL/DUC/H:3 | | 251 | 0.33 | 203.59 |
| MUR/LWAL/DNO/H:1 | Non-Ductile unreinforced masonry walls, 1, 2 and 3 stories | 14738 | 19.45 | 3516.93 |
| MUR/LWAL/DNO/H:2 | | 19682 | 25.97 | 4753.33 |
| MUR/LWAL/DNO/H:3 | | 7384 | 9.74 | 1783.33 |
| MUR-ADO/LWAL/DNO/H:1 | Non-Ductile unreinforced masonry with adobe blocks walls, 1 and 2 stories | 231 | 0.30 | 48.17 |
| MUR-ADO/LWAL/DNO/H:2 | | 254 | 0.34 | 54.71 |

| | | | | |
|---|---|---|---|---|
| MUR-STDRE/LWAL/DNO/H:1 | Non-Ductile unreinforced masonry with dressed | 675 | 0.89 | 134.67 |
| MUR-STDRE/LWAL/DNO/H:2 | stone walls, 1 and 2 stories | 941 | 1.24 | 187.74 |
| MUR-STRUB/LWAL/DNO/H:1 | Non-Ductile Unreinforced masonry with semi- | 210 | 0.28 | 40.19 |
| MUR-STRUB/LWAL/DNO/H:2 | Dressed stone, 1 and 2 stories | 292 | 0.39 | 56.02 |
| W/WLI/DUC/H:1 | Ductile light wood members, 1 and 2 stories | 1515 | 2.00 | 357.22 |
| W/WLI/DUC/H:2 | | 2246 | 2.96 | 554.13 |

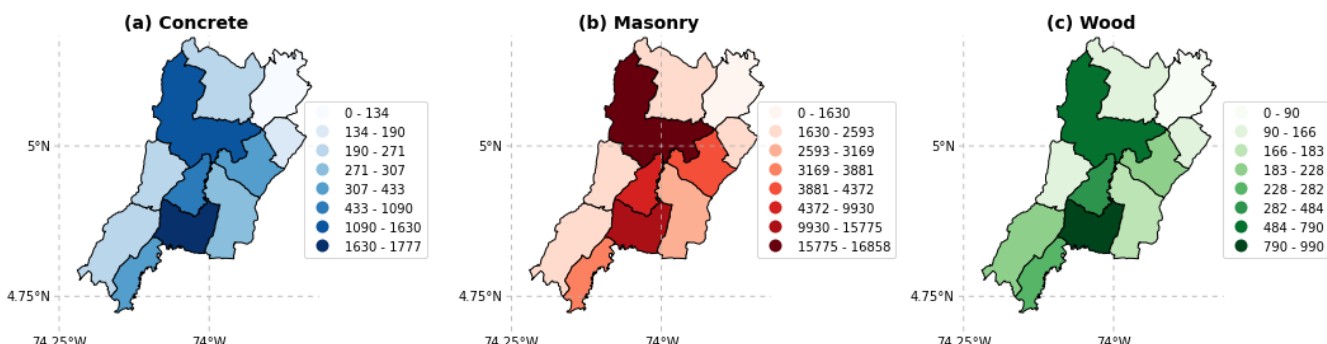

**Figure 8.** Spatial distribution of buildings whose construction materials are (a) concrete, (b) masonry, and (c) wood within each municipality.

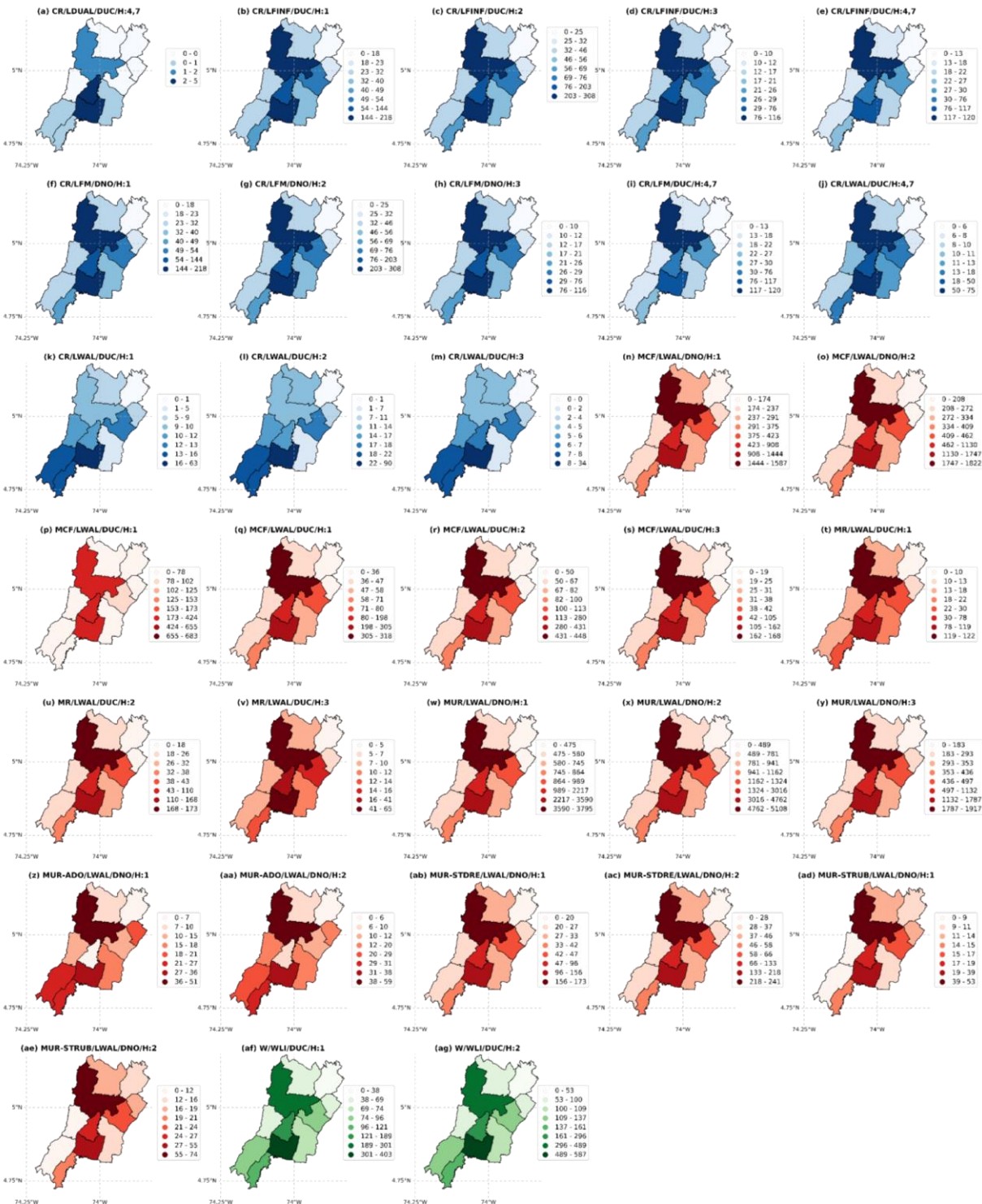

**Figure 9.** Spatial distribution of the building counts per class within each municipality.

### 3.3. Physical vulnerability of residential building stock to seismic ground shaking

A large part of the building inventory was constructed using unreinforced masonry and with characteristics that make them non-ductile or with low ductility. Therefore, it is necessary to make an appropriate assignation of the fragility curves to evaluate their physical vulnerability to ground-shaking. In the absence of specific curves locally developed for the Sabana Centro province, fragility curves available in the literature were selected to represent these structures. Thereafter, a literature review was undertaken to select the fragility functions that most closely resemble the characteristics of the Sabana Centro building inventory. The Physical Vulnerability Suite of the GEM Foundation (OpenQuake Platform - Vulnerability, 2021) was considered for the review. The GEM database for the specific case of Colombia has the curves developed by Acevedo et al. (2017) for unreinforced masonry houses constructed in Antioquia, Colombia. There are some curves for reinforced concrete buildings with geographical applicability in Manizales, Colombia by Bonett (2003) and the dataset of Villar-Vega. (2014) for South America. Although the set of curves covers different types of buildings, they are calculated based on different methodologies and different damage states.

Another available dataset of fragility curves are those developed by Martins and Silva (2021), who covered nearly 500 building classes at global level including Colombia. The fragility is calculated from nonlinear dynamic analyses performed on equivalent single-degree-of-freedom (SDOF) oscillators. They considered four damage states that are also intended to study in the present research: slight, moderate, extensive and collapse. The corresponding damage thresholds were defined based on the spectral displacement of the structures. At regional level also are the set of fragility curves for the residential building stock in South America (Villar-Vega et al., 2017), covering 54 common building classes. The methodology used for the derivation of the curves is similar to the one used in Martins and Silva. (2021).

Based on the information collected, the fragility curves available in Martins and Silva. (2021) were used mainly and complemented with those of Villar-Vega et al. (2017). These curves were selected in order to prevent an unbiased comparison of risk between the different municipalities in the region due to the different methodologies used to develop the fragility curves. Therefore, a set of 33 fragility functions was used to represent the probability of exceeding a level of damage conditioned to ground shaking intensity. These functions are comprised of 28 sets of curves reported by Martins and Silva. (2021) and five sets developed by Villar-Vega et al. (2021). The last one are assigned to Non-Ductile confined masonry, 1, 2 and 3 stories and Ductile light wood members, 1 and 2 stories, since in the former these building classes were not included. These fragility functions are described by a cumulative probability curve with a lognormal distribution and examples of some of them are presented in Figure 10. This set of fragility curves was used to calculate the damage to the buildings included in the exposure model. Based on these curves, vulnerability functions were developed to evaluate the losses in the region. The loss ratios used in this study are 2%, 10%, 50%, and 100% for the slight, moderate, extensive and collapse damage, respectively.

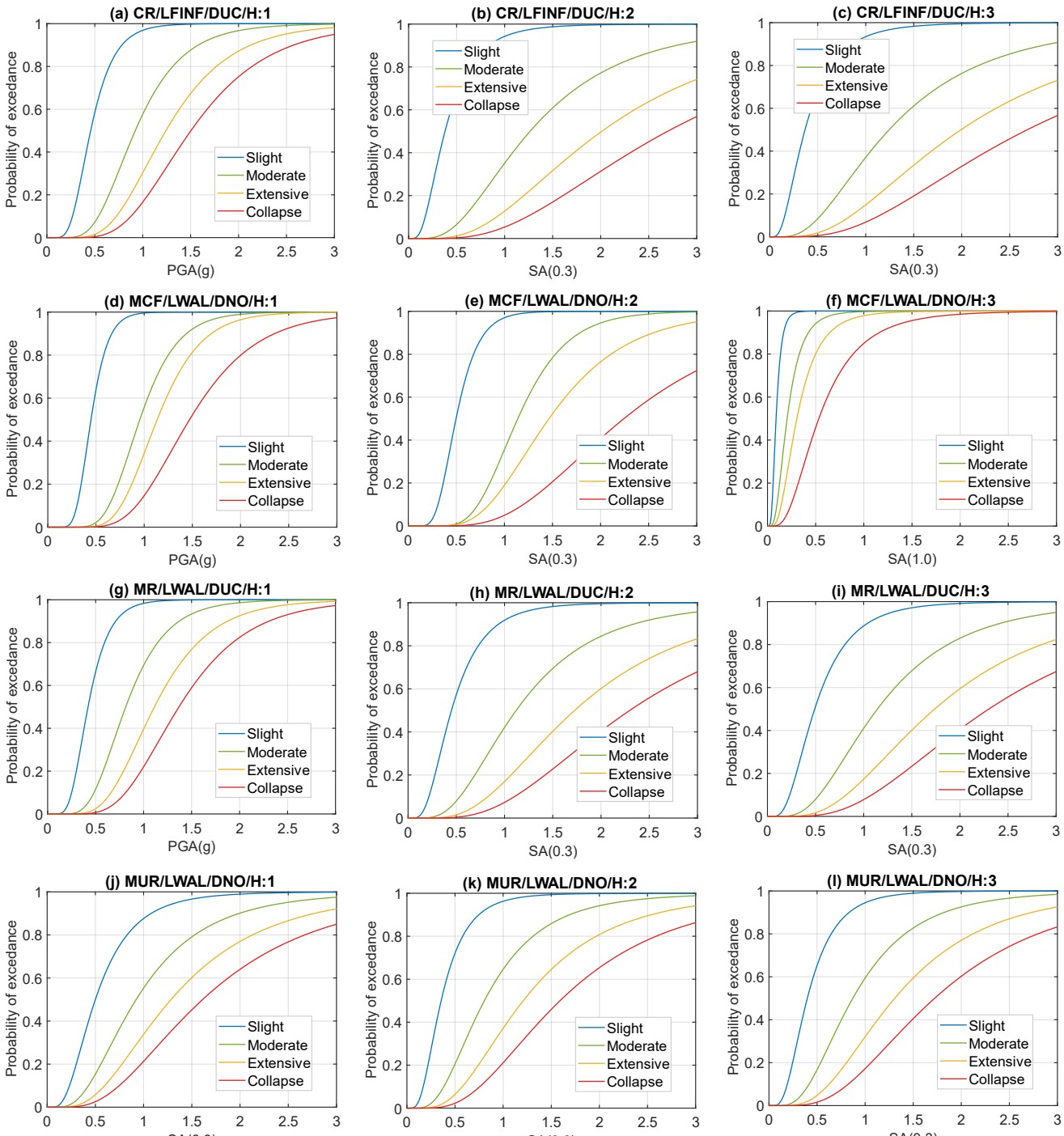

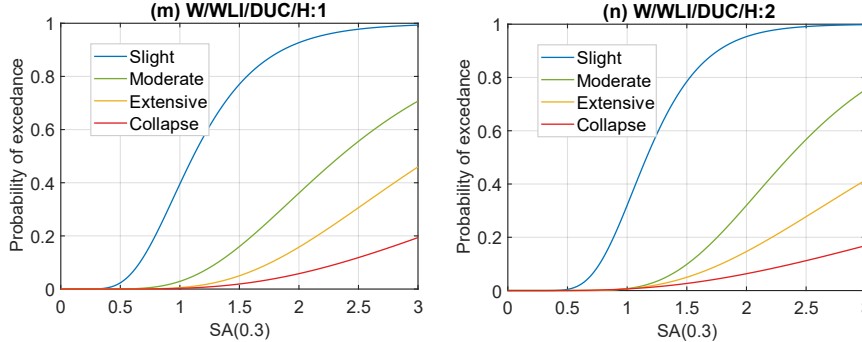

**Figure 10.** Fragility curves for 14 of the 33 building classes listed in Table 4. The curves describe the differential seismic vulnerabilities for the predominant building classes for each type of material: reinforced concrete (CR), confined masonry (MCF), unreinforced masonry (MUR) and wood (W).

### 3.4. Social Vulnerability (SV)

To determine the level of social vulnerability (SV) of the municipalities of the Sabana Centro province, this paper estimated a
social vulnerability index (SVI) based on the methodology proposed by Cutter et al. (2003). The social equivalent to a quantitative physical risk assessment for earthquakes is an SVI. Social vulnerability is the reason for the different experiences of communities regarding the consequence of earthquakes (Burton, C. G., & Silva, V., 2016). The construction of composite indicators based on the mathematical combination of a set of indicators, which consists of a group of variables, is one of the most common methods to objectively assess SV (Freudenberg, 2003). There are several methodological approaches for the
construction of composite indicators, but in general, the steps include: (1) the identification of pertinent variables, (2) the aggregation of variables into indicators and composite indicators (3) multivariate analysis (4) weighting (5) convolution or link of variables and (6) visualization and dissemination of results (Burton, C. G., and Silva, V., 2016).

The SVI index aims to identify those municipalities in Sabana Centro whose inhabitants are more vulnerable to an earthquake based on a selection of specific variables, indicators, and composite indicators.  The indicators were aggregated
into five composite indicators constructed for the SVI  of SARA project[2]: population, economy, infrastructure, education, and health. The composite indicator of population considers  the indicators that capture the capacity of population to mitigate their risk and recover from earthquakes. In the current research the composite indicator of population accounted initially for the female and native indigenous population, age dependence, population density, number of households and people per household.  The composite indicator of economy  includes indicators to assess the economic health of the community (Burton,
C. G., and  Silva, V., 2016) . The single indicators considered for this composite indicator were population unemployed, looking for employment, unsatisfied basic needs (UBN), and impoverished. Poverty is an important aspect to consider because of its direct association with access to resources, which affects coping with the impacts of disasters (Fatemi et al., 2017). The composite indicator of infrastructure  considers the access to basic services (Contreras et al., 2020b). The composite indicator

---

[2] https://sara.openquake.org/development_of_indicators_of_social_vulnerability

of education links the educational level and the socioeconomic status, mitigation, and recovery potential (Burton, C. G., &
Silva, V., 2016). It is assumed that lower education level results in lower-income, poor ability to understand emergencies, and
low capacity to recover after a disaster (Cutter et al., 2003). The composite indicator of health includes the indicators related
to access to health facilities and health care (Contreras et al., 2020b). The lack of access to healthcare increases people's
susceptibility to the potential impact of disasters (Fatemi et al., 2017). Considering the aforementioned composite indicators
and the availability of information for the region, a total of 26 indicators were selected initially. However, to avoid problems
interpreting the model and overfitting, we checked the multicollinearity by looking at the variance inflation factor (VIF) of
each variable and indicator (see Table 5). The VIF was identified in a linear regression that included collinearity diagnostics
produced in SPSS (Field, 2005). We excluded those variables and indicators that were potentially correlated with others and
those that did not add significant information according to the collinearity diagnostics (Table 6). Eventually, the model
included 10 independent and relevant variables and indicators to estimate the SV in the case study area (Table 7).

**Table 5.** Variance inflation factors (VIF)

| | **Coefficients**[a] | | | | | | |
|---|---|---|---|---|---|---|---|
| | Unstandardized Coefficients | | Standardized Coefficients | t | Sig. | Collinearity Statistics | |
| Model | B | Std. Error | Beta | | | Tolerance | VIF |
| 1  (Constant) | 0.728 | 0.000 | | | | | |
| Indigenous population | 1.178 | 0.000 | 0.124 | | | 0.047 | 21.282 |
| Population density (inhabitants/km2) | -3.327 | 0.000 | -0.047 | | | 0.030 | 33.453 |
| Number of people per household | -3.378 | 0.000 | -0.012 | | | 0.029 | 34.994 |
| Population unemployed | 10.076 | 0.000 | 0.134 | | | 0.191 | 5.222 |
| Population with unsatisfied basic needs | -8.921 | 0.000 | -0.099 | | | 0.023 | 43.108 |
| Total population in poverty | 7.205 | 0.000 | 0.129 | | | 0.238 | 4.203 |
| Households with no electric energy access | 4.234 | 0.000 | 0.153 | | | 0.078 | 12.894 |
| No sewage system | 1.160 | 0.000 | 0.123 | | | 0.351 | 2.848 |
| Illiteracy rate | -1.619 | 0.000 | -0.027 | | | 0.209 | 4.785 |
| Deceased due to COVID-19 | 11.379 | 0.000 | 0.710 | | | 0.044 | 22.967 |
| a. Dependent Variable: SV | | | | | | | |

**Table 6.** Excluded variables/indicators

| Model | Beta In | t | Sig. | Partial Correlation | Collinearity Statistics | | |
|---|---|---|---|---|---|---|---|
| | | | | | Tolerance | VIF | Minimum Tolerance |
| 1  Female population | .[b] | | | | 0.000 | | 0.000 |
| Age dependence | .[b] | | | | 0.000 | | 0.000 |
| Total population | .[b] | | | | 0.000 | | 0.000 |
| Number of households | .[b] | | | | 0.000 | | 0.000 |
| Population looking for employment | .[b] | | | | 0.000 | | 0.000 |
| Household with computer and internet | .[b] | | | | 0.000 | | 0.000 |
| Households with access to improved water source | .[b] | | | | 0.000 | | 0.000 |
| Education level completed primary | .[b] | | | | 0.000 | | 0.000 |
| Education level secondary | .[b] | | | | 0.000 | | 0.000 |
| Population enrolled in education institution | .[b] | | | | 0.000 | | 0.000 |
| Hospital , clinics per 1000 population | .[b] | | | | 0.000 | | 0.000 |
| Population with no healthcare | .[b] | | | | 0.000 | | 0.000 |
| Population registered to national healthcare | .[b] | | | | 0.000 | | 0.000 |
| COVID-19 cases confirmed | .[b] | | | | 0.000 | | 0.000 |
| COVID-19 cases active | .[b] | | | | 0.000 | | 0.000 |
| People recovered from COVID-19 | .[b] | | | | 0.000 | | 0.000 |

a. Dependent Variable: SV

b. Predictors in the Model: (Constant), People dead due to COVID-19, Total population in poverty, No sewage system, Number of people per household, Native indigenous population, Population unemployed, Illiteracy rate, Population density (inhabitants/km2), Households with no electric energy access, population with unsatisfied basic needs

**Table 7.** Selected variables/indicators

| | Model | Collinearity Diagnostics[a] | | | | | | | | | | |
|---|---|---|---|---|---|---|---|---|---|---|---|---|
| | | 1 | | | | | | | | | | |
| | | 1 | 2 | 3 | 4 | 5 | 6 | 7 | 8 | 9 | 10 | 11 |
| | Eigenvalue | 8.495 | 1.004 | 0.740 | 0.517 | 0.102 | 0.066 | 0.044 | 0.017 | 0.011 | 0.004 | 4.453E-05 |
| | Condition Index | 1.000 | 2.909 | 3.389 | 4.053 | 9.134 | 11.307 | 13.893 | 22.300 | 27.201 | 48.357 | 436.754 |
| Variance Proportions | (Constant) | 0.00 | 0.00 | 0.00 | 0.00 | 0.00 | 0.00 | 0.00 | 0.00 | 0.00 | 0.00 | 1.00 |
| | Native indigeneous population | 0.00 | 0.02 | 0.00 | 0.02 | 0.03 | 0.00 | 0.00 | 0.00 | 0.00 | 0.04 | 0.89 |
| | Population density (inhabitants/km2) | 0.00 | 0.00 | 0.00 | 0.00 | 0.00 | 0.00 | 0.00 | 0.01 | 0.00 | 0.24 | 0.74 |
| | Number of people per household | 0.00 | 0.00 | 0.00 | 0.00 | 0.00 | 0.00 | 0.00 | 0.00 | 0.00 | 0.00 | 1.00 |
| | Population unemployed | 0.00 | 0.00 | 0.00 | 0.00 | 0.00 | 0.03 | 0.00 | 0.21 | 0.00 | 0.75 | 0.00 |
| | Population with unsatisfied basic needs | 0.00 | 0.00 | 0.00 | 0.00 | 0.00 | 0.00 | 0.00 | 0.01 | 0.02 | 0.00 | 0.96 |

| | | | | | | | | | | | |
|---|---|---|---|---|---|---|---|---|---|---|---|
| Total population in poverty | 0.00 | 0.00 | 0.00 | 0.00 | 0.01 | 0.15 | 0.10 | 0.01 | 0.12 | 0.45 | 0.16 |
| Households with no electric energy access | 0.00 | 0.00 | 0.00 | 0.00 | 0.09 | 0.04 | 0.01 | 0.08 | 0.00 | 0.73 | 0.05 |
| No sewage system | 0.00 | 0.08 | 0.15 | 0.13 | 0.00 | 0.02 | 0.01 | 0.07 | 0.03 | 0.08 | 0.42 |
| Illiteracy rate | 0.00 | 0.00 | 0.00 | 0.00 | 0.01 | 0.02 | 0.16 | 0.01 | 0.32 | 0.02 | 0.47 |
| People dead due to COVID-19 | 0.00 | 0.00 | 0.01 | 0.02 | 0.04 | 0.00 | 0.00 | 0.11 | 0.03 | 0.08 | 0.71 |
| a. Dependent Variable: SV | | | | | | | | | | | |

Much of the information used for the indicators is from the national census (DANE, 2018) database, studies such as the multipurpose survey (EM2017), which examined the quality of life of households in Bogotá and surrounding areas[3], and the analysis of the characteristics of the population in Sabana Centro (Sabana Centro Cómo Vamos, 2019).

The min-max normalization was used to standardize the SV indicators from zero to one to estimate the SVI per municipality. Higher scores indicate more socially vulnerable municipalities, and lower scores reflect less vulnerable ones. Then, the indicators were integrated by summing them with equal weight, as followed in Contreras et al. (2020c). The resulting SVI index is therefore used to adjust the percentage of economic losses with respect to the costs presented by the building inventory, i.e., multiplying them by (1+SVI) (Carreño et al., 2007).

## 4. Results

This section presents the results of this study in terms of median building damage and median economic losses for each municipality. First, the Mw 5.95 earthquake scenario results in Chía are introduced to illustrate the methodology. Then, the results of the eighteen seismic risk scenarios for Sabana Centro are presented. These eighteen scenarios are defined based on the earthquake events presented in Table 2. These did not include directivity effects because there was insufficient information available for a reliable model. The economic losses are adjusted based on the SVI discussed in section 3.4 and are also presented. For this purpose, before presenting the economic losses, the SVI calculation will be introduced.

### 4.1. Damage forecast

The predicted damage for the Mw 5.95 earthquake scenario in Chía considered for the region is presented in Table 8. Mean and standard deviation are presented for each of the damage states considered in the scenario. The respective distribution of ground motion field for the 5.95 earthquake is presented in Figure 11.

---

[3] https://sdpbogota.maps.arcgis.com/apps/MapJournal/index.html?appid=c984e588b0764efbb424ffc2207b5cf6

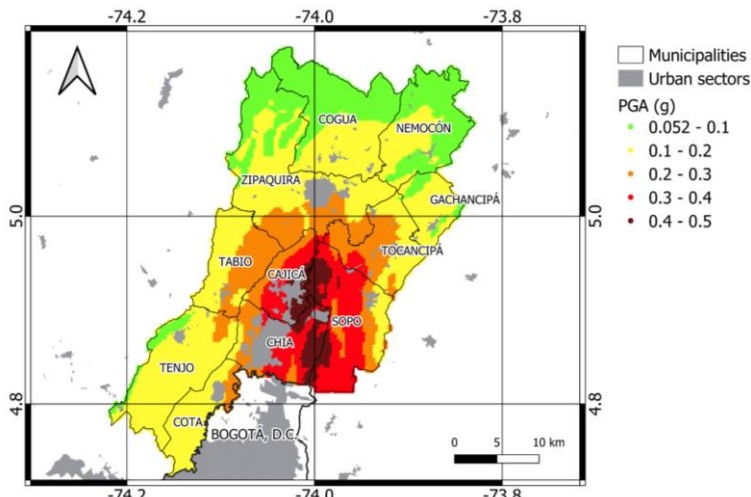

**Figure 11.** Estimated maximum peak ground acceleration (PGA) in bedrock for the Mw 5.95 Chia earthquake. The highest acceleration is presented in dark red and the lowest in green.

**Table 8.** Number and percentage of buildings expected to suffer damage in the region after the Mw 5.95 earthquake scenario in Chía. The mean and standard deviation (Stdv) for each of the GMPE and damage states are presented. The mean[a] is calculated with the corresponding weights for each GMPE (Abrahamson et al. (2014): 0.35 and Cauzzi et al. (2014): 0.65). The total number and percentage of buildings for each damage state are at the end of the table.

| Municipality | GMPE | No damage | | Slight | | Moderate | | Extensive | | Collapse | |
|---|---|---|---|---|---|---|---|---|---|---|---|
| | | Mean | Stdv | Mean | Stdv | Mean | Stdv | Mean | Stdv | Mean | Stdv |
| Cajicá | Abrahamson | 5520 | 904 | 3700 | 420 | 1133 | 298 | 560 | 193 | 590 | 332 |
| | Cauzzi | 2745 | 1067 | 3015 | 727 | 1486 | 347 | 1084 | 268 | 3176 | 1270 |
| | Mean[a] | 3716 | | 3255 | | 1362 | | 901 | | 2271 | |
| Chía | Abrahamson | 9173 | 1493 | 5809 | 689 | 1773 | 486 | 872 | 311 | 916 | 545 |
| | Cauzzi | 4817 | 1779 | 4870 | 1170 | 2285 | 568 | 1645 | 427 | 4926 | 2084 |
| | Mean[a] | 6342 | | 5199 | | 2105 | | 1374 | | 3522 | |
| Cogua | Abrahamson | 2688 | 209 | 257 | 148 | 38 | 42 | 13 | 18 | 9 | 24 |
| | Cauzzi | 2712 | 211 | 233 | 147 | 37 | 43 | 13 | 20 | 10 | 28 |
| | Mean[a] | 2704 | | 241 | | 37 | | 13 | | 10 | |
| Cota | Abrahamson | 3175 | 484 | 932 | 264 | 233 | 124 | 103 | 70 | 120 | 145 |
| | Cauzzi | 3313 | 508 | 817 | 284 | 206 | 121 | 99 | 71 | 128 | 168 |
| | Mean[a] | 3264 | | 858 | | 216 | | 100 | | 126 | |
| Gachancipá | Abrahamson | 1777 | 232 | 519 | 134 | 116 | 60 | 47 | 31 | 40 | 42 |
| | Cauzzi | 1637 | 289 | 530 | 153 | 152 | 74 | 77 | 45 | 103 | 101 |
| | Mean[a] | 1686 | | 526 | | 140 | | 66 | | 81 | |
| Nemocón | Abrahamson | 1408 | 164 | 337 | 102 | 66 | 40 | 25 | 19 | 18 | 21 |
| | Cauzzi | 1282 | 212 | 364 | 113 | 99 | 54 | 49 | 32 | 60 | 64 |
| | Mean[a] | 1326 | | 355 | | 88 | | 41 | | 45 | |
| Sopó | Abrahamson | 1258 | 341 | 1116 | 197 | 476 | 101 | 290 | 81 | 520 | 269 |
| | Cauzzi | 1403 | 428 | 1009 | 220 | 414 | 115 | 267 | 88 | 566 | 319 |
| | Mean[a] | 1352 | | 1047 | | 436 | | 275 | | 550 | |
| Tabio | Abrahamson | 1745 | 319 | 775 | 159 | 225 | 89 | 108 | 56 | 135 | 122 |
| | Cauzzi | 1918 | 346 | 668 | 180 | 185 | 89 | 93 | 55 | 124 | 126 |
| | Mean[a] | 1858 | | 705 | | 199 | | 98 | | 128 | |

| | | | | | | | | | | |
|---|---|---|---|---|---|---|---|---|---|---|
| Tenjo | Abrahamson | 1976 | 248 | 627 | 140 | 142 | 67 | 58 | 35 | 48 | 44 |
| | Cauzzi | 1678 | 330 | 657 | 165 | 210 | 84 | 117 | 56 | 189 | 156 |
| | Mean[a] | 1782 | | 647 | | 186 | | 96 | | 140 | |
| Tocancipá | Abrahamson | 2807 | 463 | 1443 | 222 | 406 | 141 | 187 | 86 | 189 | 147 |
| | Cauzzi | 2196 | 593 | 1361 | 303 | 513 | 159 | 317 | 118 | 646 | 418 |
| | Mean | 2410 | | 1390 | | 475 | | 272 | | 486 | |
| Zipaquirá | Abrahamson | 16638 | 1568 | 2067 | 1048 | 349 | 341 | 125 | 155 | 98 | 207 |
| | Cauzzi | 16873 | 1589 | 1847 | 1047 | 326 | 344 | 124 | 164 | 108 | 247 |
| | Mean[a] | 16791 | | 1924 | | 334 | | 125 | | 105 | |
| **Total** | **Number of buildings** | **43.230** | | **16,145** | | **5579** | | **3361** | | **7463** | |
| | **Percentage of buildings (%)** | **57.05** | | **21.31** | | **7.36** | | **4.44** | | **9.85** | |

In the region, 42.95% of the buildings considered in the exposure model are expected to suffer some degree of damage. This result represents 32,598 out of the 75,778 analyzed buildings. Table **8** shows that the type of damage with the highest occurrence is slight (21.31%), followed by collapse (9.85%); moderate (7.36%) and extensive damage (4.44%). Overall, 14.28% might suffer extensive or collapse damage, hence they will not fulfil their life safety functionality. Nearly ten percent of collapse rises concerns from a decision maker perspective, but two Colombian events put the results in perspective: the Mw 6.1 earthquake in Armenia (1999) and the Mw 5.5 earthquake in Popayán (1983). In the former, the records indicate that 17551 buildings were destroyed, 18421 had severe damage and 43474 had moderate damage. In the latter, which occurred at an estimated depth between 12 km and 15 km, 12% of buildings suffered complete damage. In both earthquakes, damage concentrated in unreinforced masonry buildings, constructed prior to the enactment of the Colombian seismic design code in 1998. More than 60% of the building stock in Sabana Centro is comprised of that type of buildings, and what is more, 35% are two- and three-story houses (Table 4), which are more vulnerable than those of one-story houses (Heresi and Miranda., 2022). These buildings are expected to withstand significant damage during an earthquake such as the Chía Mw 5.95 shown here, which is similar in magnitude and depth to the Armenia earthquake and for which the percentage of collapse herein presented is similar to that from the Popayán earthquake.

In terms of municipalities, the higher damage occurs in Chía and Cajicá, with 3522 and 2271 collapsed buildings. Compared to the total buildings of each municipality, collapses account for 19.0% and 19.7%, respectively. Overall, they account for 5793 out of the 7463 collapsed buildings for this scenario (77%). In contrast, Cogua was the municipality with the least number of damaged buildings, as roughly 90% of the inventory did not experience any type of damage and only 0.33% of them collapsed. Nemocón had the least damages after Cogua, with 2.4% of collapses. These results are reasonable because Chía and Cajicá are closer to the epicenter in this scenario and they have the highest building inventory of the region, together with Zipaquirá. Besides, a significant part of their building inventory is comprised of nonductile unreinforced masonry. On the other hand, despite having a similar distribution of the building inventory, Cogua and Nemocón are the farthest municipalities from the epicenter. The main difference between these two is that Nemocón has softer soils, with roughly one fourth of the municipality under 180 m/s, thus the higher percentage of collapses.

The highest concentration of building collapses in the region is expected for houses constructed of unreinforced masonry (Table 9), mostly involving non-ductile unreinforced masonry walls of one and two stories (22.8% and 30.5% respectively). Notably, these two types of buildings account for 53.30% of collapses. Three-story unreinforced masonry houses account for 10.12% of buildings, making the overall contribution of this structural system more than six out of ten collapses. The percentage of three-story houses collapsed was smaller than the one from two story houses (which are less vulnerable) because three-story houses are less frequent in the region.

**Table 9.** Expected number and percentage of buildings by class that might collapse as a result of the Mw 5.95 earthquake scenario in Chía, and their corresponding economic losses presented in Millions of dollars and as percentage of total losses.

| Building classes | Number of collapsed buildings | Percentage out of the collapsed buildings (%) | Direct economic losses (M.USD) | Percentage out of the economic losses (%) |
|---|---|---|---|---|
| CR/LDUAL/DUC/H:4,7 | 2 | 0.02 | 1.49 | 0.17 |
| CR/LFINF/DUC/H:1 | 24 | 0.32 | 6.77 | 0.75 |
| CR/LFINF/DUC/H:2 | 68 | 0.91 | 19.74 | 2.19 |
| CR/LFINF/DUC/H:3 | 27 | 0.36 | 7.88 | 0.87 |
| CR/LFINF/DUC/H:4,7 | 42 | 0.56 | 45.24 | 5.02 |
| CR/LFM/DNO/H:1 | 77 | 1.03 | 18.17 | 2.02 |
| CR/LFM/DNO/H:2 | 225 | 3.01 | 53.28 | 5.92 |
| CR/LFM/DNO/H:3 | 89 | 1.19 | 21.12 | 2.35 |
| CR/LFM/DUC/H:4,7 | 31 | 0.42 | 33.13 | 3.68 |
| CR/LWAL/DUC/H:4,7 | 10 | 0.14 | 11.81 | 1.31 |
| CR/LWAL/DUC/H:1 | 1 | 0.02 | 0.63 | 0.07 |
| CR/LWAL/DUC/H:2 | 9 | 0.12 | 3.79 | 0.42 |
| CR/LWAL/DUC/H:3 | 3 | 0.04 | 1.27 | 0.14 |
| MCF/LWAL/DNO/H:1 | 156 | 2.09 | 16.05 | 1.78 |
| MCF/LWAL/DNO/H:2 | 868 | 11.63 | 51.23 | 5.69 |
| MCF/LWAL/DNO/H:3 | 513 | 6.88 | 53.45 | 5.94 |
| MCF/LWAL/DUC/H:1 | 17 | 0.23 | 5.02 | 0.56 |
| MCF/LWAL/DUC/H:2 | 93 | 1.24 | 25.95 | 2.88 |
| MCF/LWAL/DUC/H:3 | 42 | 0.56 | 11.61 | 1.29 |
| MR/LWAL/DUC/H:1 | 16 | 0.21 | 5.56 | 0.62 |
| MR/LWAL/DUC/H:2 | 45 | 0.6 | 15.98 | 1.77 |
| MR/LWAL/DUC/H:3 | 17 | 0.23 | 5.99 | 0.67 |
| MUR/LWAL/DNO/H:1 | 1702 | 22.8 | 155.82 | 17.3 |
| MUR/LWAL/DNO/H:2 | 2276 | 30.5 | 223.88 | 24.86 |
| MUR/LWAL/DNO/H:3 | 755 | 10.12 | 75.51 | 8.39 |
| MUR-ADO/LWAL/DNO/H:1 | 33 | 0.45 | 2.6 | 0.29 |
| MUR-ADO/LWAL/DNO/H:2 | 32 | 0.43 | 2.73 | 0.3 |
| MUR-STDRE/LWAL/DNO/H:1 | 75 | 1.01 | 5.84 | 0.65 |

| | | | | |
|---|---|---|---|---|
| MUR-STDRE/LWAL/DNO/H:2 | 105 | 1.41 | 8.65 | 0.96 |
| MUR-STRUB/LWAL/DNO/H:1 | 22 | 0.3 | 1.8 | 0.2 |
| MUR-STRUB/LWAL/DNO/H:2 | 30 | 0.4 | 2.47 | 0.27 |
| W/WLI/DUC/H:1 | 24 | 0.32 | 2.44 | 0.27 |
| W/WLI/DUC/H:2 | 33 | 0.44 | 3.61 | 0.4 |
| **Total** | **7463** | **100** | **900.49** | **100** |

The number of expected damaged buildings for each municipality due to the Chía Mw 5.95 scenario is presented in Figure 12. It also shows their percentages for each damage state relative to the province's total. These results show that Chía, Cajicá, and Sopó have the highest percentage of collapsed buildings, with 47.20%, 30.43%, and 7.37%, respectively. In this scenario, the least affected municipalities are Cogua and Nemocón, with less than 1% of collapse percentages.

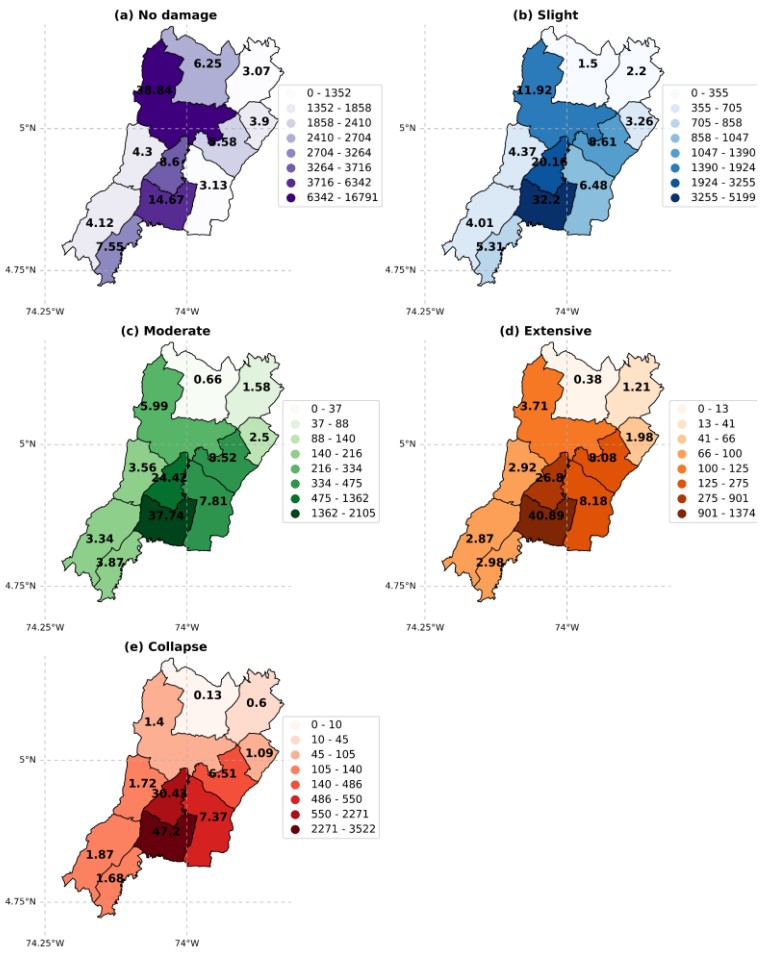

**Figure 12.** Number of buildings expected to experience: (a) no damage, (b) slight, (c) moderate, (d) extensive damage, or (e) collapse as a result of the Mw 5.95 earthquake scenario in Chía. The corresponding percentage values of buildings are presented within each municipality.

The damage calculations were conducted for all the seismic events presented in Table 2. Table 10 shows the resulting percentage of buildings that suffer damage for each of the eighteen seismic risk scenarios. The results show that the worst scenario for the region is the Mw 6.95 Cota, which has 21.37% collapsed buildings, and the highest percentages of severe and moderate damage. Interestingly, the Mw 6.85 Tocancipá had 8.12% of collapsed buildings, roughly 2.5 times less than the Mw 6.95 Cota. This difference is a consequence of the uneven distribution of the buildings stock in the region, as nearly 40% is in Chía and Cajicá, which are close to Cota. A similar situation occurs for the earthquakes of Mw 6.25 in Sopó, Tabio and Cajicá.

**Table 10.** Expected damage for each of the eighteen scenarios presented in Table 2. The name of the scenarios has the magnitude of the events and the municipality where they are located.

| Scenario | Percentage of buildings for each damage state | | | | |
|---|---|---|---|---|---|
| | No damage | Slight | Moderate | Extensive | Collapse |
| 5.35 Tenjo | 76.99 | 14.41 | 3.86 | 1.95 | 2.79 |
| 5.55 Cota | 74.44 | 15.17 | 4.35 | 2.32 | 3.72 |
| 5.65 Zipaquirá | 62.77 | 22.12 | 6.48 | 3.41 | 5.23 |
| 5.95 Chía | 57.05 | 21.31 | 7.36 | 4.44 | 9.85 |
| 5.95 Gachancipá | 66.78 | 19.60 | 5.66 | 3.01 | 4.94 |
| 5.95 Tenjo | 73.69 | 15.72 | 4.41 | 2.35 | 3.83 |
| 6.15 Tocancipá | 63.26 | 20.44 | 6.25 | 3.49 | 6.56 |
| 6.25 Nemocón | 65.26 | 19.73 | 5.81 | 3.19 | 6.01 |
| 6.25 Sopó | 44.90 | 25.57 | 9.48 | 5.90 | 14.14 |
| 6.25 Tabio | 54.11 | 24.11 | 8.02 | 4.64 | 9.12 |
| 6.35 Cajicá | 38.73 | 26.89 | 10.63 | 6.83 | 16.92 |
| 6.35 Cogua | 47.20 | 26.56 | 9.49 | 5.66 | 11.09 |
| 6.45 Cogua | 39.74 | 27.14 | 10.61 | 6.74 | 15.77 |
| 6.55 Sopó | 35.51 | 26.97 | 11.08 | 7.29 | 19.15 |
| 6.65 Nemocón | 51.80 | 24.04 | 8.29 | 4.95 | 10.92 |
| 6.65 Tabio | 42.97 | 25.86 | 9.84 | 6.23 | 15.10 |
| 6.85 Tocancipá | 53.10 | 25.37 | 8.57 | 4.84 | 8.12 |
| 6.95 Cota | 32.49 | 26.86 | 11.51 | 7.78 | 21.37 |

Figure **13** shows the variability of the results for each damage state of Table 10. The results illustrate the notable variability of the damage estimates between the different seismic scenarios. This variability is higher for the No damage state, which ranges between 32.49% and 76.99%, corresponding to the Mw 6.95 Cota and the Mw 5.35 Tenjo scenarios. These two also had the

highest and smallest percent of collapse, respectively. The median results of the eighteen scenarios for the damage states are 53.6%, 24.1%, 8.1%, 4.7% and 9.5%, values that are close to those of the Mw 6.25 Tabio.

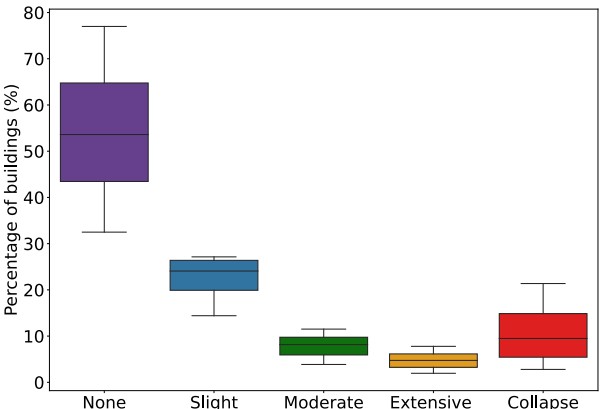

**Figure** 13. Percent of buildings expected to experience either no damage ('None'), slight, moderate, extensive damage, or collapse as a result of the 18 earthquakes scenarios presented in Table 2.

## 4.2. Social Vulnerability Index (SVI)

According to the methodology presented in section 3.4, the SVI is calculated and shown in Table 11.

**Table 11.** SVI for the municipalities of the Sabana Centro province

| Composite indicators | **Municipalities** | | | | | | | | | | |
|---|---|---|---|---|---|---|---|---|---|---|---|
| | Cajicá | Chía | Cogua | Cota | Gachancipá | Nemocón | Sopó | Tabio | Tenjo | Tocancipá | Zipaquirá |
| Population | 0.47 | 0.30 | 0.62 | 0.69 | 0.57 | 0.32 | 0.36 | 0.31 | 0.00 | 0.46 | 0.68 |
| Economy | 0.56 | 0.50 | 0.42 | 0.57 | 0.83 | 0.56 | 0.31 | 0.06 | 0.54 | 0.66 | 0.49 |
| Infrastructure | 0.69 | 0.50 | 0.13 | 0.25 | 0.00 | 0.11 | 0.01 | 0.05 | 0.09 | 0.14 | 0.269 |
| Education | 0.06 | 0.00 | 1.00 | 0.26 | 0.41 | 0.96 | 0.28 | 0.33 | 0.26 | 0.27 | 0.28 |
| Health | 0.33 | 0.70 | 0.09 | 0.06 | 0.00 | 0.03 | 0.08 | 0.03 | 0.05 | 0.15 | 1.00 |
| Index | 0.42 | 0.40 | 0.45 | 0.37 | 0.36 | 0.40 | 0.21 | 0.16 | 0.19 | 0.34 | 0.54 |

Considering population, the most vulnerable municipality is Cota, and the least vulnerable is Tenjo. Regarding economy, the most vulnerable is Gachancipá, and the least vulnerable is Tabio. In the case of infrastructure, the most vulnerable municipality is Cajicá, and the less is Gachancipá. In terms of education, the municipality of Cogua is the most vulnerable, followed by Nemocón, , unlike Chía. The health composite indicator shows that Zipaquirá is the most vulnerable municipality, and the less one is Gachancipá. The economy composite indicator shows higher vulnerability indices than the other categories for most municipalities. Evaluating all of the categories, it was found that Zipaquirá is the municipality with the highest SVI, while Tabio is the least vulnerable

### 4.3. Economic losses from the Mw 5.95 Chía earthquake scenario

The direct total economic loss arising from the considered Mw 5.95 Chía scenario is US$900.49 million, which represents 14.41% of the total replacement cost of the building inventory. The municipalities that contribute the most to these losses are Chía, with 52.60% and Cajicá, with 23.97%, as shown in Table 12. This result was expected since these municipalities' urban growth is high compared to the other municipalities. The smallest contribution comes from Cogua and Nemocón, with 0.06% and 0.18% respectively. Overall, in the case of this Mw 5.95 earthquake scenario, the direct economic loss in terms of replacement costs would be approximately 21% of the region's GDP. The economic losses for each municipality are presented in Table 12 and Figure 14a. Cajicá is the one with the highest percentage of losses, with 22.66% of the replacement costs. Other municipalities for which high economic losses are expected are Chía and Sopó, with 21.71% and 17.22%, respectively. The municipalities with the lowest losses are Zipaquirá (0.93%) and Cogua (0.67%).

**Table 12.** Economic losses for the region as a result of the considered Mw 5.95 earthquake scenario. Economic losses with SV consider the percentage of losses with respect to the total losses per municipality and are adjusted using the SVI. Consequently, the losses in M.USD are adjusted.

| Municipality | Cost of building Inventory (M. USD) | Direct economic losses | | | Economic losses with social vulnerability | |
| --- | --- | --- | --- | --- | --- | --- |
| | | Losses (M. USD) | Losses with respect of the total (%) | Percentage of municipality cost (%) | Losses + SVI (%) | Losses after considering the SVI (M. USD) |
| Cajicá | 952.41 | 215.80 | 23.97 | 22.66 | 32.19 | 306.61 |
| Chía | 2182.19 | 473.68 | 52.60 | 21.71 | 30.38 | 662.98 |
| Cogua | 87.74 | 0.58 | 0.06 | 0.67 | 0.97 | 0.85 |
| Cota | 646.16 | 24.19 | 2.69 | 3.74 | 5.11 | 33.03 |
| Gachancipá | 49.94 | 2.37 | 0.26 | 4.74 | 6.46 | 3.22 |
| Nemocón | 42.82 | 1.62 | 0.18 | 3.77 | 5.27 | 2.25 |
| Sopó | 252.55 | 43.48 | 4.83 | 17.22 | 20.81 | 52.56 |
| Tabio | 205.93 | 11.60 | 1.29 | 5.63 | 6.52 | 13.42 |
| Tenjo | 539.13 | 35.14 | 3.90 | 6.52 | 7.74 | 41.74 |
| Tocancipá | 742.61 | 86.95 | 9.66 | 11.71 | 15.65 | 116.21 |
| Zipaquirá | 547.03 | 5.08 | 0.56 | 0.93 | 1.43 | 7.85 |
| Total | 6248.51 | 900.49 | 100.00 | | | 1240.72 |

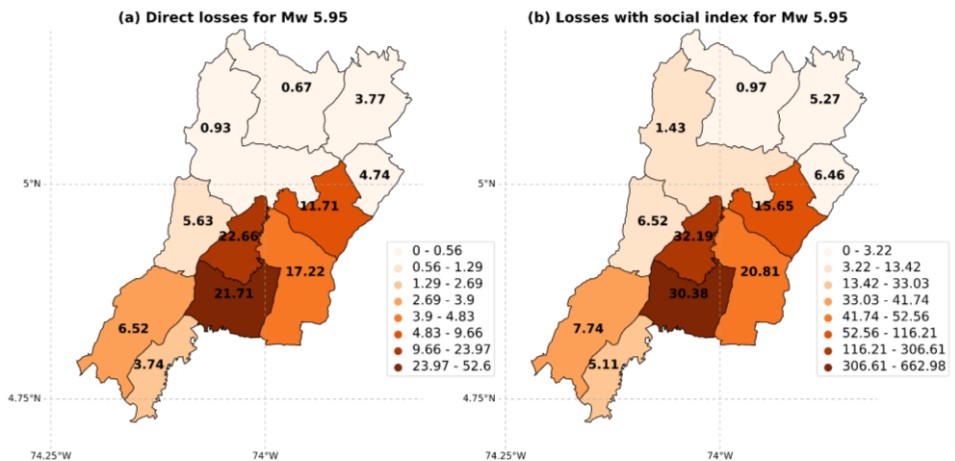

**Figure 14.** (a) Expected losses in millions of US dollars for the region as a result of the Mw 5.95 earthquake scenario and (b) the expected losses after considering the SVI. The percentage of losses with respect to the total is presented within the municipalities.

460    Figure 14(b) shows the adjusted economic losses by municipality and includes SV. After considering the SV in the region, the economic losses increase by 27.42%, from $900.49 to US$1240.72 million. The municipality that would have the highest economic losses is Cajicá (22.66% of the building replacement cost), with US$215.80 million. When the SV is included, its potential economic losses increase to US$306.61 million (32.19% of the replacement cost of Cajicá buildings). In the case of Zipaquirá, the most socially vulnerable municipality, the losses were initially US$5.08 million and increase to

US$7.85 million when the SVI is accounted for.

    The percentage of economic losses concerning the building types relative to the total losses in the region are shown in Table 13. Forty one percent of losses come from unreinforced masonry buildings (MUR/LWAL/DNO/H1 and MUR/LWAL/DNO/H2). This result is expected because this building type has a high seismic vulnerability, while the lowest contribution would come from CR/LDUAL/DUC/H:4,7 houses, with less than 0.02%. Three story non ductile houses

constructed in reinforced concrete frames and confined masonry experienced the highest collapsed buildings per taxonomy, with nearly 20% for both. In the case of the frames, 18.63% of two-story houses collapsed, more than twice of those of one story. In the case of nonductile confined masonry this ratio was five, while for ductile confined masonry it was four. These results agree with the findings by Heresi and Miranda (2022) and add evidence to the need of avoiding lumping low rise houses (1-3 stories) into one single taxonomy for seismic risk calculations.

**Table 13.** Expected number and percentage of buildings by class that might collapse as a result of the Mw 5.95 earthquake scenario in Chía and their respective economic losses.

| Building class | Number of collapsed buildings | Percentage of collapsed buildings (%) | Percentage of collapsed by taxonomy (%) | Economic Losses (M. USD) | Percentage out of the economic losses (%) |
|---|---|---|---|---|---|
| CR/LDUAL/DUC/H:4,7 | 2 | 0.02 | 11.63 | 1.49 | 0.17 |
| CR/LFINF/DUC/H:1 | 24 | 0.32 | 2.81 | 6.77 | 0.75 |
| CR/LFINF/DUC/H:2 | 68 | 0.91 | 5.61 | 19.74 | 2.19 |
| CR/LFINF/DUC/H:3 | 27 | 0.36 | 5.98 | 7.88 | 0.87 |
| CR/LFINF/DUC/H:4,7 | 42 | 0.56 | 8.96 | 45.24 | 5.02 |
| CR/LFM/DNO/H:1 | 77 | 1.03 | 9.01 | 18.17 | 2.02 |
| CR/LFM/DNO/H:2 | 225 | 3.01 | 18.63 | 53.28 | 5.92 |
| CR/LFM/DNO/H:3 | 89 | 1.19 | 19.67 | 21.12 | 2.35 |
| CR/LFM/DUC/H:4,7 | 31 | 0.42 | 6.74 | 33.13 | 3.68 |
| CR/LWAL/DUC/H:4,7 | 10 | 0.14 | 3.53 | 11.81 | 1.31 |
| CR/LWAL/DUC/H:1 | 1 | 0.02 | 0.72 | 0.63 | 0.07 |
| CR/LWAL/DUC/H:2 | 9 | 0.12 | 3.86 | 3.79 | 0.42 |
| CR/LWAL/DUC/H:3 | 3 | 0.04 | 3.51 | 1.27 | 0.14 |
| MCF/LWAL/DNO/H:1 | 156 | 2.09 | 2.53 | 16.05 | 1.78 |
| MCF/LWAL/DNO/H:2 | 868 | 11.63 | 12.31 | 51.23 | 5.69 |
| MCF/LWAL/DNO/H:3 | 513 | 6.88 | 19.40 | 53.45 | 5.94 |
| MCF/LWAL/DUC/H:1 | 17 | 0.23 | 1.40 | 5.02 | 0.56 |
| MCF/LWAL/DUC/H:2 | 93 | 1.24 | 5.33 | 25.95 | 2.88 |
| MCF/LWAL/DUC/H:3 | 42 | 0.56 | 6.46 | 11.61 | 1.29 |
| MR/LWAL/DUC/H:1 | 16 | 0.21 | 3.34 | 5.56 | 0.62 |
| MR/LWAL/DUC/H:2 | 45 | 0.60 | 6.74 | 15.98 | 1.77 |
| MR/LWAL/DUC/H:3 | 17 | 0.23 | 6.91 | 5.99 | 0.67 |
| MUR/LWAL/DNO/H:1 | 1702 | 22.80 | 11.55 | 155.82 | 17.30 |
| MUR/LWAL/DNO/H:2 | 2276 | 30.50 | 11.56 | 223.88 | 24.86 |
| MUR/LWAL/DNO/H:3 | 755 | 10.12 | 10.23 | 75.51 | 8.39 |
| MUR-ADO/LWAL/DNO/H:1 | 33 | 0.45 | 14.41 | 2.60 | 0.29 |
| MUR-ADO/LWAL/DNO/H:2 | 32 | 0.43 | 12.70 | 2.73 | 0.30 |
| MUR-STDRE/LWAL/DNO/H:1 | 75 | 1.01 | 11.14 | 5.84 | 0.65 |
| MUR-STDRE/LWAL/DNO/H:2 | 105 | 1.41 | 11.18 | 8.65 | 0.96 |
| MUR-STRUB/LWAL/DNO/H:1 | 22 | 0.30 | 10.71 | 1.80 | 0.20 |
| MUR-STRUB/LWAL/DNO/H:2 | 30 | 0.40 | 10.19 | 2.47 | 0.27 |
| W/WLI/DUC/H:1 | 24 | 0.32 | 1.57 | 2.44 | 0.27 |
| W/WLI/DUC/H:2 | 33 | 0.44 | 1.47 | 3.61 | 0.40 |
| **Total** | 7463 | 100.00 | | 900.49 | 100.00 |

## 4.4 Mean direct economic losses for the earthquake scenarios

The seismic ground motion fields expected for each earthquake scenario listed in Table 2 were simulated 1,000 times to account for their aleatoric uncertainty as advised by Silva (2016), making use of the OQ Engine. The physical vulnerability was calculated in a similar manner as for the Mw 5.95 earthquake scenario. Figure 15 shows the loss exceedance curves that describe the probability of exceeding a given percent of economic losses for each earthquake scenario.

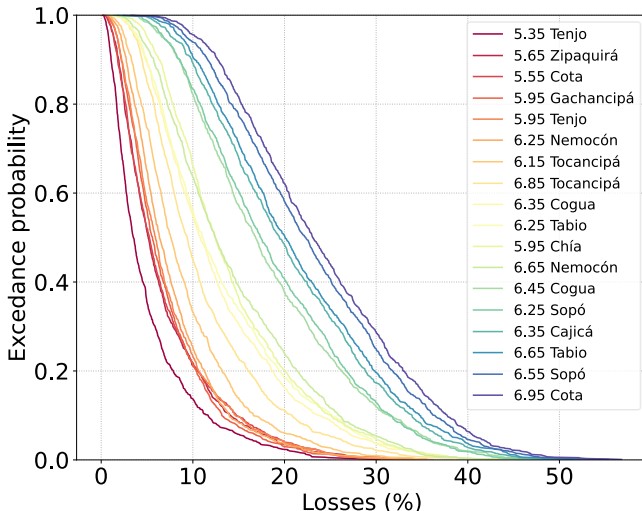

**Figure 15.** Loss-exceedance curves (LEC) as a function of percentage of economic losses in the region for the 18 earthquake scenarios considered (Table 2) whose ground motion fields were simulated 1000 times. The legend is sorted according to the line position in the figure from left to right.

Among the simulated scenarios, the most significant economic losses might occur with the scenario that considers an earthquake of Mw 6.95 in Cota and the smallest with the simulation of the earthquake of Mw 5.35 in Tenjo. For example, the probability that economic losses exceed 20% in the "6.95 Cota" scenario is 61%, while for the scenario "5.35 Tenjo", the probability is 3%. Besides magnitude, the uneven distribution of the building stock in the region is an important factor that exerts an influence on economic losses, as shown by the differences between the three Mw 6.25 events in Figure 15. The highest economic losses are in the municipality of Cota with 24.29% of losses (1517.56 US$ million). On average, economic losses when including social vulnerability increase by 37% as shown in Table 14.

**Table 14.** Expected Economic Losses in the region for each of the eighteen scenarios considered. Economic losses when including SV consider the percentage of losses with respect to the total losses per municipality and are adjusted with the SVI.

| Scenario | Direct economic losses | | Economic losses with SV | |
|---|---|---|---|---|
| | Losses (M.USD) | Percentage of Losses (%) | Losses after considering the SVI (M. USD) | Losses + SVI (%) |
| 5.35 Tenjo | 322.99 | 5.17 | 437.28 | 7.00 |
| 5.55 Cota | 434.35 | 6.95 | 588.35 | 9.42 |
| 5.65 Zipaquirá | 426.26 | 6.82 | 592.19 | 9.48 |
| 5.95 Chía | 900.49 | 14.41 | 1240.72 | 19.86 |
| 5.95 Gachancipá | 438.62 | 7.02 | 599.55 | 9.60 |
| 5.95 Tenjo | 461.39 | 7.38 | 615.75 | 9.85 |
| 6.15 Tocancipá | 571.58 | 9.15 | 778.38 | 12.46 |

| | | | |
|---|---|---|---|
| 6.25 Nemocón | 484.05 | 7.75 | 660.98 | 10.58 |
| 6.25 Sopó | 1181.64 | 18.91 | 1628.70 | 26.07 |
| 6.25 Tabio | 824.01 | 13.19 | 1120.51 | 17.93 |
| 6.35 Cajicá | 1295.51 | 20.73 | 1790.78 | 28.66 |
| 6.35 Cogua | 794.91 | 12.72 | 1101.68 | 17.63 |
| 6.45 Cogua | 1154.40 | 18.47 | 1596.73 | 25.55 |
| 6.55 Sopó | 1450.55 | 23.21 | 2002.18 | 32.04 |
| 6.65 Nemocón | 888.24 | 14.22 | 1211.52 | 19.39 |
| 6.65 Tabio | 1343.27 | 21.50 | 1824.25 | 29.19 |
| 6.85 Tocancipá | 695.64 | 11.13 | 949.69 | 15.20 |
| 6.95 Cota | 1517.56 | 24.29 | 2093.23 | 33.50 |

## 5. Discussion

### 5.1. Damage and losses

500 The findings presented in this work show that the Sabana Centro province is exposed to a considerable level of seismic risk. The simulations of eighteen seismic scenarios with a return period of 475 years extracted from a PSHA show that the damaged buildings ranged between 23% and 67.5% of the building stock depending on the earthquake epicenter. Worryingly, the median value of buildings that would experience extensive damage or collapse in the eighteen scenarios is 14.2%. This situation stems from the fact that 83% of the houses in the province are constructed using non-ductile structural systems. These houses

505 accounted for more than 90% of collapses in most of the scenarios. The damage results also highlight the importance of discretizing buildings with a same structural system by heights, at least for houses between one and three stories as suggested by Heresi and Miranda (2022) because two- and three-story houses had a significantly higher percentage of collapses compared to one story houses.

In terms of economic losses, the median expected cost of the eighteen earthquakes selected from the PSHA in the

510 province is 19% of its GDP, which accounts for US$ 809.46 million, almost 13% of the replacement cost of the building inventory. This result only accounts for the cost of physical damage of the building stock, however, and does not represent all of the potential impacts.

### 5.2. Effects of social vulnerability (SV)

Very few research studies has tested the correlation between social vulnerability (SV) and losses. To our best knowledge, the

515 relationship between SV and modeled losses has been so far informative rather than indicating that total losses (measured as dollar losses or debris generated) increase with SV (Schmidtlein et al., 2011). However, it was found that only relative losses (dollar losses per average family income) tend to increase with SV. Case study areas with a low SV tend to have more material goods with significant monetary value (dollar) exposed to risk, than areas with high SV. Therefore, we should expect a negative correlation between property losses and SV (Cutter and Finch, 2008). It is important to understand that while total loss (dollar)

in case study areas with high SV is lower, the impact of those losses in their communities is high (Schmidtlein et al., 2011). Integrating the level of SV to the physical losses will not produce a significant increase in the last ones, considering that they are negatively correlated (Cutter and Finch, 2008), but the result will be a more holistic risk assessment, also useful to prioritize actions at the regional level. Sabana Centro is a province with a significant level of SV representing many areas in Colombia and other countries in South America, with essential deficiencies in areas like health and education. The estimated integration

of SV with the economic losses increases them to 26% of the GDP, representing approximately 1.11 billion US-dollars. The results of this study show that including SV is important in risk analysis, as it allows one to go beyond only considering economic loss assessments with respect to physical damage.

### 5.3. Caveats and limitations

There are several limitations that should be addressed in future studies. One of them involves the selection of the building's

seismic fragility functions. Recent research has demonstrated that assumptions about several input parameters used in physical seismic vulnerability significantly influence risk assessment in urban areas (Hoyos and Hernández, 2021). One concern held by the authors is that the field observations and surveys show that a good part of the building stock of Sabana Centro is the result of informal construction. Presently, these buildings are constructed using either confined masonry or infilled RC frames due to the influence of the Colombian design code (which holds similarities with the ACI-318 (Arroyo et al., 2019)), which

forbade unreinforced masonry. Research about fragility functions like these buildings in Puerto Rico (Murray et al., 2022) and Villavicencio (Feliciano et al. 2022) show that the collapse probability may be even twice than that of code conforming buildings. The fragility functions by Martins and Silva. (2021) and Villar-Vega. (2017) used in this research do not account for the particularities of these buildings, thus the authors hold the hypothesis that the damage estimates should be considered as a lower bound.

It is important to mention that the exposure model for residential buildings developed in this study considered two types of exposure modelling approaches. On the one hand, the census-based part is a top-down approach from aggregated data. On the other hand, the rapid remote surveys constitute a bottom-up approach. Although the latter allowed an assessment of the validity of the assumed building classes, both approaches were not fully integrated through a probabilistic approach (Pittore et al., 2020). Although this method requires more computational efforts, it is worth exploring in future studies.

Another aspect that was not explicitly addressed in this study was the consideration of spatial cross-correlation models in modelling the ground motion fields. Several studies (e.g., Weatherill et al., 2015, Heresi and Miranda, 2022) have demonstrated the relevance of such models in seismic risk assessment for building portfolios when sets of fragility functions that consider several IM are implemented. Although when such models are accommodated, the loss outcomes typically show a greater dispersion (and more likely to give extreme values), while when such models are disregarded, the mean loss values

forecasted have been observed to be practically the same as for the cases when a spatial correlation model was used (e.g., Michel et al., 2017; Gomez Zapata et al, 2021). Therefore, the results presented in this study are still informative, but once again, we remark they should be treated as lower bounds for the considered risk scenarios.

Furthermore, due to the proximity of the study area to the Usme fault, near-fault effects might be expected. Hence, the study of possible directivity effects might be relevant for future studies. The evaluation of this feature has been shown to be relevant in both seismic ground motion (Türker et al., 2022) and earthquake loss models (Gentile and Galasso, 2021). Therefore, a better understanding of their role in risk scenarios will benefit the outlined results.

Another point is that this study did not consider human casualties. These were not included due to the lack of accurate information about housing occupation, since in this province, it is expected that more than one family share one dwelling. Notwithstanding, the median results of physical damage assessment for the eighteen scenarios suggest that 14.2% of buildings will have a seismic performance below the life safety level. The limited number and quality conditions of hospitals and health care facilities in the province would further exacerbate the potential impact of an earthquake on the population.

## 6.   Conclusions and future work

This paper presented the results of a seismic risk assessment for the Sabana Centro province in Colombia, which also accounted for the effects of SV. Eighteen earthquake scenarios with a return period of 475 years were selected from a hazard disaggregation study. Each scenario was simulated 1000 times using the OQ Engine to calculate the physical damage and economic losses. These were adjusted based on a SVI that included the effects of the ongoing COVID-19 pandemic. The key findings from the results of this study are as follows:

- Sabana Centro is a region in a high seismic risk. Ten percent of the buildings would collapse and 4.44% would experience extensive damage considering the 5.95 Mw Chía scenario. The damage is concentrated on non-ductile unreinforced masonry houses, which account for 63.4% of the building stock. The most significant contribution to economic losses (76.57%) comes from the municipalities of Chía and Cajicá. Overall, losses for this scenario represent 21% of the region GDP

- The mean expected economic losses of the eighteen scenarios range between  US$ 322.99 and US$ 1517.56 million, which represents 5.17% and 24.29% of the replacement cost of the building inventory, which represent between 8% and 38% of the region's GDP.

- Incorporating the SV plays an important role in loss estimation. The adjusted economic losses for the eighteen scenarios region range between US$ 437 and US$ 2093 million, om average a 36.6% increase compared to the losses from building damage.

Overall, these results show that a seismic event corresponding to the design earthquake (475 years return period) would cause significant damage to the infrastructure and severe economic and social losses. Given the prevalence of unreinforced masonry houses, an effective mitigation strategy for this region is to develop seismic retrofitting programs for these buildings, especially for municipalities with higher population growth, which contribute the most to damage and losses.

The development of this study revealed two prominent areas for future research. The first is developing a robust framework to incorporate SV into the loss estimations, with a strong basis on how each social category should be weighted. 585 Second, despite a careful selection of the fragility functions based on literature review, the estimations of this study can be further refined by using a more complete dataset with fragility functions developed explicitly for Colombia, particularly for older masonry houses.

*Code and data availability.* The building surveys have been made available in the open repository: Arroyo et al. (2022).

*Competing interests.* The authors declare that they have no conflict of interest. The funders had no role in the design of the study; in the 590 collection, analyses, or interpretation of data; in the writing of the manuscript; or in the decision to publish the results.

*Funding.* The first two authors received funding from the research division of the Universidad de La Sabana, internal grant ING-260-2020. Tamara Cabrera was funded by the Research Center for Integrated Disaster Risk Management (CIGIDEN), ANID/FONDAP/15110017. The contribution from Diana Contreras was funded by the Engineering and Physical Sciences Research Council (EPSRC) [Grant No.: 595 EP/P025641/1]. Juan Camilo Gómez Zapata received financial support for the research from the RIESGOS 2.0 project, funded by the German Federal Ministry of Education and Research (BMBF) Grant No. 03G0905A-H, as part of the funding program CLIENT II – International Partnerships for Sustainable Innovations. This same project provided funds for its publication.

*Acknowledgements.* The authors would like to thank the Colombian Geological Survey for their support in the characterization of the hazard 600 for the study area. We thank the Universidad de la Sabana, especially the work team of "Sabana Centro Cómo Vamos" for the data made available for the social vulnerability assessment. Our gratitude also goes to Fabrice Cotton for the advice during the elaboration of this study. We also thank Kevin Fleming for the careful proofreading.

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
