# Peer review of "Seismic risk scenarios for the residential buildings in the Sabana Centro province in Colombia"

_Natural Hazards and Earth System Sciences, 2022_

## Community Comment (CC1)

**Review nhess-2022-73**

The study deals with the analyses of various seismic risk scenarios for the Sabana Centro region in Colombia, located in the northern region of the capital city Bogotá, which concentrates important industrial facilities, educational facilities. It is an interesting study that follows the state-of-the-art procedures of scenario risk analyses (at least up to the computation of direct losses), for a region that hasn't been studied before, and thus is a good contribution to the scientific literature that communicates the seismic risk in the country.

Having said this, there are some comments about important issues in the study, that hopefully will help improve its clarity, coherence, and thoroughness. It must be said that after doing the review, the reviewer saw that many of the comments and limitations of the study were included in the discussion section as further developments, however there are many that should be included to make the results sound and representative of the region of study, otherwise many of the presented results could be very misleading.

**Section 2:**

It would be interesting to include why the Sabana Centro region is of particular interest. In previous studies of the major cities, generated GDP or % of the population in comparison with the whole country were presented as reasons for the study of a particular city or region. Additionally, it is mentioned that it concentrates many economic and industrial activities, but at the end the analysis only deals with the residential building stock.

Line 105: "The majority of the building stock of the region is comprised of one- and two-story houses" It would be good to show a reference with the numbers based on the 2018 Census for this. It is interesting that this is mentioned and still no two-story houses are considered in the analyses.

**Section 3.1 Seismic hazard:**

Line 171: "eighteen crustal events were selected from the catalogue to be used in this study" Does this mean that only 'historical' events included in the catalogue were included? No new possible events from the event-based tables from the PSHA model?

Table 2: Include column with the distance to the population centroid taken as reference point in the disaggregation to be able to compare this selection of scenarios with the disaggregation graphs presented in Figure 4 and their representativeness in the overall 475-years return period hazard in the region.

Given the proximity of the events based on the disaggregation, directivity effects should be considered for some of these events. As further seen in the discussion this was not considered but it would be good to mention it and not leaving it till the discussion as a further development.

**Section 3.2 Exposure model for the residential building stock:**

Was the replacement cost updated to 2021? 2022? In which way was this done if indeed it has been updated? If not, it should be done and explained.

Were the new inflated exposure building numbers (based on population as proxy) in any way compared to the dwellings or building numbers reported in the 2018 Census for these regions?

Table 4 only considers unreinforced masonry of 1 storey, which is known to be less vulnerable than the unreinforced masonry of 2 stories, which is actually more common in many urban areas. This typology should be included (assuming something probably based on census data or the surveys), as in the region it is very common to find 2-storey, in some cases more than single storey houses (as previously mentioned in the study also). In the current version, the study may be underestimating the losses.

**Section 3.3 Physical vulnerability of residential building stock to seismic ground shaking**

Chilean wood structures are known to be in better shape than those in Colombia, and they consider a different type of construction technique. The same goes to the curves used in HAZUS, which are not as representative of the local conditions and may be underestimating the risk. If they are going to be used a more thorough explanation of the limitation of using them should be included.

Table 5 needs a clarification of what each curve considers in each of the damage states. If the vulnerability model is considering a unique consequence model, there may be incompatibility between the loss ratios of the derived vulnerabilities, as each one considers each damage state in a specific different way. This is one of the main issues when combining vulnerability functions from different sources. This is particularly true given the damage results of the studies are shown considering these categories of the damage states. The reviewer saw this mentioned in the further developments of the discussion, but is should be included in the computation of the vulnerability curves here in some way, for the results to be coherent.

Given the exposure is not considering separately the 2-storey housing, there are no vulnerabilities for 2-storeys considered, even when it is more common in the urban environment than the single storey houses. This should be included.

**Section 3.4. Social Vulnerability (SV)**

The main criticism of the paper is in the consideration of the social vulnerability index as a percentage increase using this expression (1+SVI). As stated in the study "The min-max normalization was used to standardize the SV indicators from zero to one to estimate the SVI per municipality. Higher scores indicate more socially vulnerable municipalities, and lower scores reflect less vulnerable ones. Then, the indicators were integrated by summing them with equal weight, as followed in Contreras et al. (2020c). The resulting SVI index is therefore used to adjust the percentage of economic losses with respect to the costs presented by the building inventory, i.e., multiplying them by (1+SVI) (Carreño et al., 2007)." The problem with this is that there is no analysis

done on the significance of the variables included within the study and no way to know if there are variables that shouldn't be included and if anything is counted double. Additionally, considering this index as a "percentage increase" is extremely misleading. If there was a way to correlate the SVI of each variable in economic terms to the direct economic loss, then this could be done. But this is not done and there is no parametric study or anything else to validate any of the assumptions. This SVI cannot be considered a percentage unless there is backup data validating this. This has been done also in fatality modelling where the models that are presented in any publication are previously calibrated and validated with data from historic events. Moreover, considering previous events reporting post-loss amplification that include costs from the response and recovery stages in some disasters, it has been shown that numbers over 30-40% are almost non-existent (What is demand surge? Olsen and Porter 2010), while this study mention cases with increases of up to 60%. There may be a problem with the explanation of the methodology, but as it is right now it is very difficult understand how it can relate to economic losses, especially direct physical losses after an event.

(These limitations are also afterwards mentioned by the authors in the discussion, but it is a MAJOR limitation of the inclusion of the SVI methodology in the results in this study, as there is no validation or calibration of any kind for the methodology)

Table 9 numbers are misleading as a direct non-weighted average of the 18 scenarios is not probabilistically and statistically sound. It should consider the contribution of each event, otherwise the less probable events are counted in the same way as the more probable ones. In this way, as when computing AAL from a probabilistic analysis, the contributions should consider the probability of occurrence of each scenario. After saying this, it is advised not to present this table and instead present one with the analysis of each scenario done separately as in a deterministic approach, unless it is possible to demonstrate that the 18 scenarios included account for the 100% of the 475 years return period loss and a weighted average is calculated based on the contribution of each.

**Section 4.4:**

As stated previously, to present these analyses, it would be important to show the contribution of the 18 scenarios to the total hazard in the region (based on the disaggregation results). If not Figure 13 is misleading, considering that it says: "Within the municipalities, the mean percentage of losses is presented with respect to **the total expected losses in the region**".

"The economic losses experienced by a province due to an earthquake depends on the event's epicenter as it is depicted in Figure 12." Figure 12 does not show in any way anything regarding the epicenter. It may not be only the epicenter but also the Mw for each event the cause of the differences, so this statement is not provable from the Figure. Delete it.

**Discussion:**

Just until this section this is stated: "The simulations of eighteen seismic scenarios with a return period of 475 years show that half of the building stock will experience some degree of damage". How was this '475 years' return period calculated? Even when the disaggregation was done for the

'475 years' return period, how can it be confirmed that the 18 scenarios add up to the 100% contribution for the hazard for this return period? Either way this statement should be included in some way in previous sections and not only until the discussion.

**Effects of SV:**

The main criticism for this approach is also stated by the authors in this sentence: "First, as not all social aspects exert equal effect after an earthquake, it is necessary to develop a weighted approach to best estimate a more realistic SVI for earthquake events. A second improvement required is to devise a better way of estimating the economic impact of social vulnerability. One potential approach is to generate a database of past earthquakes with different consequences that include the economic costs." These are not needed future improvements but major limitations of the proposed approach. In this kind of analyses, as when performing a linear regression, it is important to avoid counting double and establish the significance of each variable within the analysis, if not this could be overestimating the vulnerability and losses in the region considerably. Also, using the min and max approach is very subjective as many variables as unemployment and poverty are tempered with by local organisms. This kind of indicators are good to compare and prioritize actions within regions but cannot be used in the way they are presented in this study to increase direct physical losses.

Minor modifications:

Line 118: Typo: This **GROWTH** counts for 64% of the total population of the region

Line 248: Incomplete sentence: "The number of masonry buildings represents the 88.61% of the total buildings in Sabana Centro, whereas those of concrete and wood represent 3.16% and 8.24%, respectively, with."

Line 261: Repeated to "In the absence of specific curves locally developed for the Sabana Centro province, fragility curves were selected to **to** represent these structures"

---

## Author Comment (AC1)

This article presents seismic risk results from 18 earthquake scenarios in the Sabana Centro province, an intermediate hazard zone in Colombia. The 18 scenario events were chosen based on hazard disaggregation results on a site within the region of interest, for a 475 years return period hazard level. The epicenters of the scenario events were located within the region. The exposure model was gathered from previous studies, complemented with census data and remote surveys. Finally, the set of fragility curves for the considered typologies was gathered from previous studies. Results show that, on average, the occurrence of one of the scenario earthquakes might result in about $800 million USD of economic losses (about $1000 million USD when adjusted by social vulnerability) and about 20% of all the buildings collapsing.

The study is well performed and written. In terms of scientific merit, although the article does not provide new methods for seismic risk assessment, it provides novel results on the seismic risk faced by the Sabana Centro province. I have some comments that may help to improve the quality of the article (I have marked the most important issues that I comment).

In particular, compared to the damage produced by previous earthquakes around the World, the numbers presented in this article seem to be too high, especially considering that: (1) these are mean values, not low-probability values (thus they are not even the "worst-case scenario"); and (2) the authors state that the risk results should be considered as a lower bound. Therefore, I strongly suggest a careful revision and discussion of the results.

**Dear Dr. Heresi**

You kindly spent time delving into our manuscript, and we are grateful. Thank you for your appreciation of the study's performance and writing style, and acknowledging the novelty of the results on the seismic risk assessment in the Sabana Centro province since the study performed by the Japan International Cooperation Agency (JICA) in 2001 has been neither discussed nor updated. Thank you for your comments. We will do our best to provide a complete answer to all of them.

*Specific Comments (Individual scientific questions/issues)*

1. Line 130: The authors state that the Bucaramanga's seismic nest earthquakes occur at depths between 140 and 200km. However, in Figure 3 it is possible to observe several events at depths between 70 and 150km. I suggest the authors comment something about this inconsistency.

R/: Thank you for your comment. The statement on line 130 corresponds to the general definition of the Bucaramanga seismic nest. In Figure 3 the authors show the complete catalog without source distribution. That is, the events that are the origin of the Bucaramanga seismic nest are considered as those cortical events, which can occur over the seismic nest. To avoid this misunderstanding, the following sentence will be included in the revised version of the article to clarify the information presented in the figure:

"Figure 3 shows the events of the complete catalog, considering those of seismic nest origin as well as those events of cortical environment."

2. Line 144: The description of Figure 3 does not match what is observed in the figure. The authors state that Figure 3 shows: (1) close (distance < 75km) and shallow (depth < 70km) events; and (2) far (150km < distance < 200km) and deep (depth > 70km) events with magnitudes greater than 6.5. However, the figure presents many other events. For example, there are shallow events at distances larger than 75km with magnitudes lower than 6.5, deep events with magnitudes lower than 6.5, etc. The description and the figure should be consistent.

R/: Thank you for your suggestion. To be consistent between the figure and the description we have improved the discussion of the figure by describing the events according to the distance from the disaggregation, as well as other information like the magnitude and depth.

3. Line 155: The authors state that the population centroid is located within Tenjo. Moreover, they use this municipality throughout the article as a reference (e.g., Figures 2 and 3). However, given the map provided in Figure 6c and the population of each municipality provided in Table 1, it seems like the centroid should be located somewhere between Zipaquirá and Chía (probably within Cajica), which are the two municipalities with the largest populations.

R/: We agree with you. The interpretation of the sentence is that the disaggregation was done for the population center of the entire study region; however, what was meant is that this procedure was done for the population center of Tenjo. This municipality was taken as a reference point at the authors' discretion. We rewrote the sentence so that it is clearer.

4. Figure 4: What is the distance type of the disaggregation? Rupture distance? Epicentral distance? Joyner-Boore distance? Given the depth of the deep sources, the difference between the distance types might be important.

R/: Thank you for your comment. For disaggregation, the authors used a distance to the projection of the rupture surface corresponding to the Joyner-Boore distance. To clarify both comments, we will include the following sentence in the revised manuscript:

The disaggregation was developed for a point within the region of analysis, which corresponds to the population centroid of the municipality of Tenjo (longitude: -74.144, latitude: 4.872), considering the Joyner-Boore distance to the projection of the rupture surface.

5. Table 2: The authors provide a list of the selected scenarios. A map showing these events might be useful for visualizing the epicenters with respect to the municipalities.

R/: We appreciate your recommendation. We will add a new figure with this information in the manuscript.

6. Table 3 and its description: Although the logic tree was proposed by Arcila et al. (2020), I suggest the authors provide a justification for the different weights to the Cauzzi et al. (2014) and Abrahamson et al. (2014) ground-motion models.

R/: The logic tree proposed by Arcila et al. (2020) for shallow crustal regions has three GMPEs: Cauzzi et al. (2014), Abrahamson et al. (2014) and Idriss. (2014). The corresponding weights are 0.39, 0.211 and 0.399, respectively. The Sabana Centro region has a significant number of soft soils for which the Vs30 is lower than 450m/s. The Idriss (2014) GMPE is not defined for these types of soils. Consequently, for this study we decided to exclude this GMPE and use a logic tree that included the Cauzzi et al. (2014) and the Abrahamson et al. (2014) GMPEs, which do account for soft soils.

To update the weights of the Abrahamson and Cauzzi et al. (2014) GMPE in the new logic tree, we adjusted them distributing the weight of the Idriss (2014) GMPE (0.399) proportionally to their weights in the initial logic tree. We did so because this allowed us to keep the same level of relative weights between the two GMPEs used for our study. This led to the following calculations:

$$Cauzzi_{updated} = \mathbf{0.39} + \frac{0.399}{0.39 + 0.211} = \mathbf{\color{red}0.65}$$

$$Abrahamson_{updated} = \mathbf{0.211} + \frac{0.399}{0.39 + 0.211} = \mathbf{\color{red}0.35}$$

7. **\*Important comment\*** In Line 105, the authors state that "The majority of the building stock of the region is comprised of one- and two-story houses." However, as shown in Table 4, the considered building typologies include buildings with either 1 or 4 stories. The question is then, how are two-story houses classified into this system? This question is especially important because it has been previously demonstrated that one- and two-story houses present a significantly different seismic behavior and therefore levels of damage and losses (see, for instance, Heresi and Miranda 2022). In particular, classifying two-story houses as one-story structures may result in a significant underestimation of the seismic risk of these two-story structures.

Heresi, P., & Miranda, E. (2022). Evaluation of relative seismic performance between one- and two-story houses. Journal of Earthquake Engineering, 26(2), 857-886.

R/: Thank you for this important comment. Other reviewers also raised their attention to the need of updating the exposure model, further discretizing between building heights. We agree specially that the difference between one- and two-story houses is significant. Based on this comment and the other reviewers' suggestions, we decided to update the exposure model using information of the 2018 national census, with further differentiation between building heights. To discretize the building height, we used information collected using remote and field surveys of 9000 houses in the Sabana Centro region, conducted by students of Universidad de La Sabana. We found that for houses between 1 and 3 stories, 34% are one-story houses, 48% are two-story houses and 18% are three-story houses. Accordingly, we updated our exposure model.

In addition, we decided to use the fragility functions calculated by Martins and Silva. (2021), which all use the same modeling approach, and they account for the differences in height.

Furthermore, we also updated the discussion section, including the suggested reference and the above-mentioned facts.

8. Table 5 presents the main parameters of the considered fragility curves. As stated by the authors, these fragility curves were selected from different studies after a thorough literature review. Although this is perfectly fine, it has an important drawback that should be commented on: the final set of fragility curves comprise curves developed with very different methods (e.g., analytical vs empirical) which have very different reliabilities (e.g., generally speaking, empirical fragility curves developed after earthquakes have higher uncertainties both in the probability of damage and in the ground-shaking intensity). The authors are encouraged to discuss about the limitations and reliability of the considered fragility curves, taking into account the methods, the data, and the assumptions used to develop them. They address some of these issues in the Caveats and Limitations section, specifically the issue of fragility curves not being developed directly for Colombian structures and not having a uniform description of the damage states, but there are other issues that are missing in this section, as those previously stated in my comment.

R/: Thank you for this comment. Other reviewers also raised their attention to this fact. For consistency in the revised version of the manuscript, we decided to use only analytical fragility functions. Most of them were those calculated by Martins and Silva (2021) and Villar-Vega et al. (2017). These functions are analytical and all of them use the same modelling approach. We used these functions for several buildings, with some exceptions such as the thin reinforced concrete buildings, for which we kept the fragility functions developed by Arroyo et al. (2021), because these are also analytical and use a more accurate model for these types of structures. We kept the comments about the issue that the fragility functions by Martins and Silva. (2021) were not developed accounting for the particularities of Colombian construction.

9. **Important comment** Results show that a Mw5.95 event at Chía is expected to cause the collapse of more than 17% of the buildings in the region, and some level of damage in about half of the building portfolio. In particular, 6722 out of 14959 (about 45%) of houses made out of non-ductile unreinforced masonry with adobe block walls (1-story) are expected to collapse, according to the authors. Moreover, in Chía, more than 44% of the buildings are expected to collapse due to this Mw5.95 scenario. These numbers seem incredibly high for a Mw5.95 event at a first glance (even more when the authors state, in Line 441, that these estimates should be considered as a lower bound). Note that these are mean (i.e., expected) values, not low-probability values that might represent a somewhat "worst-case scenario" (or, in other words, somehow answer the question "how big may be the consequence if this earthquake occurs tomorrow?"). To put these numbers in perspective, we can compare them with the damage produced by the 2010 Haiti earthquake, Mw7.0:

- According to DesRoches et al. (2011), the 2010 Haiti earthquake damaged nearly half of the structures in the epicentral region.
- Eberhard et al. (2013) performed two field surveys of: (1) 107 structures in Port-au-Prince, where 30 (28%) of them collapsed and other 35 (33%) had enough damage to require repairs; and (2) 52 structures in Léogâne (closest population center to the

epicenter), where 32 (62%) of them collapsed and other 16 (31%) had enough damage to require repairs.
- Rathje et al. (2011) performed a field survey of over 400 structures in Port-au-Prince. Of the 414 surveyed structures, 157 (38%) had significant damage (i.e., collapse or very heavy damage, EMS Grade 4).

Considering that the Haiti earthquake was not only 32 times larger in terms of magnitude, but also affected a more socially vulnerable country, it is expected that a Mw 5.95 event in the region of interest would result in considerably less damage and losses, especially if we talk about mean values.

In terms of losses, in Figure 12 we can observe that some of the earthquake scenarios have a 20% probability of producing more than 50% of the total replacement cost as economic losses (about 40% of the GDP of the region!). Considering that these curves were computed neglecting the spatial correlation of ground motion intensities (comment about this below), this probability for such a high loss is extremely large. For perspective, the 2010 Chile earthquake, Mw8.8, produced an economic loss of about 14% of the GDP of the country at the moment of the event.

The previous remarks highlight the importance of comparing risk results from scenario events with previous events to put the numbers in perspective. I suggest the authors include comparisons like the ones proposed above, but also include other events, such as, for example, the 2020 Puerto Rico earthquake, Mw6.4. Moreover, in the Introduction, the authors mention two historical earthquakes that affected the region of interest, which may also be used to evaluate the reliability of the resulting damage produced by the considered scenario earthquakes. These comparisons would further support the risk results of the article.

DesRoches, R., Comerio, M., Eberhard, M., Mooney, W., & Rix, G. J. (2011). Overview of the 2010 Haiti earthquake. *Earthquake Spectra*, *27*(S1), S1-S21.

Eberhard, M. O., Baldridge, S., Marshall, J., Mooney, W., & Rix, G. J. (2010). The Mw 7.0 Haiti earthquake of January 12, 2010: USGS/EERI advance reconnaissance team report. *US Geological Survey Open-File Report*, *1048*(2013), 64.

Rathje, E. M., Bachhuber, J., Dulberg, R., Cox, B. R., Kottke, A., Wood, C., ... & Rix, G. (2011). Damage patterns in Port-au-Prince during the 2010 Haiti earthquake. *Earthquake Spectra*, *27*(S1), S117-S136.

R/: We highly appreciate this comment and acknowledge the importance of comparing with previous events to put the results in a broader perspective. In this regard, we would like to frame the discussion in terms of the 1999 Armenia Earthquake in Colombia. This event had a Mw 6.1 magnitude at 15 km depth. Most of the building stock was comprised of URM buildings, built prior to the 1998 seismic design code of Colombia and like the buildings in Sabana Centro. This earthquake costed 1.6% of the national GDP (roughly five times higher than the earthquake considered in this paper). In terms of damage, the records indicate that 17551 were destroyed, 18421 had severe damage and 43474 had moderate damage. Another earthquake in Colombia was the Mw 5.5 in Popayán in 1983, which occurred at an estimated

depth of 12 to 15 km. According to Colombian records, in this earthquake, 12% of buildings suffered complete damage and 34% experienced severe damage.

These two events put the numbers in perspective and support that the results of this paper are reasonable. In the discussion in the revised manuscript, we will include these facts and compare our results with those of these earthquakes.

Regarding the reviewer comment about the lower bound, the authors has this concern because the field surveys have shown that a significant number of buildings in Sabana Centro are informally constructed, rising the possibility that none of the existing fragility functions can represent this type of construction properly.

10. Table 10 presents the resulting SVI for the 11 municipalities of the region. Although the authors previously explain the variables involved in this index (Table 6), I have two comments about this:

I suggest the authors provide more detailed information about how the index of each category is obtained. This explanation would improve the reproducibility of the reported results.

R/: Thank you very much for this suggestion; the selection of the indicators that are part of the composite indicators: population, economy, infrastructure, education, health, and the variables considered for the indicator COVID-19 was already explained in section 3.4. The composite indicators, single indicators and variables considered initially to estimate the SV were listed in Table 6, in the submitted version. However, to provide more detailed information about how the SV index is constructed, we elaborate more on the explanation of the methodology. We did this in several ways: first, we explain in further detail each composite indicator, second, we checked the multicollinearity of the indicators based on the variance inflation factor (VIF) (see Table 6), and we excluded those that were potentially correlated with others and those that did not add significant information.

11. There are many variables used for the SVI that are strongly correlated. For example, in the "Population" category, there are 7 variables, where, for instance, "Female population" and "Total population" are expected to be strongly correlated, unless the percentage of women varies significantly from one municipality to another for some reason. As the authors did not provide too much detail on how the index is computed, I'm not sure if they tested for collinearity between these variables, for example. We can even expect some correlation between different categories. For instance, municipalities with a high index in Economy will probably have also a high index in Infrastructure. These correlations might result in biased SVI's when all the variables are considered.

R/: Thank you very much for this comment. The reviewer is right indicators such as female population and total population are strongly correlated (1.000**). Then, to avoid problems interpreting the model and overfitting, we checked the multicollinearity by looking at the variance inflation factor (VIF) of each variable and indicator. The VIF was identified in a linear regression that included collinearity diagnostics produced in SPSS (Field, 2005).

12. **\*Important comment\*** As one of the limitations, the authors state that they did not consider the spatial cross-correlation when modelling the ground motion fields. However, they do not justify this arbitrary exclusion. For example, the OQ-Engine has models of spatial correlation already implemented, and therefore I do not see a good reason for neglecting it. As the authors correctly state, the inclusion of a spatial correlation model would increase the dispersion of the curves presented in Figure 12, making them more "realistic". Thus, I suggest either including a spatial correlation model, or giving a strong justification for its arbitrary exclusion.

R/: We appreciate the reviewer comment, and we will update the discussion section based on the following arguments:

The reviewer expresses some concerns about having disregarded a spatial correlation model to model the ground motion fields. It is worth noting that we always consider the ground motion variability through uncorrelated random fields (that allowed us to create Figure 12). It is also worth noting that the sentence in line 451 did not refer to simple spatial correlation models, but rather to spatial inter-period cross-correlation models (IPCCM), i.e. when several intensity measures (IM) are simultaneously required by their set of fragility functions to calculate the physical vulnerability of building stocks to earthquakes. Related to this, it is important to note that the current OpenQuake engine only provides the option to simulate spatially correlated ground motion fields (e.g. Jayaram and Baker, 2009), but it does not provide spatially cross-correlated random fields.

Moreover, it is important to highlight the relation between the spatial extent and density of the building stocks with respect to the decision of including or not IPCCM. For instance, the study by Michel et al. (2017) found that for building portfolios that are spaced a few kilometers apart, the influence of cross-correlation in risk assessment is very small compared to the one imposed by ground motion variability itself. This feature is similar to the one we encounter in the Sabana Centro region where the main urban areas (cascos urbanos) between neighboring municipalities are separated by several kilometers. Conversely, other studies have found that the role of including either simpler spatial correlation models (e.g. Bazzurro and Luco, 2005) or IPCCM (Gomez Zapata et al, 2022a,b) for a dense and spatially aggregated building portfolio is comparatively more relevant, which is not to be the case of our study area. In fact, the aforementioned cited study remarks on this issue by stating that, since the spatial correlation of ground motion IMs decreases rapidly with distance (e.g. Schiappapietra and Douglas, 2020), its effect on loss-estimations is maximized when it is applied to a dense exposure model (i.e. with aggregation areas (~1 x 1km grid) significantly smaller than the correlation distance of the ground motions (~20 km) because buildings within a grid cell are treated as if the inter-station distance was zero. Since our exposure model in Sabana Centro is composed of only 11 geocells where the buildings are therein spatially aggregated and with centroid-to centroid distances of the same order as the ground motion correlation lengths, we can expect that the relevance of including a spatial correlation model would not be high. Of course, this feature is inherent to the decision of the aggregation areas (11 municipalities).

Aligned with the former, and as described by Stafford. (2012) and by Gomez Zapata. (2021), one can suspect that when the dimension of the geo-cells in the exposure model is larger than

a typical seismic ground motion correlation length, an artificial bias in the ground motion correlation has to be expected which may be the case in our study. Therefore, more meaningful future studies with higher resolution exposure models are anyway required but are without our scope. These possible future improvements along with local fragility models (perhaps with more IM) will certainly require the incorporation of spatially correlated or cross-correlated models for which we can then confirm the important relationship and similarity between the correlation of ground motions and the damage correlation of exposed structures as comprehensively presented by Heresi and Miranda. (2022).

*Technical corrections*

13. Line 43: Change "7248 injured" for "7248 injured people" or "7248 injuries".

14. There is an inconsistency in the use of thousand separators. For example, in Line 45 the authors state "… and 35000 buildings that collapsed…", but then in Line 86, they write "resulted in 200,000 deaths". In Table 1, the authors use thousand separators again.

15. Line 118: Review the word "gro".

16. Line 158: The authors use the Quetame earthquake for defining the rupture geometry of the scenario events. I suggest adding an annotation in Figure 3, showing which one is the Quetame earthquake, for those of us who are not familiar with the historic seismicity of Colombia.

17. Line 249: There is an incomplete phrase.

R/: Thank you for taking the time to make technical corrections, we have addressed all of them in the manuscript.

**References**

Abrahamson, N. A., Silva, W. J., and Kamai, R.: Summary of the ASK14 Ground Motion Relation for Active Crustal Regions, Earthquake Spectra, 30, 1025–1055, https://doi.org/10.1193/070913eqs198m, 2014

Arroyo, O., Feliciano, D., Carrillo, J., and Hube, M. A.: Seismic performance of mid-rise thin concrete wall buildings lightly reinforced with deformed bars or welded wire mesh, Engineering Structures, 241, https://doi.org/10.1016/j.engstruct.2021.112455, 2021

Arcila, M. M., García, J., Montenjo, J. S., Eraso, J. F., Valcárcel, J. A., Mora, M. G., Viganò, D., Pagani, M., and Díaz, F. J.: Modelo nacional de amenaza sísmica para Colombia, libros del Servicio Geológico Colombiano, https://doi.org/10.32685/9789585279469, 2020

Bazzurro P, Luco N: Accounting for uncertainty and correlation in earthquake loss estimation. In: Proceedings of the ninth international conference on safety and reliability of engineering systems and structures. presented at the ICOSSAR, Rome, Italy, 2005

Cauzzi, C., Faccioli, E., Vanini, M., and Bianchini, A.: Updated predictive equations for broadband (0.01–10 s) horizontal response spectra and peak ground motions, based on a global dataset of digital acceleration records, Bulletin of Earthquake Engineering 13, 1587–1612, https://doi.org/10.1007/s10518-014-9685-y, 2014.

Gomez-Zapata, J. C., Brinckmann, N., Harig, S., Zafrir, R., Pittore, M., Cotton, F., and Babeyko, A.: Variable-resolution building exposure modelling for earthquake and tsunami scenario-based risk assessment: an application case in Lima, Peru, Nat. Hazards Earth Syst. Sci., 21, 3599–3628, https://doi.org/10.5194/nhess-21-3599-2021, 2022.

Gómez Zapata, J.C., Pittore, M., Cotton, F. et al.: Epistemic uncertainty of probabilistic building exposure compositions in scenario-based earthquake loss models. Bull Earthquake Eng 20, 2401–2438, https://doi.org/10.1007/s10518-021-01312-9, 2022.

Gómez Zapata JC, Zafrir R, Pittore M, Merino Y. Towards a Sensitivity Analysis in Seismic Risk with Probabilistic Building Exposure Models: An Application in Valparaíso, Chile Using Ancillary Open-Source Data and Parametric Ground Motions. ISPRS International Journal of Geo-Information, 11(2):113, https://doi.org/10.3390/ijgi11020113, 2022.

Heresi, P., Miranda, E.: Structure-to-structure damage correlation for scenario-based regional seismic risk assessment. Structural Safety 95, 102155. https://doi.org/10.1016/j.strusafe.2021.102155, 2021.

Idriss, I. M.: An NGA-West2 empirical model for estimating the horizontal spectral values generated by shallow crustal earthquakes, Earthquake Spectra, 30, 1155–1177, https://doi.org/10.1193/070613eqs195m, 2014.

Jayaram, N.; Baker, J.W.: Correlation model for spatially distributed ground-motion intensities. Earthq. Eng. Struct. Dyn, 38, 1687–1708, 2009.

Martins, L. and Silva, V.: Development of a fragility and vulnerability model for global seismic risk analyses, Bulletin of Earthquake Engineering, 19, 6719–6745, https://doi.org/10.1007/s10518-020-00885-1, 2021.

Michel, C., Hannewald, P., Lestuzzi, P., Fäh, D., and Husen, S.: Probabilistic mechanics-based loss scenarios for school buildings in Basel (Switzerland), Bulletin of Earthquake Engineering, 15, 1471–1496, https://doi.org/10.1007/s10518-016-0025-2, 2017.

Schiappapietra, E., & Douglas, J.: Modelling the spatial correlation of earthquake ground motion: Insights from the literature, data from the 2016–2017 Central Italy earthquake sequence and ground-motion simulations. Earth-science reviews, 203, 103139, 2020.

Villar-Vega, M., Silva, V., Crowley, H., Yepes, C., Tarque, N., Acevedo, A. B., Hube, M. A., Gustavo, C. D., and Santa María, H.: Development of a fragility model for the residential building stock in South America, Earthquake Spectra, 33, 581–604, https://doi.org/10.1193/010716eqs005m, 2017.

---

## Author Comment (AC2)

The article addresses a relevant issue for a seismic country as Colombia; it gives relevant information about the seismic risk of the Sabana Centro providence. Nonetheless, there is no novelty on the article and key information is not given in the paper. Results from the selected scenarios indicate consequences of concern not well supported. My main concern regards the selected fragility curves: I find it complex to perform a risk analysis using fragility functions developed by different methodologies and as stated by the authors, with different limit states definitions. I believe this is one issue that requires additional explanation. For example, comparison between fragility functions is not presented. Do the set of curves behave as expected? Information given in Table 5 is not enough. The article should present figures that allows for a visually appreciation of the curves. A brief explanation of the methodologies uses for the curves' development should also be included, as well as an opinion about how reliable the curves are. How can the authors explain that the number of collapse buildings is almost three times the number of buildings with extensive damage? Furthermore, the number of collapse buildings exceeds the number of buildings in any other damage state.

**Dear Dr. Ana Acevedo**

You kindly spent time delving into our manuscript, and we are grateful. Thank you for your appreciation of the study's performance and writing style and acknowledging the relevant information for seismic risk in the region. Thank you for all your comments. We will do our best to provide a complete answer to all of them.

1. My main concern regards the selected fragility curves: I find it complex to perform a risk analysis using fragility functions developed by different methodologies and as stated by the authors, with different limit states definitions. I believe this is one issue that requires additional explanation. For example, comparison between fragility functions is not presented. Do the set of curves behave as expected? Information given in Table 5 is not enough. The article should present figures that allows for a visually appreciation of the curves. A brief explanation of the methodologies uses for the curves' development should also be included, as well as an opinion about how reliable the curves are.

R/: Thank you for your main comment. We agree with your point. Other reviewers also raised their attention to this fact. Therefore, for consistency in the revised version of the manuscript, we decided to use only analytical fragility functions. Most of them were those calculated by Martins and Silva. (2021) and Villar-Vega et al. (2017). These functions are analytical and all of them use the same modelling approach. We used these functions for several buildings, with some exceptions such as the thin reinforced concrete buildings, for which we kept the fragility functions developed by Arroyo et al. (2021), because these are also analytical and use a more accurate mathematical model for these types of structures which allows capturing local behavior of the walls. We kept the comments about the issue that the fragility functions by Martins and Silva. (2021) were not developed accounting for the particularities of Colombian construction.

2. How can the authors explain that the number of collapse buildings is almost three times the number of buildings with extensive damage? Furthermore, the number of collapse buildings exceeds the number of buildings in any other damage state.

R/: We will update the discussion in the revised manuscript, with the results obtained using a new exposure model and the set of fragility function described in the answer of comment # 1.

Regarding the question of about the number of collapses in the initial manuscript, in the scenarios developed using the initial selected set of fragility functions, the intensity measure for several earthquakes was in the range where collapse had the highest probability.

**Additional comments:**

3. Why do all the scenarios are crustal shallow events? In the article it is mentioned that for SA (1.0s) there is an important contribution of subduction events. As the number of scenarios is important (18) some of them should be subduction events.

R/: We looked at the contributions by tectonic environments to the seismic hazard at the PGA Sa (0.3) and Sa(1.0s) In figure 1 it is possible to observe that the seismic hazard is controlled by active shallow crust events, being its contribution greater than the 75% in all the municipalitie1s. Moreover, at the PGA it is greater than 90% for municipalities like Chía and Cajicá, where a significant number of one and two-story building concentrate. The contribution of the Bucaramanga seismic nest is more relevant in the municipalities at the north of the region in Cogua, Nemocón and Zipaquirá. In such cases, the contribution of the Bucaramanga seismic nest is around 25%. Based on this, we decided to keep the eighteen crustal scenarios and run one from the Bucaramanga seismic nest. We would like to point that the damage results for this scenario are low (collapses range between 0.1% and 0.2%).

[Figure]

[Figure]

Figure 1. Contribution of tectonic environments to the seismic hazard of the Sabana Centro municipalities

4. The authors mentioned the use of population census data to infer the number of buildings added to the original exposure model of SARA. It is not clear why the authors did not use the census data to directly obtain the number of buildings. The 2018 Census provides relevant information that can be used to have a more precise number of buildings.

R/: We appreciate a lot this comment. We used population as a simple and practical process to directly obtain the number of buildings. The census provided information to obtain the number of dwellings. We compared our results with those reported by the 2018 census and the difference is 28%. This difference is mainly because the census considered dwellings

whose wall material is poured concrete and we did not include it since it was not possible to assign a percentage of construction.

Based on your comments and those of the other reviewers, we decided to update the exposure model and use the information of the dwellings in the 2018 census instead of the one based on population. Nonetheless, we think if worth highlighting that in the absence of census information, inferring based on population can provide a result with a moderate degree of approximation.

5. How was the building typology assigned to the added buildings to the original exposure model?

R/: Initially we updated the number of buildings keeping the same typologies and the same relative percentages between building typologies. Based on your comments and those of the other reviewers to have a more reliable estimation we decided to update the exposure model following the methodology used in Yepes-Estrada et al. (2017), where taxonomies are assigned according to the materials of the walls and roofs of the dwellings. The information used for this process was obtained directly from the census data.

6. Which replacement cost did the authors use? The authors only mention that the cost is assigned according to the socio-economic levels, but it is not clear which cost was used and how was it computed: cost per area? Cost per building? It is suggested to include the replacement cost in Colombian pesos as the exchange currency fluctuates.

R/: Thank you for this comment, we have realized that we did not mention it in the manuscript. We have used the cost per building. We will include a sentence mention that is the cost per building, and we will include the replacement cost in Colombian pesos.

7. It is not clear how the information of the base exposure model (SARA) was complemented with the information of the 6249 surveys. Furthermore, all these buildings belong to the same municipality. A description of the buildings characteristics of each municipality should be included.

R/: We used the information of the surveys to update the percentage of the taxonomies only in the municipality of Chía because we had remote field surveys of this municipality that were conducted by students of Universidad de La Sabana. We will include a sentence clarifying this procedure.

8. The authors mention that 8.24% of the stock are wood buildings. How does this information compare to the Census data? (The Census provides information about building's wall material). In addition, the authors assigned a fragility function for wood buildings developed for Chilean buildings. Although the reference of the fragility functions used for wood has not yet been published, it is not clear that Colombian wood buildings have the same seismic behavior as Chilean wood buildings. A support for the use of Chilean wood fragility functions is needed.

R/: In the previous exposure model we only compared the total number of dwellings. For the updated exposure model we used the census data which includes walls and roof materials. This allowed us a better estimation of the total number of buildings for each taxonomy.

In the case of wood houses, we decided to use the fragility curves calculated for Chilean buildings. These curves were developed by the authors using detailed drawings of Chilean houses, which are similar in configuration to the wood houses found in Sabana Centro. Regarding the method, we used a single DOF oscillator. In general, the wood houses found in the field surveys in the region have good quality materials and adequate construction, thus we consider that the fragility curves used are suitable.

9. It is not clear why the authors use only two building heights: 1 and 4. Does the exposure model only comprise building with 1 and 4 stories? Or does the exposure model have buildings of several number of stories, but the authors decided to group them in just to building heights? Whatever the option, for a region where most of the buildings are low-rise buildings (as stated in the paper) a differentiation of number of stories is very important.

R/: Thank you for this important comment. Other reviewers also raised their attention to the need of updating the exposure model, further discretizing between building heights. We agree specially that the difference between one- and two-story houses is significant. Based on this comment and the other reviewers' suggestions, we decided to update the exposure model using information of the 2018 national census, with further differentiation between building heights. To discretize the building height, we used information collected using remote and field surveys of 9000 houses in the Sabana Centro region, conducted by students of Universidad de La Sabana. We found that for houses between 1 and 3 stories, 34% are one-story houses, 48% are two-story houses and 18% are three-story houses. Accordingly, we updated our exposure model.

In addition, we decided to use the fragility functions calculated by Martins, which all use the same modeling approach, and they account for the differences in height. Furthermore, we also updated the discussion section, including the suggested reference and the above-mentioned facts.

10. Results should include the uncertainty as 1000 ground motion fields were generated and two GMPEs were used.

R/: Thank you for this recommendation. We will include statistics in the updated version of the manuscript.

11. The taxonomy MCF/DNO/H:1 is not correct as it is missing the lateral load resisting system.

R/: Thank you for noting this omission. We fixed this in the updated manuscript.

12. The taxonomy CR/LFINF/DUM/H:4 is used for buildings constructed using thin RC walls. This is not the original definition in the GEM taxonomy. It is suggested to use a different taxonomy.

R/: Thank you for this observation. We will use an appropriate taxonomy consistent with the GEM methodology.

13. The taxonomy W/H1 is missing the information about the lateral load resisting system and the ductility level.

R/: Thank you for noting this omission. We fixed this in the updated manuscript.

14. It is not clear why the authors present mean values for the 18 seismic events. As each scenario has a different epicenter and different consequences mean values are not representative (results for each scenario should be presented by themselves). See Table 19 and Figures 10 and 13.

R/: We appreciate this comment. We agree that 18 scenarios are not enough to represent 100% of the hazard. We should though mention that the selected scenarios do represent a significant percent of the contribution to the seismic hazard, as the examination of figure 4 of the manuscript can reveal, particularly for the PGA. We added this comment in the corresponding section of the revised manuscript.

In the light of the comment, we decided to remove table 9 and instead include a figure that shows the statistical variability of each damage state for the eighteen scenarios. We also decided to remove Figure 13. We kept figure 12 but we improve our analysis of this figure. The main point here is that Sabana Centro has an uneven distribution of the building stock, therefore, similar earthquakes with the same magnitude and similar depth such as the Mw 6.25 Sopó, Mw 6.25 Tabio and Mw 6.25 Nemocón have different consequences in terms of economic losses.

15. The sentence of line 410 "One out of four buildings will experience extensive damage or collapse" is a strong conclusion that requires a big certainty to be written. I suggest the authors to revise the fragility curves of the masonry buildings (as most of the buildings are of this typology) and to compare the ground motion fields with the building damages to be sure that results are correct. Furthermore, as all the buildings form this typology are one-story buildings results should not be as bad as shown in the article.

R/: Based on this comment and the other reviewers concerns about not discretizing between the height for buildings in the range of one to three stories, we decided to update the exposure model using information of the 2018 national census, with further differentiation between building heights. In addition, we decided to use the fragility functions calculated by Martins and Silva. (2021), which all use the same modeling approach, and they account for the differences in height. Furthermore, we also updated the discussion section, including the suggested recommendation and the above-mentioned facts.

16. Line 440. The authors mention "the damage and losses estimates presented in this study should be considered as lower bound". See the previous comments.

R/: The statement refers to the fact that existing fragility functions do not account for the informal construction practices which are common in the Sabana Centro region.

17. Figure 4. Add a color scale. It is difficult to read the percentage associate to each bin.

R/: We added a color scale to the figure

18. Figure 9. It is suggested to include the earthquake epicenter as well as a figure with the ground motion field generated by the event.

R/: We added a figure with the epicenter and the ground motion field.

19. Figure 11. Expected losses including SVI should be greater than the expected losses without SVI. This is not shown at Chía and Sopó. For the ease of understanding it is suggested to use the same color scale in both maps of the figure.

R/: We updated the figure with the same color scales.

20. Figure 12. It is suggested to include the uncertainty in the figure.

R/: We did not include uncertainty in this figure; however, we did include a new figure in the paper to represent the variability in the damage and losses.

21. Table 2. Add the distance for the epicenter to the study area. Complement the information with a figure in which the epicenters are shown. As the events have an associated municipality, is the epicenter located at each municipality? How feasible is this? Results indicate important consequences that can be misinterpreted if the article does not mention the possibility that such events occur with epicenters in each municipality.

R/: Based on your suggestion and other reviewers' comments, we replaced table 2 with a figure that includes the location of the epicenters. We also clarified the statement about the epicenter location.

22. Line 179. How does the "significant number of low-rise stiff buildings" relate to the selected crustal events?

R/: Thank you for this comment. We realized that the sentences is clearer if we limit the discussion to the dissagregation (Figure 4 in the paper), which shows that the highest contribution to the PGA comes from crustal events with magnitudes that range between 5.0 to 6.5 and distances smaller than 30km. We updated the paper removing the sentence quoted by the reviewer.

**References**

Arroyo, O., Feliciano, D., Carrillo, J., and Hube, M. A.: Seismic performance of mid-rise thin concrete wall buildings lightly reinforced with deformed bars or welded wire mesh, Engineering Structures, 241, https://doi.org/10.1016/j.engstruct.2021.112455, 2021.

Martins, L. and Silva, V.: Development of a fragility and vulnerability model for global seismic risk analyses, Bulletin of Earthquake Engineering, 19, 6719–6745, https://doi.org/10.1007/s10518-020-00885-1, 2021.

Villar-Vega, M., Silva, V., Crowley, H., Yepes, C., Tarque, N., Acevedo, A. B., Hube, M. A., Gustavo, C. D., and Santa María, H.: Development of a fragility model for the residential building stock in South America, Earthquake Spectra, 33, 581–604, https://doi.org/10.1193/010716eqs005m, 2017.

Yepes-Estrada, C, Silva, V., Valcárcel, J., Acevedo, A. B., Tarque, N., Hube, M. A., Coronel, G., and Santa María, H.: Modeling the Residential Building Inventory in South America for Seismic Risk Assessment, Earthquake Spectra, 33, 299–322, https://doi.org/10.1193/101915eqs155dp, 2017.

---

## Author Comment (AC3)

The study deals with the analyses of various seismic risk scenarios for the Sabana Centro region in Colombia, located in the northern region of the capital city Bogotá, which concentrates important industrial facilities, educational facilities. It is an interesting study that follows the state-of-the-art procedures of scenario risk analyses (at least up to the computation of direct losses), for a region that hasn't been studied before, and thus is a good contribution to the scientific literature that communicates the seismic risk in the country.

Having said this, there are some comments about important issues in the study, that hopefully will help improve its clarity, coherence, and thoroughness. It must be said that after doing the review, the reviewer saw that many of the comments and limitations of the study were included in the discussion section as further developments, however there are many that should be included to make the results sound and representative of the region of study, otherwise many of the presented results could be very misleading.

**Dear reviewer**

You kindly spent time delving into our manuscript, and we are grateful. Thank you for acknowledging the value of our study. We will do our best to provide a complete explanation of the limitation of the study to make the results sound and representative, as you request in your comments.

**Section 2**

1. It would be interesting to include why the Sabana Centro region is of particular interest. In previous studies of the major cities, generated GDP or % of the population in comparison with the whole country were presented as reasons for the study of a particular city or region.

R/: Thank you for your comment. This is an important region in the department of Cundinamarca and for the authors is an interesting area for its economic and social growth rate in recent years. We state this in the manuscript, but to improve the statement, we added the following sentence to the updated manuscript:

Cundinamarca is one of the four most populated regions of the country and Sabana Centro is the province that contributes the most population (18%) to the department and also contributes 32% of the department's GDP.

2. Additionally, it is mentioned that it concentrates many economic and industrial activities, but at the end the analysis only deals with the residential building stock.

R/: The previous sentence was mentioned to highlight the fact about the province and its contribution within the region. As the reviewer says, we only deal with buildings for residential use. This is the predominant use in the study area. In addition, there is not enough information in the region to characterize commercial and industrial buildings.

3. Line 105: "The majority of the building stock of the region is comprised of one- and two-story houses" It would be good to show a reference with the numbers based on the 2018

Census for this. It is interesting that this is mentioned and still no two-story houses are considered in the analyses.

R/: Unfortunately, there is no information in the census related to the building height and the information in the national cadastral database does not include Sabana Centro. The phrase was based on the authors' fieldwork observations in the region based on 9000 remote and field surveys. Based on your comment we will include this information in the section that deals with the exposure model.

**Section 3.1 Seismic hazard**

4. Line 171: "eighteen crustal events were selected from the catalogue to be used in this study" Does this mean that only 'historical' events included in the catalogue were included? No new possible events from the event-based tables from the PSHA model?

R/: We appreciate your comment because our sentence was not clear. We indeed used eighteen events from the PSHA model we develop. Aware that our explanation is unclear, we updated it in the revised manuscript to clearly state our procedure.

5. Table 2: Include column with the distance to the population centroid taken as reference point in the disaggregation to be able to compare this selection of scenarios with the disaggregation graphs presented in Figure 4 and their representativeness in the overall 475-years return period hazard in the region. Given the proximity of the events based on the disaggregation, directivity effects should be considered for some of these events. As further seen in the discussion this was not considered but it would be good to mention it and not leaving it till the discussion as a further development.

R/: We appreciate this comment. Based on your suggestion and other reviewer's comments, we will include a figure instead of Table 2, allowing to visualize the location of the events. We will include a discussion about the directivity effects on this section.

**Section 3.2 Exposure model for the residential building stock**

6. Was the replacement cost updated to 2021? 2022? In which way was this done if indeed it has been updated? If not, it should be done and explained. Were the new inflated exposure building numbers (based on population as proxy) in any way compared to the dwellings or building numbers reported in the 2018 Census for these regions?

R/: Thank you for your questions and recommendations. The replacement cost was not updated in the original manuscript. Based on your comment, we decided to update the replacement costs using information of current prices for housing in the region. We compared the number of dwellings with those reported by the 2018 census and the difference is 28%. This difference is mainly because the census considered dwellings whose wall material is poured concrete and we did not include it since it was not possible to assign a percentage of construction. Clarifications in this regard are included in the manuscript and we decided to update the exposure model and use the information of the dwellings in the 2018 census

instead of the one based on population. Nonetheless, we think if worth highlighting that in the absence of census information, inferring based on population can provide a result with a moderate degree of approximation.

7. Table 4 only considers unreinforced masonry of 1 storey, which is known to be less vulnerable than the unreinforced masonry of 2 stories, which is actually more common in many urban areas. This typology should be included (assuming something probably based on census data or the surveys), as in the region it is very common to find 2-storey, in some cases more than single storey houses (as previously mentioned in the study also). In the current version, the study may be underestimating the losses.

R/: Thank you for this important comment. Other reviewers also raised their attention to the need of updating the exposure model, further discretizing between building heights. We agree specially that the difference between one- and two-story houses is significant. Based on this comment and the other reviewers' suggestions, we decided to update the exposure model using information of the 2018 national census, with further differentiation between building heights. To discretize the building height, we used information collected using remote and field surveys of 9000 houses in the Sabana Centro region, conducted by students of Universidad de La Sabana. We found that for houses between 1 and 3 stories, 34% are one-story houses, 48% are two-story houses and 18% are three-story houses. Accordingly, we updated our exposure model.

In addition, we decided to use the fragility functions calculated by Martins and Silva. (2021), which all use the same modeling approach, and they account for the differences in height. Furthermore, we also updated the discussion section.

**Section 3.3 Physical vulnerability of residential building stock to seismic ground shaking**

8. Chilean wood structures are known to be in better shape than those in Colombia, and they consider a different type of construction technique. The same goes to the curves used in HAZUS, which are not as representative of the local conditions and may be underestimating the risk. If they are going to be used a more thorough explanation of the limitation of using them should be included.

R/: Regarding the selection of fragility functions, we acknowledge the reviewer is correct about the representativeness of fragility functions.

After updating the exposure model accounting for differences between one- and two-story houses, for consistency in the revised version of the manuscript, we decided to use only analytical fragility functions. Most of them were those calculated by Martins and Silva. (2021) and Villar-Vega et al. (2017). These functions are analytical and all of them use the same modelling approach. We used these functions for several buildings, with some exceptions such as the thin reinforced concrete buildings, for which we kept the fragility functions developed by Arroyo et al. (2021), because these are also analytical and use a more accurate model for these types of structures. We kept the comments about the issue that the

fragility functions by Martins and Silva. (2021) were not developed accounting for the particularities of Colombian construction.

In the case of wood houses, we decided to keep the fragility curves calculated for Chilean buildings. These curves were developed by the authors using detailed drawings of Chilean houses, which are similar in configuration to the wood houses found in Sabana Centro. Regarding the method, we used a single DOF oscillator, like those used by Martins and Silva. (2021) and Villar-Vega et al. (2017). In general, the wood houses found in the field surveys in the region have good quality materials and adequate construction, thus we consider that the fragility curves used are suitable.

9. Table 5 needs a clarification of what each curve considers in each of the damage states. If the vulnerability model is considering a unique consequence model, there may be incompatibility between the loss ratios of the derived vulnerabilities, as each one considers each damage state in a specific different way. This is one of the main issues when combining vulnerability functions from different sources. This is particularly true given the damage results of the studies are shown considering these categories of the damage states. The reviewer saw this mentioned in the further developments of the discussion, but is should be included in the computation of the vulnerability curves here in some way, for the results to be coherent.

R/: We appreciate this important comment. Indeed, we agree with your argument and as we mentioned in our previous response, we decided to use a set of fragility functions derived using the same methodology.

10. Given the exposure is not considering separately the 2-storey housing, there are no vulnerabilities for 2-storeys considered, even when it is more common in the urban environment than the single storey houses. This should be included.

R/: Again, thank you for pointing us to this fact. As we mentioned in response to comment # 7, we updated our exposure model accounting for one, two and three-story houses.

**Section 3.4. Social Vulnerability (SV)**

11. Major comment: One of the main criticisms of the paper is the consideration of the social vulnerability index as a percentage increase using the expression (1+SVI). As stated in the study "The min-max normalization was used to standardize the SV indicators from zero to one to estimate the SVI per municipality. Higher score indicate more socially vulnerable municipalities, and lower scores reflects less vulnerable ones. Then, the indicators were integrated by summing them with equal weight, as followed in Contreras et al. (2020c). The resulting SVI index is therefore used to adjust the percentage of economic losses with respect to the cost presented by the building inventory, i.e. multiplying them by (1+SVI) (Carrenio et al., 2007).". The problem with this is that there is no analysis done on the significance of the variables included within the study and no way to know if there are variables that shouldn't be included and if anything is counted double.

Thank you for your comment. The reviewer is correct; in the submitted manuscript version, there was no analysis done on the significance of the variables included within the study and no way to know if there are variables that shouldn't be included and if anything is counted double. Therefore, considering this observation which is the same written by Reviewer 1, to avoid problems interpreting the model and overfitting, we checked the multicollinearity by looking at the variance inflation factor (VIF) of each variable and indicator. The VIF was identified in a linear regression that included collinearity diagnostics produced in SPSS (Field, 2005). We excluded those variables and indicators that were potentially correlated with others and those that did not add significant information according to the collinearity diagnostics (Table 2). Eventually, the model included only independent and relevant indicators to estimate the SV in the case study area (Tables 3 and 4).

**Coefficent[a]**

| Model | Unstandardized Coefficients | | Standardized Coefficients | t | Sig. | Collinearity Statistics | |
|---|---|---|---|---|---|---|---|
| | B | Std. Error | Beta | | | Tolerance | VIF |
| 1 (Constant) | 0.728 | 0.000 | | | | | |
| Indigenous population | 1.178 | 0.000 | 0.124 | | | 0.047 | 21.282 |
| Population density (inhabitants/km2) | -3.327 | 0.000 | -0.047 | | | 0.030 | 33.453 |
| Number of people per household | -3.378 | 0.000 | -0.012 | | | 0.029 | 34.994 |
| Population unemployed | 10.076 | 0.000 | 0.134 | | | 0.191 | 5.222 |
| Population with unsatisfied basic needs | -8.921 | 0.000 | -0.099 | | | 0.023 | 43.108 |
| Total population in poverty | 7.205 | 0.000 | 0.129 | | | 0.238 | 4.203 |
| Households with no electric energy access | 4.234 | 0.000 | 0.153 | | | 0.078 | 12.894 |
| No sewage system | 1.160 | 0.000 | 0.123 | | | 0.351 | 2.848 |
| Illiteracy rate | -1.619 | 0.000 | -0.027 | | | 0.209 | 4.785 |
| Deceased due to COVID-19 | 11.379 | 0.000 | 0.710 | | | 0.044 | 22.967 |

a. Dependent Variable: SV

Table 1. Variance inflation factor (VIF) detected.

**Excluded Variables/indicators[a]**

| Model | Beta In | t | Sig. | Partial Correlation | Collinearity Statistics | | |
|---|---|---|---|---|---|---|---|
| | | | | | Tolerance | VIF | Minimum Tolerance |
| 1 Female population | .[b] | | | | 0.000 | | 0.000 |
| Age dependance | .[b] | | | | 0.000 | | 0.000 |
| Total population | .[b] | | | | 0.000 | | 0.000 |
| Number of households | .[b] | | | | 0.000 | | 0.000 |

**Excluded Variables/indicators[a]**

| Model | Beta In | t | Sig. | Partial Correlation | Collinearity Statistics | | |
|---|---|---|---|---|---|---|---|
| | | | | | Tolerance | VIF | Minimum Tolerance |
| Population looking for employment | .[b] | | | | 0.000 | | 0.000 |
| Household with computer and internet | .[b] | | | | 0.000 | | 0.000 |
| Households with access to improved water source | .[b] | | | | 0.000 | | 0.000 |
| Education level completed primary | .[b] | | | | 0.000 | | 0.000 |
| Education level secondary | .[b] | | | | 0.000 | | 0.000 |
| Population enrolled in education institution | .[b] | | | | 0.000 | | 0.000 |
| Hospital , clinics per 1000 population | .[b] | | | | 0.000 | | 0.000 |
| Population with no healthcare | .[b] | | | | 0.000 | | 0.000 |
| Population registered to national healthcare | .[b] | | | | 0.000 | | 0.000 |
| COVID-19 cases confirmed | .[b] | | | | 0.000 | | 0.000 |
| COVID-19 cases active | .[b] | | | | 0.000 | | 0.000 |
| People recovered from COVID-19 | .[b] | | | | 0.000 | | 0.000 |

a. Dependent Variable: SV

b. Predictors in the Model: (Constant), People dead due to COVID-19, Total population in poverty, No sewage system, Number of people per household, Native indigenous population, Population unemployed, Illiteracy rate, Population density (inhabitants/km2), Households with no electric energy access, population with unsatisfied basic needs

Table 2. Excluded variables/indicators.

**Collinearity Diagnostics[a]**

| Model | | Eigenvalue | Condition Index | Variance Proportions | | | | | | | | | | |
|---|---|---|---|---|---|---|---|---|---|---|---|---|---|---|
| | | | | (Constant) | indigenous population | Population density (inhabit-ants/km2) | Number of people per household | Population unemployed | Population with unsatisfied basic needs | Total population in poverty | Households with no access to electricity | No Sewage system | Illiteracy rate | People dead due to COVID-19 |
| 1 | 1 | 8.495 | 1.000 | 0.00 | 0.00 | 0.00 | 0.00 | 0.00 | 0.00 | 0.00 | 0.00 | 0.00 | 0.00 | 0.00 |
| | 2 | 1.004 | 2.909 | 0.00 | 0.02 | 0.00 | 0.00 | 0.00 | 0.00 | 0.00 | 0.00 | 0.08 | 0.00 | 0.00 |
| | 3 | 0.740 | 3.389 | 0.00 | 0.00 | 0.00 | 0.00 | 0.00 | 0.00 | 0.00 | 0.00 | 0.15 | 0.00 | 0.01 |
| | 4 | 0.517 | 4.053 | 0.00 | 0.02 | 0.00 | 0.00 | 0.00 | 0.00 | 0.00 | 0.00 | 0.13 | 0.00 | 0.02 |
| | 5 | 0.102 | 9.134 | 0.00 | 0.03 | 0.00 | 0.00 | 0.00 | 0.00 | 0.01 | 0.09 | 0.00 | 0.01 | 0.04 |
| | 6 | 0.066 | 11.307 | 0.00 | 0.00 | 0.00 | 0.00 | 0.03 | 0.00 | 0.15 | 0.04 | 0.02 | 0.02 | 0.00 |
| | 7 | 0.044 | 13.893 | 0.00 | 0.00 | 0.00 | 0.00 | 0.00 | 0.00 | 0.10 | 0.01 | 0.01 | 0.16 | 0.00 |
| | 8 | 0.017 | 22.300 | 0.00 | 0.00 | 0.01 | 0.00 | 0.21 | 0.01 | 0.01 | 0.08 | 0.07 | 0.01 | 0.11 |
| | 9 | 0.011 | 27.201 | 0.00 | 0.00 | 0.00 | 0.00 | 0.00 | 0.02 | 0.12 | 0.00 | 0.03 | 0.32 | 0.03 |
| | 10 | 0.004 | 48.357 | 0.00 | 0.04 | 0.24 | 0.00 | 0.75 | 0.00 | 0.45 | 0.73 | 0.08 | 0.02 | 0.08 |
| | 11 | 4.453E-5 | 436.754 | 1.00 | 0.89 | 0.74 | 1.00 | 0.00 | 0.96 | 0.16 | 0.05 | 0.42 | 0.47 | 0.71 |

a. Dependent Variable: SV

Table. 3. Selected variables/indicators.

| Composite indicators | Indicators |
|---|---|
| Population | Native Indigenous population |
| | Population density (inhabitants/km2) |
| | Number of people per household |
| Economy | Population unemployed |
| | Population with unsatisfied basic needs (UBN) |
| | Total population in poverty |
| Infrastructure | Households with no electric energy access |
| | No sewage system |
| Education | Illiteracy rate |
| Health | Deceased due to COVID-19 |

Table. 4. Selected composite indicators and indicators.

12. Additionally, considering this index as a "percentage increase" is extremely misleading. If there was a way to correlate the SVI of each variable in economic terms to the direct economic loss, then this could be done. But this is not done and there is no parametric study or anything else to validate any of the assumptions.

R/: Thank you very much to the reviewer for raising this interesting question. Little research has tested the correlation between social vulnerability (SV) and losses. To our best knowledge, the relationship between SV and modeled losses has been so far informative rather than indicating that total losses (measured as dollar losses or debris generated) increase with SV (Schmidtlein et al, 2011). However, it was found that only relative losses (dollar losses per average family income) tend to increase with SV. Case study areas with a low SV tend to have more material goods with significant monetary value (dollar) exposed to risk, than areas with high SV. Therefore, we should expect a negative correlation between property losses and SV (Cutter and Finch, 2008). It is important to understand that while total loss (dollar) in case study areas with high SV is lower, the impact of those losses in their communities is high (Schmidtlein et al., 2011).

We will include a discussion of this in the updated manuscript.

13. This SVI cannot be considered a percentage unless there is backup data validating this. This has been done also in fatality modelling where the models that are presented in any publication are previously calibrated and validated with data from historic events. Moreover, considering previous events reporting post-loss amplification that include costs from the response and recovery stages in some disasters, it has been shown that numbers over 30-40% are almost non-existent  (what is demand surge? Olsend and Porter 2010), while this study mention cases with increases of up to 60%. There may be a problem with the explanation of the methodology, but as it is right now it is very difficult understand how it can relate to economic losses, especially direct physical losses after an event.

R/: Thanks again for this question. Social vulnerability assessment considers variables and indicators with different units, e.g., number, percentages, number of people/m2, etc. This is the main reason to use normalization to construct the social vulnerability index (SVI), which is why there is no unit, and the levels of SV are expressed in ordinal categories, e.g., high, medium, and low, according to ranges defined by authors. The reason to use percentage is

that this value is integrated with the percentage of economic losses with respect to the cost presented by the building inventory.

14. (These limitations are also afterwards mentioned by the authors in the discussion, but it is a MAJOR limitation of the inclusion of the SVI methodology in the results in this study, as there is no validation or calibration of any kind for the methodology)

R/: Data related to the damages caused by natural hazards are usually low quality. It is difficult to compare the severity of the event's characteristics in different zones of any case study area (Schmidtlein et al, 2011), because it is wrong to assume that losses are uniformly distributed (Cutter and Finch, 2008). Unfortunately, SV is estimated at the national or regional scale, making it difficult to calibrate or validate any SV model based on damages after earthquakes, which will be an ideal procedure, as the reviewer suggested.

Overall, considering all your valuable comments about the SV, we decided to improve the calculations of these indexes, as indicated in the response to previous comments. We did, however, decided to keep the estimation of the losses based on this index, but we were careful to clearly state that we did so to provide an approximation of the impact of social vulnerability on economic losses, and that there is a need of a more accurate procedure to do this. In addition to commenting this on the losses section, we also added a paragraph to the discussion.

15. Table 9 numbers are misleading as a direct non-weighted average of the 18 scenarios is not probabilistically and statistically sound. It should consider the contribution of each event, otherwise the less probable events are counted in the same way as the more probable ones. In this way, as when computing AAL from a probabilistic analysis, the contributions should consider the probability of occurrence of each scenario. After saying this, it is advised not to present this table and instead present one with the analysis of each scenario done separately as in a deterministic approach, unless it is possible to demonstrate that the 18 scenarios included account for the 100% of the 475 years return period loss and a weighted average is calculated based on the contribution of each.

R/: We appreciate this comment. We agree that 18 scenarios are not enough to represent 100% of the hazard. We should though mention that the selected scenarios do represent a significant percent of the contribution to the seismic hazard, as the examination of figure 4 can reveal, particularly for the PGA. We added this comment in the corresponding section of the revised manuscript.

In the light of the comment, we decided to remove table 9 and instead include a figure that shows the statistical variability of each damage state for the eighteen scenarios.

**Section 4.4:**

16. As stated previously, to present these analyses, it would be important to show the contribution of the 18 scenarios to the total hazard in the region (based on the disaggregation results). If not Figure 13 is misleading, considering that it says: "Within the municipalities, the mean percentage of losses is presented with respect to the total expected losses in the region".

"The economic losses experienced by a province due to an earthquake depends on the event's epicenter as it is depicted in Figure 12." Figure 12 does not show in any way anything regarding the epicenter. It may not be only the epicenter but also the Mw for each event the cause of the differences, so this statement is not provable from the Figure. Delete it.

R/: We understand the reviewer comment and agree that each scenario has a different contribution to the hazard. Accordingly, we decided to remove Figure 13. We kept figure 12 but we improve our analysis of this figure. The main point here is that Sabana Centro has a uneven distribution of the building stock, therefore, similar earthquakes with the same magnitude and similar depth such as the Mw 6.25 Sopó, Mw 6.25 Tabio and Mw 6.25 Nemocón have different consequences in terms of economic losses.

**Discussion:**

17. Just until this section this is stated: "The simulations of eighteen seismic scenarios with a return period of 475 years show that half of the building stock will experience some degree of damage". How was this '475 years' return period calculated? Even when the disaggregation was done for the '475 years' return period, how can it be confirmed that the 18 scenarios add up to the 100% contribution for the hazard for this return period? Either way this statement should be included in some way in previous sections and not only until the discussion.

R/: Thank you for this observation. As we mentioned in answer to question 15, we will add this to the corresponding section in the manuscript, before the discussion.

**Effects of SV**

18. Major comment: The main criticism for this approach is also stated by the authors in this sentence: "First, as not all social aspects exert equel effect after an earthquake, it is necessary to develop a weighted approach to best estimate a more realistice SVI for earthquake events.

R/: Thank you for bringing this statement to our attention. We have decided to eliminate it because the allocation of a weighted approach not only will not capture all social aspects after an earthquake but allocating weights to the composite indicators will add subjectivity to the analysis.

19. A second improvement required is to devise a better way of estimating the economic impact of social vulnerability. One potential approach is to generate a database of past earthquakes with different consequences that include the economic costs". There are not needed future improvements but major limitations of the proposed approach. In this kind analyses, as when performing a linear regression, it is important to avoid counting double and establish the significance of each variable within the analysis, if not this could be overestimating the vulnerability and losses in the region considerably.

R/: Thank you again for this observation. We agree with the reviewer that to avoid problems interpreting the model and overfitting, we checked the multicollinearity by looking at the variance inflation factor (VIF) of each variable and indicator. The VIF was identified in a linear regression that included collinearity diagnostics produced in SPSS (Field, 2005). We

excluded those variables and indicators that were potentially correlated with others and those that did not add significant information according to the collinearity diagnostics (Table 2). Eventually, the model included only independent and relevant indicators to estimate the SV in the case study area (Tables 3 and 4 shown above in the answer to comment 11).

However, in this sentence, we were referring to the economic impact of the SV after an earthquake, not the method for the vulnerability assessment. Considering that this sentence can be misunderstood, we also decided to eliminate it.

20. Also, using the min and max approach is very subjective as many variables as unemployment and poverty are tempered with by local organism. This kind of indicators are good to compare and prioritize actions withing regions but cannot be used in the way they are presented in this study to increase direct physical losses.

Thanks for this comment. However, it is necessary to differentiate the method, the indicators and data sources in the assessment of SV, the operationalization of the analysis, and the actions. Regarding the method, as we explained before, SV assessment considers variables and indicators with different units. This is the main reason to use normalization to construct the social vulnerability index (SVI). With respect to the indicators, we agree with the reviewer that governments can temper data regarding unemployment and poverty in the most vulnerable countries, this data does not exist, or it is not accessible, but any analysis has an uncertainty level, and we decided to accept it using official numbers. As we explained before based on the literature review, integrating the level of SV to the physical losses will not produce a significant increase in the last one, considering that they are negatively correlated (Cutter and Finch, 2008). Integrating the SV analysis makes the risk assessment more holistic and useful to prioritize actions at the scale of the assessment because it can be national, regional, or municipal.

**Minor modifications:**

21. Line 118: Typo: This GROWTH counts for 64% of the total population of the region

22. Line 248: Incomplete sentence: "The number of masonry buildings represents the 88.61% of the total buildings in Sabana Centro, whereas those of concrete and wood represent 3.16% and 8.24%, respectively, with."

23. Line 261: Repeated to "In the absence of specific curves locally developed for the Sabana Centro province, fragility curves were selected to to represent these structures"

R/: We appreciate the reviewer pointing to these typos and we fixed them in the updated manuscript.

**References**

Arroyo, O., Feliciano, D., Carrillo, J., and Hube, M. A.: Seismic performance of mid-rise thin concrete wall buildings lightly reinforced with deformed bars or welded wire mesh, Engineering Structures, 241, https://doi.org/10.1016/j.engstruct.2021.112455, 2021.

Cutter, S. L., & Finch, C.: Temporal and spatial changes in social vulnerability to natural hazards. Proceedings of the National Academy of Sciences, 105(7), 2301-2306. doi:doi:10.1073/pnas.0710375105, 2008.

Field, A.: Discovering statistics using SPSS. Chenai: Sage publications, 2005.

Martins, L. and Silva, V.: Development of a fragility and vulnerability model for global seismic risk analyses, Bulletin of Earthquake Engineering, 19, 6719–6745, https://doi.org/10.1007/s10518-020-00885-1, 2021.

Schmidtlein, M. C., Shafer, J. M., Berry, M., & Cutter, S. L.: Modeled earthquake losses and social vulnerability in Charleston, South Carolina. Applied Geography, 31(1), 269-281. doi:https://doi.org/10.1016/j.apgeog.2010.06.001, 2011.

Villar-Vega, M., Silva, V., Crowley, H., Yepes, C., Tarque, N., Acevedo, A. B., Hube, M. A., Gustavo, C. D., and Santa María, H.: Development of a fragility model for the residential building stock in South America, Earthquake Spectra, 33, 581–604, https://doi.org/10.1193/010716eqs005m, 2017.

---

## Author Response (AR1)

**Reviewer 1: Ana Acevedo**

**Dear Dr. Ana Acevedo**

You kindly spent time delving into our manuscript, and we are grateful. Thank you for your appreciation of the study's performance and writing style and acknowledging the relevant information for seismic risk in the region. Thank you for all your comments. We will do our best to provide a complete answer to all of them.

We have used a color code to answer your valuable questions, comments, and suggestion. Please find them already answered in grey and the respective answers in black. The corresponding paragraph in the paper with changes is in blue

1. My main concern regards the selected fragility curves: I find it complex to perform a risk analysis using fragility functions developed by different methodologies and as stated by the authors, with different limit states definitions. I believe this is one issue that requires additional explanation. For example, comparison between fragility functions is not presented. Do the set of curves behave as expected? Information given in Table 5 is not enough. The article should present figures that allows for a visually appreciation of the curves. A brief explanation of the methodologies uses for the curves' development should also be included, as well as an opinion about how reliable the curves are.

R/: Thank you for your main comment. We agree with your point. Other reviewers also raised their attention to this fact. Therefore, for consistency in the revised version of the manuscript, we decided to use only analytical fragility functions. Most of them were those calculated by Martins and Silva. (2021) and Villar-Vega et al. (2017). These functions are analytical and all of them use the same modelling approach to avoid incorporating inconsistencies in the damage and losses estimations. We kept the comments about the issue that the fragility functions by Martins and Silva. (2021) were not developed accounting for the particularities of Colombian construction.

Accordingly, we included the following texts in different sections:

**Introduction**

'(…) Regarding the structural vulnerability of the building stock, a database of fragility functions developed for the residential building stock in South America Villar-Vega et al. (2017) and those developed for global seismic risk analysis (Martins and Silva, 2021). (…)'

**3.3 Physical vulnerability of residential building stock to seismic ground shaking**

' (…) The Physical Vulnerability Suite of the GEM Foundation (OpenQuake Platform - Vulnerability, 2021). was considered for the review. The GEM database for the specific case of Colombia has the curves developed by Acevedo et al. (2017) for unreinforced masonry houses constructed in Antioquia, Colombia. There are some curves for reinforced concrete buildings with geographical applicability in Manizales, Colombia by Bonett (2003) and the dataset of Villar-Vega. (2014) for South America. Although the set of curves covers different types of buildings, they are calculated based on different methodologies and different damage states.

Another available dataset of fragility curves is those developed by Martins and Silva (2021), who covered nearly 500 building classes at global level including Colombia. The fragility is calculated from nonlinear dynamic analyses performed on equivalent single-degree-of-freedom (SDOF) oscillators. They considered four damage states that are also intended to study in the present research: slight, moderate, extensive and collapse. The respective damage thresholds were defined based on the spectral displacement of the structures. At regional level also are the set of fragility curves for the residential building stock in South America (Villar-Vega et al., 2017), covering 54 common building classes. The methodology used for the derivation of the curves is similar to the one used in Martins and Silva. (2021).

Based on the information collected, the fragility curves available in Martins and Silva. (2021) were used mainly and complemented with those of Villar-Vega et al. (2017). These curves were selected in order to prevent an unbiased comparison of risk between the different municipalities in the region due to the different methodologies used to develop the fragility curves. Therefore, a set of 33 fragility functions was used to represent the probability of exceeding a level of damage conditioned to ground shaking intensity. These functions are comprised of 28 sets of curves reported by Martins and Silva. (2021) and five sets developed by Villar-Vega et al. (2021). The last one is assigned to non-Ductile confined masonry, 1, 2 and 3 stories and Ductile light wood members, 1 and 2, since in the former these building classes

were not included. These fragility functions are described by a cumulative probability curve with a lognormal distribution and examples of some of them are presented in Figure 10.

[Figure]

**Figure 10.** Fragility curves for 14 of the 33 building classes listed in Table. The curves describe the differential seismic vulnerabilities for the predominant building classes for each type of material: reinforced concrete (CR), confined masonry (MCF), unreinforced masonry (MUR) and wood (W).

2. How can the authors explain that the number of collapse buildings is almost three times the number of buildings with extensive damage? Furthermore, the number of collapse buildings exceeds the number of buildings in any other damage state.

R/: We updated the result and discussion in the revised manuscript, with the results obtained using a new exposure model and the set of fragility function described in the answer of comment # 1.

Regarding the question of about the number of collapses in the initial manuscript, in the scenarios developed using the initial selected set of fragility functions, the intensity measure for several earthquakes was in the range where collapse had the highest probability for nonductile systems based on unreinforced masonry, which are ubiquitous in the region and drive the number of collapses. In the updated manuscript the collapses are 9.85% in the Mw 5.95 Chía scenario, twice the buildings with extensive damage and less than half the slight damage. The reason for this high percentage of collapse is that the earthquake in this scenario is close to Cajicá and Chía, which have 40% of the building stock of Sabana Centro and a significant number of them are non-ductile buildings (82%). The collapsed buildings in these municipalities accounted for 19% of their stock, driving up the overall percentage of the region. We also would like to note that in the Popayán earthquake in 1985, the collapses accounted for 12% of the stock.

We state these facts in section 4.1of the updated manuscript.

**4.1. Damage forecast**

'(…) . Nearly ten percent of collapse rises concerns from a decision maker perspective, but two Colombian events put the results in perspective: the Mw 6.1 earthquake in Armenia (1999) and the Mw 5.5 earthquake in Popayán (1983). In the former, the records indicate that 17551 buildings were destroyed, 18421 had severe damage and 43474 had moderate damage. In the latter, which occurred at an estimated depth between 12 km and 15 km, 12% of buildings suffered complete damage. In both earthquakes, damage concentrated in unreinforced masonry buildings, constructed prior to the enactment of the Colombian seismic design code in 1998. More than 60% of the building stock in Sabana Centro is comprised of that type of buildings, and what is more, 35% are two- and three-story houses (Table 4), which are more vulnerable than those of one-story houses (Heresi and Miranda., 2022). These buildings are expected to withstand significant damage during an earthquake such as the Chía Mw 5.95 shown here, which is similar in magnitude and depth to the Armenia earthquake and for which the percentage of collapse herein presented is similar to that from the Popayán earthquake.

In terms of municipalities, the higher damage occurs in Chía and Cajicá, with 3522 and 2271 collapsed buildings. Compared to the total buildings of each municipality, collapses account for 19.0% and 19.7%, respectively. Overall, they account for 5793 out of the 7463 collapsed buildings for this scenario (77%). In contrast, Cogua was the municipality with the least number of damaged buildings, as roughly 90% of the inventory did not experience any type of damage and only 0.33% of them collapsed. Nemocón had the least damages after Cogua, with 2.4% of collapses. These results are reasonable because Chía and Cajicá are closer to the epicenter in this scenario and they have the highest building inventory of the region, together with Zipaquirá. Besides, a significant part of their building inventory is comprised of nonductile unreinforced masonry. On the other hand, despite having a similar distribution of the building inventory, Cogua and Nemocón are the farthest municipalities from the epicenter. The main difference between these two is that Nemocón has softer soils, with roughly one fourth of the municipality under 180 m/s, thus the higher percentage of collapses. (…)'

**Additional comments:**

3. Why do all the scenarios are crustal shallow events? In the article it is mentioned that for SA (1.0s) there is an important contribution of subduction events. As the number of scenarios is important (18) some of them should be subduction events.

R/: We looked at the contributions by tectonic environments to the seismic hazard at the PGA Sa (0.3) and Sa(1.0s) In figure 1 it is possible to observe that the seismic hazard is controlled by active shallow crust events, being its contribution greater than the 75% in all the municipalities, with most of them under 20% and in some cases like Chia (the largest population), it is below 10%. The contribution of the Bucaramanga seismic nest is more relevant in the municipalities at the north of the region in Cogua, Nemocón and Zipaquirá, and the first two have low populations compared to the others in the province. In such cases, the contribution of the Bucaramanga seismic nest is around 25%. Based on this, we decided to keep the eighteen crustal scenarios.

[Figure]

Figure 1. Contribution of tectonic environments to the seismic hazard of the Sabana Centro municipalities.

Figure 2 presents the results of the disaggregation of the seismic hazard in the municipalities of Cota, Cajicá and Cogua, considering both the distance and magnitude of the events. Such results were obtained from SGC.

[Figure]

**Figure 2**. Seismic hazard disaggregation by magnitude and distance of earthquakes for Cota, Cajicá and Cogua. Ground motion Intensities expressed in terms of PGA, SA (0.3) and SA(1.0).

The municipalities of figure 2 were selected considering the contribution of the tectonic environments (figure 1). The municipality of Cota is considered as an example of municipalities of the Sabana Centro Region where the seismic hazard is controlled by shallow events. In the case of Cota, the events with the larger contributions are located at a distance lower than 60 km, with magnitudes ranging between 5.5 and 7.0 Mw. On the other hand, the municipality of Cogua is considered as an example of municipalities of the Sabana Centro Region where the contribution of the Bucaramanga seismic nest is relevant (more than 20%). For spectral accelerations of a 0.3 second vibration period, events located at distances varying between than 60 and 180 km, with magnitudes ranging between 6.0 and 8.0 Mw have a contribution close to the 40%. In this range of distances, the greater contribution is observed in magnitudes varying between 6.5 and 7.0 Mw.

Based on the disaggregation by coordinates and tectonic environments and coordinates and ranges of magnitudes, the location of an event from the Bucaramanga seismic nest was selected. The dip, slip and rake angles of the rupture were assigned considering information from the ISC-GEM Catalog and the national seismic hazard model of Colombia (Arcila et al., 2020). The properties of the rupture are presented in the following table:

| Propiedad | D4 |
|---|---|
| Fuente | GCMT |
| ID | 105195 |
| Longitud | -73.753 |
| Latitud | 5.7356 |
| Magnitud (Mw) | 6.75 |
| Profundidad (km) | 105 |
| Límite superior de la sismicidad (km) | 100 |
| Límite inferior de la sismicidad (km) | 140 |
| Rake (°) | -90 |
| Dip (°) | 37 |

We decided to run this scenario and obtained the following results:

| | No damage | Slight | Moderate | Extensive | Collapse |
|---|---|---|---|---|---|
| Number of buildings | 71366 | 3562 | 532 | 196 | 121 |
| Percentage (%) | 94.18 | 4.70 | 0.70 | 0.26 | 0.16 |

All these things considered, we decided to keep the eighteen scenarios selected initially.

4. The authors mentioned the use of population census data to infer the number of buildings added to the original exposure model of SARA. It is not clear why the authors did not use the census data to directly obtain the number of buildings. The 2018 Census provides relevant information that can be used to have a more precise number of buildings.

R/: We appreciate a lot this comment. We used population as a simple and practical process to directly obtain the number of buildings. The census provided information to obtain the number of dwellings. We compared our results with those reported by the 2018 census and the difference is 2.8%. This difference is mainly because the census considered dwellings whose wall material is poured concrete and we did not include it since it was not possible to assign a percentage of construction.

Based on your comments and those of the other reviewers, we decided to update the exposure model and use the information of the dwellings in the 2018 census instead of the one based on population. Nonetheless, we think if worth highlighting that in the absence of census information, inferring based on population can provide a result with a moderate degree of approximation. The number of buildings inferred with the population was 79,222 and with the census data it is 75,778, a difference of 4%.

**Introduction**

'(…) Information available from the national census was used to create the exposure model. The methodology used in (Yepes-Estrada et al., 2017) was followed to assign the number of buildings per municipality. (…)'

**3.2. Exposure model for the residential building stock**

' (…) The building exposure model for the region has information about the building classes, the number of buildings, inhabitants, and the buildings' replacement costs. To develop this model for Sabana Centro, the methodology used by the South America Risk Assessment (SARA) project to develop exposure models in South America (Yepes-Estrada et al., 2017) was taken as a basis. The source of information to assign the number of buildings was the 2018 national census (DANE, 2018). The census allowed having information on the number of dwellings and typical wall and roof materials, which were used to infer the different classes of buildings by municipality. A total of 156,628 dwellings were calculated; this number differs from that reported by the national census by 2.8%, since it did not consider dwellings whose wall material is poured concrete. This material was not included since there was no information available to relate it to any type of building class. The set of dwellings were related to the same building classes and same relationships ('mapping schemes') used in Yepes-Estrada et al. (2017). As the census information is reported in terms of dwellings the procedure used in Yepes-Estrada et al. (2017) to calculate the number of buildings was also followed. (…)'

5. How was the building typology assigned to the added buildings to the original exposure model?

R/: Initially we updated the number of buildings keeping the same typologies and the same relative percentages between building typologies. Based on your comments and those of the other reviewers to have a more reliable estimation we decided to update the exposure model following the methodology used in Yepes-Estrada et al. (2017), where taxonomies are assigned according to the materials of the walls and roofs of the dwellings. The information used for this process was obtained directly from the census data.

6. Which replacement cost did the authors use? The authors only mention that the cost is assigned according to the socio-economic levels, but it is not clear which cost was used and how was it computed: cost per area? Cost per building? It is suggested to include the replacement cost in Colombian pesos as the exchange currency fluctuates.

R/: Thank you for this comment, we have included a sentence that mentions it. We have used the cost per building which was calculated based on the information available in the Territorial Statistical Systems (TerriData). We decided to keep the replacement cost in USD but introduced information on the most recent average exchange rate from dollars to Colombian pesos.

**3.2. Exposure model for the residential building stock**

'(…) The building replacement cost refers to the cost of structural and non-structural components of a building and it is a value associated with the building's rehabilitation. This study has only considered the structural cost per building calculated based on cadastral information available in the Territorial Statistical Systems (TerriData[1]) of the country. This replacement cost was computer per building, expressed in USD. As the currency in Colombia is in Colombian pesos, the exchange to U.S. dollars was made for an average exchange rate of 4080 pesos. (…)'

7. It is not clear how the information of the base exposure model (SARA) was complemented with the information of the 6249 surveys. Furthermore, all these buildings belong to the same municipality. A description of the buildings characteristics of each municipality should be included.

R/: We used the information of the surveys to update the percentage of the taxonomies only in the municipality of Chía because we had remote field surveys of this municipality that were conducted by students of Universidad de La Sabana. This information was also used to disaggregate the number of buildings per height, allowing to improve damage estimates. It was useful because the methodology used in Yepes-Estrada et al. (2017) aggregated the building classes in a range of heights and we separated it in 1,2 and 3 stories and research by Heresi and Miranda (2022) showed that this practice should not be used because it leads to misestimation of damages.

**3.2. Exposure model for the residential building stock**

'(…) Details of this process can be consulted in Arroyo et al. (2022). This procedure allowed to compare the percentages of the building classes calculated based on the SARA methodology and discretize the buildings by height. (…)'

8. The authors mention that 8.24% of the stock are wood buildings. How does this information compare to the Census data? (The Census provides information about building's wall material). In addition, the authors assigned a fragility function for wood buildings developed for Chilean buildings. Although the reference of the fragility functions used for wood has not yet been published, it is not clear that Colombian wood buildings have the same seismic behavior as Chilean wood buildings. A support for the use of Chilean wood fragility functions is needed.

R/: In the previous exposure model we only compared the total number of dwellings. We updated the exposure model using the census data which includes walls and roof materials. This allowed us a better estimation of the total number of buildings for each taxonomy. In the case of wood buildings there is a total of 3762 (4.96%) which is more representative of what is in the region according to field inspections.
* * *
[1] https://terridata.dnp.gov.co/index-app.html#/

In general, the wood houses found in the field surveys in the region have good quality materials and adequate construction, thus we initially consider the Chilean fragility curves. On their recommendation and that of the other reviewers, we decided to replace those fragility curves with those calculated in Villar-Vega et al. (2017), applicable to regional scales.

9. It is not clear why the authors use only two building heights: 1 and 4. Does the exposure model only comprise building with 1 and 4 stories? Or does the exposure model have buildings of several number of stories, but the authors decided to group them in just to building heights? Whatever the option, for a region where most of the buildings are low-rise buildings (as stated in the paper) a differentiation of number of stories is very important.

R/: Thank you for this important comment. Other reviewers also raised their attention to the need of updating the exposure model, further discretizing between building heights. We agree specially that the difference between one- and two-story houses is significant. Based on this comment and the other reviewers' suggestions, we decided to update the exposure model using information of the 2018 national census, with further differentiation between building heights. To discretize the building height, we used information collected using remote and field surveys of 6249 houses in the Sabana Centro region, conducted by students of Universidad de La Sabana. We found that for houses between 1 and 3 stories, 34% are one-story houses, 48% are two-story houses and 18% are three-story houses. Accordingly, we updated our exposure model.

In addition, we decided to use the fragility functions calculated by Martins and Silva. (2021), which all use the same modeling approach, and the ones of Villar-Vega et al. (2017) which account for the differences in height.

10. Results should include the uncertainty as 1000 ground motion fields were generated and two GMPEs were used.

R/: Thank you for this recommendation. We included statistics in the updated version of the manuscript. The following table presented the standard deviation for the damages calculated in each municipality and for each GMPE:

**4.1 Damage forecast**

**Table 8.** Number and percentage of buildings expected to suffer damage in the region after the Mw 5.95 earthquake scenario in Chía. The mean and standard deviation (Stdv) for each of the GMPE and damage states are presented. The mean[a] is calculated with the corresponding weights for each GMPE (Abrahamson et al. (2014): 0.35 and Cauzzi et al. (2014): 0.65). The total number and percentage of buildings with some states of damage are at the end of the table.

| Municipality | GMPE | No damage | | Slight | | Moderate | | Extensive | | Collapse | |
|---|---|---|---|---|---|---|---|---|---|---|---|
| | | Mean | Stdv | Mean | Stdv | Mean | Stdv | Mean | Stdv | Mean | Stdv |
| Cajicá | Abrahamson | 5520 | 904 | 3700 | 420 | 1133 | 298 | 560 | 193 | 590 | 332 |
| | Cauzzi | 2745 | 1067 | 3015 | 727 | 1486 | 347 | 1084 | 268 | 3176 | 1270 |
| | Mean[a] | 3716 | | 3255 | | 1362 | | 901 | | 2271 | |
| Chía | Abrahamson | 9173 | 1493 | 5809 | 689 | 1773 | 486 | 872 | 311 | 916 | 545 |
| | Cauzzi | 4817 | 1779 | 4870 | 1170 | 2285 | 568 | 1645 | 427 | 4926 | 2084 |
| | Mean[a] | 6342 | | 5199 | | 2105 | | 1374 | | 3522 | |
| Cogua | Abrahamson | 2688 | 209 | 257 | 148 | 38 | 42 | 13 | 18 | 9 | 24 |
| | Cauzzi | 2712 | 211 | 233 | 147 | 37 | 43 | 13 | 20 | 10 | 28 |
| | Mean[a] | 2704 | | 241 | | 37 | | 13 | | 10 | |
| Cota | Abrahamson | 3175 | 484 | 932 | 264 | 233 | 124 | 103 | 70 | 120 | 145 |
| | Cauzzi | 3313 | 508 | 817 | 284 | 206 | 121 | 99 | 71 | 128 | 168 |
| | Mean[a] | 3264 | | 858 | | 216 | | 100 | | 126 | |
| Gachancipá | Abrahamson | 1777 | 232 | 519 | 134 | 116 | 60 | 47 | 31 | 40 | 42 |
| | Cauzzi | 1637 | 289 | 530 | 153 | 152 | 74 | 77 | 45 | 103 | 101 |
| | Mean[a] | 1686 | | 526 | | 140 | | 66 | | 81 | |
| Nemocón | Abrahamson | 1408 | 164 | 337 | 102 | 66 | 40 | 25 | 19 | 18 | 21 |
| | Cauzzi | 1282 | 212 | 364 | 113 | 99 | 54 | 49 | 32 | 60 | 64 |
| | Mean[a] | 1326 | | 355 | | 88 | | 41 | | 45 | |
| Sopó | Abrahamson | 1258 | 341 | 1116 | 197 | 476 | 101 | 290 | 81 | 520 | 269 |
| | Cauzzi | 1403 | 428 | 1009 | 220 | 414 | 115 | 267 | 88 | 566 | 319 |
| | Mean[a] | 1352 | | 1047 | | 436 | | 275 | | 550 | |
| Tabio | Abrahamson | 1745 | 319 | 775 | 159 | 225 | 89 | 108 | 56 | 135 | 122 |

| Municipality | Model | | | | | | | | | | |
|---|---|---|---|---|---|---|---|---|---|---|---|
| | Cauzzi | 1918 | 346 | 668 | 180 | 185 | 89 | 93 | 55 | 124 | 126 |
| | Mean[a] | 1858 | | 705 | | 199 | | 98 | | 128 | |
| Tenjo | Abrahamson | 1976 | 248 | 627 | 140 | 142 | 67 | 58 | 35 | 48 | 44 |
| | Cauzzi | 1678 | 330 | 657 | 165 | 210 | 84 | 117 | 56 | 189 | 156 |
| | Mean[a] | 1782 | | 647 | | 186 | | 96 | | 140 | |
| Tocancipá | Abrahamson | 2807 | 463 | 1443 | 222 | 406 | 141 | 187 | 86 | 189 | 147 |
| | Cauzzi | 2196 | 593 | 1361 | 303 | 513 | 159 | 317 | 118 | 646 | 418 |
| | Mean | 2410 | | 1390 | | 475 | | 272 | | 486 | |
| Zipaquirá | Abrahamson | 16638 | 1568 | 2067 | 1048 | 349 | 341 | 125 | 155 | 98 | 207 |
| | Cauzzi | 16873 | 1589 | 1847 | 1047 | 326 | 344 | 124 | 164 | 108 | 247 |
| | Mean[a] | 16791 | | 1924 | | 334 | | 125 | | 105 | |

| Total | | | | | | |
|---|---|---|---|---|---|---|
| | Number of buildings | 43.230 | 16,145 | 5579 | 3361 | 7463 |
| | Percentage of buildings (%) | 57.05 | 21.31 | 7.36 | 4.44 | 9.85 |

11. The taxonomy MCF/DNO/H:1 is not correct as it is missing the lateral load resisting system.

R/: Thank you for noting this omission. We fixed this in the updated manuscript.

12. The taxonomy CR/LFINF/DUM/H:4 is used for buildings constructed using thin RC walls. This is not the original definition in the GEM taxonomy. It is suggested to use a different taxonomy.

R/: Thank you for this observation. We updated the taxonomy consistent with the GEM methodology.

13. The taxonomy W/H1 is missing the information about the lateral load resisting system and the ductility level.

R/: Thank you for noting this omission. We fixed this in the updated manuscript.

For comments 11, 12 and 13, we used the GEM taxonomy with four attributes: the main construction material type, lateral load-resisting system, the expected level of ductility, and the number of stories for all the building classes as shown in Table 4 of the manuscript. For some of the unreinforced masonry the material technology was used as information was available. For the ductility we only used two options: Ductile (DUC) and non-ductile (DNO).

**3.2. Exposure model for the residential building stock**

**Table 4.** Summary of the building typologies in the exposure model defined for the study area. The building classes are defined based on the GEM v.2.0.

| Building class | Description | Number of buildings | Proportion (%) | Replacement cost (M. USD) |
|---|---|---|---|---|
| CR/LDUAL/DUC/H:4,7 | Ductile reinforced concrete dual frame-wall system, 4 to 7 stories | 14 | 0.02 | 34.94 |
| CR/LFINF/DUC/H:1 | Non-Ductile reinforced concrete infilled frames, 1, 2 and 3 stories | 855 | 1.13 | 557.05 |
| CR/LFINF/DUC/H:2 | | 1207 | 1.59 | 786.42 |
| CR/LFINF/DUC/H:3 | | 453 | 0.60 | 294.91 |
| CR/LFINF/DUC/H:4,7 | Ductile reinforced concrete infilled frames, 4 to 7 stories | 464 | 0.61 | 1131.06 |
| CR/LFM/DNO/H:1 | Non-Ductile reinforced concrete moment frames, 1, 2 and 3 stories | 855 | 1.13 | 557.05 |
| CR/LFM/DNO/H:2 | | 1207 | 1.59 | 786.42 |
| CR/LFM/DNO/H:3 | | 453 | 0.60 | 294.91 |
| CR/LFM/DUC/H:4,7 | Ductile reinforced concrete moment frames, 4 to 7 stories | 464 | 0.61 | 1131.06 |
| CR/LWAL/DUC/H:4,7 | Ductile reinforced concrete walls, 4 to 7 stories | 292 | 0.39 | 715.24 |
| CR/LWAL/DUC/H:1 | Ductile reinforced concrete walls, 1, 2 and 3 stories | 163 | 0.22 | 170.02 |
| CR/LWAL/DUC/H:2 | | 230 | 0.30 | 240.03 |
| CR/LWAL/DUC/H:3 | | 86 | 0.11 | 90.01 |
| MCF/LWAL/DNO/H:1 | Non-Ductile confined masonry, 1, 2 and 3 stories | 6154 | 8.12 | 1388.94 |
| MCF/LWAL/DNO/H:2 | | 7055 | 9.31 | 1713.00 |
| MCF/LWAL/DNO/H:3 | | 2646 | 3.49 | 642.37 |
| MCF/LWAL/DUC/H:1 | Ductile confined masonry walls, 1, 2 and 3 stories | 1230 | 1.62 | 796.62 |

| | | | | |
|---|---|---|---|---|
| MCF/LWAL/DUC/H:2 | | 1736 | 2.29 | 1124.65 |
| MCF/LWAL/DUC/H:3 | | 651 | 0.86 | 421.74 |
| MR/LWAL/DUC/H:1 | | 473 | 0.62 | 384.56 |
| MR/LWAL/DUC/H:2 | Ductile reinforced masonry walls, 1, 2 and 3 stories | 668 | 0.88 | 542.91 |
| MR/LWAL/DUC/H:3 | | 251 | 0.33 | 203.59 |
| MUR/LWAL/DNO/H:1 | | 14738 | 19.45 | 3516.93 |
| MUR/LWAL/DNO/H:2 | Non-Ductile unreinforced masonry walls, 1, 2 and 3 stories | 19682 | 25.97 | 4753.33 |
| MUR/LWAL/DNO/H:3 | | 7384 | 9.74 | 1783.33 |
| MUR-ADO/LWAL/DNO/H:1 | Non-Ductile unreinforced masonry with adobe blocks walls, 1 and 2 stories | 231 | 0.30 | 48.17 |
| MUR-ADO/LWAL/DNO/H:2 | | 254 | 0.34 | 54.71 |
| MUR-STDRE/LWAL/DNO/H:1 | Non-Ductile unreinforced masonry with dressed stone walls, 1 and 2 stories | 675 | 0.89 | 134.67 |
| MUR-STDRE/LWAL/DNO/H:2 | | 941 | 1.24 | 187.74 |
| MUR-STRUB/LWAL/DNO/H:1 | Non-Ductile Unreinforced masonry with semi-Dressed stone, 1 and 2 stories | 210 | 0.28 | 40.19 |
| MUR-STRUB/LWAL/DNO/H:2 | | 292 | 0.39 | 56.02 |
| W/WLI/DUC/H:1 | Ductile light wood members, 1 and 2 stories | 1515 | 2.00 | 357.22 |
| W/WLI/DUC/H:2 | | 2246 | 2.96 | 554.13 |

14. It is not clear why the authors present mean values for the 18 seismic events. As each scenario has a different epicenter and different consequences mean values are not representative (results for each scenario should be presented by themselves). See Table 19 and Figures 10 and 13.

R/: We appreciate this comment. In the light of the comment, we decided to remove table 9 and instead include a table that shows the percentage of buildings for each damage state in the eighteen scenarios. We also added a Figure that illustrates the variability of these damage estimates. We also present the economic losses for each scenario in Table 14 of the manuscript.

One point to note here is that Sabana Centro has an uneven distribution of the building stock, therefore, similar earthquakes with the same magnitude and similar depth such as the Mw 6.25 Sopó, Mw 6.25 Tabio and Mw 6.25 Nemocón have different consequences in terms of damage and economic losses.

The following was included in section 4.1

**4.1. Damage forecast**

[revised manuscript text omitted]

15. The sentence of line 410 "One out of four buildings will experience extensive damage or collapse" is a strong conclusion that requires a big certainty to be written. I suggest the authors to revise the fragility curves of the masonry buildings (as most of the buildings are of this typology) and to compare the ground motion fields with the building damages to be sure that results are correct. Furthermore, as all the buildings form this typology are one-story buildings results should not be as bad as shown in the article.

R/: Based on this comment and the other reviewers concerns about not discretizing between the height for buildings in the range of one to three stories, we decided to update the exposure model using information of the 2018 national census, with further differentiation between building heights. In addition, we used the fragility functions calculated by Martins and Silva. (2021), which all use the same modeling approach, and they account for the differences in height. The results now show that now the total of severely damaged and collapsed buildings is 14% and thirty percent of these collapses comes from two story non ductile unreinforced masonry buildings.

**5. Discussion**

**5.1. Damage and losses**

'(…) Worryingly, 14.3% buildings would experience extensive damage or collapse (…)'

16. Line 440. The authors mention "the damage and losses estimates presented in this study should be considered as lower bound". See the previous comments.

R/: The statement refers to the fact that existing fragility functions do not account for the informal construction practices which are common in the Sabana Centro region.

17. Figure 4. Add a color scale. It is difficult to read the percentage associate to each bin.

R/: We added a color scale to the figure

18. Figure 9. It is suggested to include the earthquake epicenter as well as a figure with the ground motion field generated by the event.

R/: We added a figure with the epicenter and the ground motion field.

**4.1. Damage forecast**

'(…) The respective distribution of  ground motion field for the 5.95 earthquake is presented in Figure 11.

[Figure]

**Figure 11.** Estimated maximum peak ground acceleration in bedrock for the Mw 5.95 Chia earthquake. The highest acceleration is presented in dark red and the lowest in green.

19. Figure 11. Expected losses including SVI should be greater than the expected losses without SVI. This is not shown at Chía and Sopó. For the ease of understanding it is suggested to use the same color scale in both maps of the figure.

R/: We updated the figure with the same color scales.

20. Figure 12. It is suggested to include the uncertainty in the figure.

R/: We did not include uncertainty in this figure because it became cluttered. We did include a table in the paper with the losses for each scenario. Table 10  presented in comment 14.

21. Table 2. Add the distance for the epicenter to the study area. Complement the information with a figure in which the epicenters are shown. As the events have an associated municipality, is the epicenter located at each municipality? How feasible is this? Results indicate important consequences that can be misinterpreted if the article does not mention the possibility that such events occur with epicenters in each municipality.

R/: Based on your suggestion and other reviewers' comments, we complemented table 2 with a figure that includes the location of the epicenters. We also clarified the statement about the epicenter location.

**3.1.1. Definition of the earthquake scenarios**

'(…) Based on this disaggregation, eighteen crustal events were selected from the probabilistic seismic hazard catalogue to be used in this study to calculate the expected damages and economic losses. The magnitude, location and geometry of ruptures are shown in Table 2 . The epicenter of each event is located within the municipality mentioned in the first column of the Table 2 and as shown in Figure 5.

[Figure]

Figure 5. Location of the 18 selected events, the color range varies according to the magnitude of the timing of the events. The gray region represents the zone where shallow events have the largest contribution.

22. Line 179. How does the "significant number of low-rise stiff buildings" relate to the selected crustal events?

R/: Thank you for this comment. We realized that the sentences are clearer if we limit the discussion to the disaggregation (Figure 4 in the paper), which shows that the highest contribution to the PGA comes from crustal events with magnitudes that range between 5.0 to 6.5 and distances smaller than 30km. We updated the paper removing the sentence quoted by the reviewer.

**References**

Martins, L. and Silva, V.: Development of a fragility and vulnerability model for global seismic risk analyses, Bulletin of Earthquake Engineering, 19, 6719–6745, https://doi.org/10.1007/s10518-020-00885-1, 2021.

Villar-Vega, M., Silva, V., Crowley, H., Yepes, C., Tarque, N., Acevedo, A. B., Hube, M. A., Gustavo, C. D., and Santa María, H.: Development of a fragility model for the residential building stock in South America, Earthquake Spectra, 33, 581–604, https://doi.org/10.1193/010716eqs005m, 2017.

Yepes-Estrada, C, Silva, V., Valcárcel, J., Acevedo, A. B., Tarque, N., Hube, M. A., Coronel, G., and Santa María, H.: Modeling the Residential Building Inventory in South America for Seismic Risk Assessment, Earthquake Spectra, 33, 299–322, https://doi.org/10.1193/101915eqs155dp, 2017.

**Reviewer 2: Pablo Heresi**

This article presents seismic risk results from 18 earthquake scenarios in the Sabana Centro province, an intermediate hazard zone in Colombia. The 18 scenario events were chosen based on hazard disaggregation results on a site within the region of interest, for 475 years return period hazard level. The epicenters of the scenario events were located within the region. The exposure model was gathered from previous studies, complemented with census data and remote surveys. Finally, the set of fragility curves for the considered typologies was gathered from previous studies. Results show that, on average, the occurrence of one of the scenario earthquakes might result in about $800 million USD of economic losses (about $1000 million USD when adjusted by social vulnerability) and about 20% of all the buildings collapsing.

The study is well performed and written. In terms of scientific merit, although the article does not provide new methods for seismic risk assessment, it provides novel results on the seismic risk faced by the Sabana Centro province. I have some comments that may help to improve the quality of the article (I have marked the most important issues that I comment).

In particular, compared to the damage produced by previous earthquakes around the World, the numbers presented in this article seem to be too high, especially considering that: (1) these are mean values, not low-probability values (thus they are not even the "worst-case scenario"); and (2) the authors state that the risk results should be considered as a lower bound. Therefore, I strongly suggest a careful revision and discussion of the results.

**Dear Dr., Heresi**

You kindly spent time delving into our manuscript, and we are grateful. Thank you for your appreciation of the study's performance and writing style and acknowledging the novelty of the results on the seismic risk assessment in the Sabana Centro province since the study performed by the Japan International Cooperation Agency (JICA) in 2001 has been neither discussed nor updated. Thank you for your comments. We will do our best to provide a complete answer to all of them.

We have used a color code to answer your valuable questions, comments, and suggestion. Please find them already answered in grey and the respective answers in black. The corresponding paragraph in the paper with changes is in blue

***Specific Comments (Individual scientific questions/issues)***

1. Line 130: The authors state that the Bucaramanga's seismic nest earthquakes occur at depths between 140 and 200km. However, in Figure 3 it is possible to observe several events at depths between 70 and 150km. I suggest the authors comment something about this inconsistency.

R/: Thank you for your comment. The statement on line 130 corresponds to the general definition of the Bucaramanga seismic nest. In Figure 3 the authors show the complete catalog without source distribution. That is, the events that are the origin of the Bucaramanga seismic nest are considered as those cortical events, which can occur over the seismic nest. To avoid this misunderstanding, the following sentence will be included in the revised version of the article to clarify the information presented in the figure:

**3.1. Seismic hazard**

'(…) Earthquakes in the Benioff area correspond to earthquakes inside the plate, which is subducting towards the east from the Colombian Pacific towards the country's interior. The Bucaramanga's seismic nest corresponds to an area where earthquakes with moment magnitudes between Mw 4.0 and 5.0 usually occur at depths between 140 and 200 km (Prieto et al., 2012). (…)'

**3.1.1. Definition of the earthquake scenarios**

'(…) In this study, earthquake events are defined in terms of the magnitude, location, and geometric characteristics of their ruptures. For the determination of the magnitude and location of the events to be considered in the estimation of damage, events from the unified earthquake catalogue developed by the SGC (SGC, 2021b) within a radius of 200 km were considered. Figure 3 shows the events of the complete catalog, considering those from the seismic nest, as well as those events from a cortical environment. (..)'

2. Line 144: The description of Figure 3 does not match what is observed in the figure. The authors state that Figure 3 shows: (1) close (distance < 75km) and shallow (depth < 70km) events; and (2) far (150km < distance < 200km) and deep (depth > 70km) events with magnitudes greater than 6.5. However, the figure presents many other events. For example, there are shallow events at distances larger than 75km with magnitudes lower than 6.5, deep events with magnitudes lower than 6.5, etc. The description and the figure should be consistent.

R/: Thank you for your suggestion. To be consistent between the figure and the description we have improved the discussion of the figure by describing the events according to the distance from the disaggregation, as well as other information like the magnitude and depth.

**3.1.1. Definition of the earthquake scenarios**

'(…)Figure 3 shows the events of the complete catalog, considering those from the seismic nest, as well as those events from a cortical environment. The figure shows events at distances less than 50 km from the center of Tenjo near the surface with depths less than 70 km and moment magnitudes ranging from Mw 4.0 to 5.5. There are also some events at depths between 70 km and 300 km at distances between 50 km and 100 km. These events range between a moment magnitude of Mw 4.0 and 6.5. It should be noted that most of the events in this area are shallow. Some far events at distances greater than 100 km are shallow events that can range between magnitudes of Mw 4.0 and Mw 7.0. It is also noted that there are deep events that can reach a moment magnitude of Mw 7.0. (…)'

3. Line 155: The authors state that the population centroid is located within Tenjo. Moreover, they use this municipality throughout the article as a reference (e.g., Figures 2 and 3). However, given the map provided in Figure 6c and the population of each municipality provided in Table 1, it seems like the centroid should be located somewhere between Zipaquirá and Chía (probably within Cajica), which are the two municipalities with the largest populations.

R/: We agree with you. The interpretation of the sentence is that the disaggregation was done for the population center of the entire study region; however, what was meant is that this procedure was done for the population center of Tenjo. This municipality was taken as a reference point at the authors' discretion. We rewrote the sentence so that it is clearer.

**3.1.1. Definition of the earthquake scenarios**

'(…) The disaggregation was developed for a point within the region of analysis, which corresponds to the population centroid of the municipality of Tenjo (longitude: -74.144, latitude: 4.872) (…)'

4. Figure 4: What is the distance type of the disaggregation? Rupture distance? Epicentral distance? Joyner-Boore distance? Given the depth of the deep sources, the difference between the distance types might be important.

R/: Thank you for your comment. For disaggregation, the authors used a distance to the projection of the rupture surface corresponding to the Joyner-Boore distance. To clarify both comments, we will include the following sentence in the revised manuscript:

**3.1.1. Definition of the earthquake scenarios**

The disaggregation was developed for a point within the region of analysis, which corresponds to the population centroid of the municipality of Tenjo (longitude: -74.144, latitude: 4.872) considering the Joyner-Boore distance to the projection of the rupture surface.

5. Table 2: The authors provide a list of the selected scenarios. A map showing these events might be useful for visualizing the epicenters with respect to the municipalities.

R/: We appreciate your recommendation. We added a new figure with this information in the manuscript.

**3.1.1. Definition of the earthquake scenarios**

'(…) Based on this disaggregation, eighteen crustal events were selected from the probabilistic seismic hazard catalogue to be used in this study to calculate the expected damages and economic losses. The magnitude, location and geometry of ruptures are shown in Table 2. The epicenter of each event is located within the municipality mentioned in the first column of the Table 2 and as shown in Figure 5.

[Figure]

Figure 5. Location of the 18 selected events, the color range varies according to the magnitude of the timing of the events. The gray region represents the zone where shallow events have the largest contribution.

6. Table 3 and its description: Although the logic tree was proposed by Arcila et al. (2020), I suggest the authors provide a justification for the different weights to the Cauzzi et al. (2014) and Abrahamson et al. (2014) ground-motion models.

R/: The logic tree proposed by Arcila et al. (2020) for shallow crustal regions has three GMPEs: Cauzzi et al. (2014), Abrahamson et al. (2014) and Idriss. (2014). The corresponding weights are 0.39, 0.211 and 0.399, respectively. The Sabana Centro region has a significant number of soft soils for which the Vs30 is lower than 450m/s. The Idriss (2014) GMPE is not defined for these types of soils. Consequently, for this study we decided to exclude this GMPE and use a logic tree that included the Cauzzi et al. (2014) and the Abrahamson et al. (2014) GMPEs, which do account for soft soils.

To update the weights of the Abrahamson and Cauzzi et al. (2014) GMPE in the new logic tree, we adjusted them distributing the weight of the Idriss (2014) GMPE (0.399) proportionally to their weights in the initial logic tree. We did so because this allowed us to keep the same level of relative weights between the two GMPEs used for our study. This led to the following calculations:

$$Cauzzi_{updated} = 0.39 + \frac{0.399}{0.39 + 0.211} = 0.65$$

$$Abrahamson_{updated} = 0.211 + \frac{0.399}{0.39 + 0.211} = 0.35$$

7. **Important comment** In Line 105, the authors state that "The majority of the building stock of the region is comprised of one- and two-story houses." However, as shown in Table 4, the considered building typologies include buildings with either 1 or 4 stories. The question is then, how are two-story houses classified into this system? This question is especially important because it has been previously demonstrated that one- and two-story houses present a significantly different seismic behavior and therefore levels of damage and losses (see, for instance, Heresi and Miranda 2022). In particular, classifying two-story houses as one-story structures may result in a significant underestimation of the seismic risk of these two-story structures.

Heresi, P., & Miranda, E. (2022). Evaluation of relative seismic performance between one-and two-story houses. Journal of Earthquake Engineering, 26(2), 857-886.

R/: Thank you for this important comment. Other reviewers also raised their attention to the need of updating the exposure model, further discretizing between building heights. We agree specially that the difference between one- and two-story houses is significant. Based on this comment and the other reviewers' suggestions, we decided to update the exposure model using information of the 2018 national census, with further differentiation between building heights. To

discretize the building height, we used information collected using remote and field surveys of 9000 houses in the Sabana Centro region (including the reported in the paper for Chía municipality), conducted by students of Universidad de La Sabana. We found that for houses between 1 and 3 stories, 34% are one-story houses, 48% are two-story houses and 18% are three-story houses. Accordingly, we updated our exposure model.

In addition, we decided to use the fragility functions calculated by Martins and Silva. (2021) and Villar-Vega et al. (2017), which all use the same modeling approach, and they account for the differences in height.

The contribution of one-, two- and three-story unreinforced masonry buildings is now stated in section 4.1. We also updated the discussion section, including the suggested reference and the above-mentioned facts.

**4.1. Damage forecast**

(….) The highest concentration of building collapses in the region is expected for houses constructed of unreinforced masonry (Table 9), mostly involving non-ductile unreinforced masonry walls of one and two stories (22.8% and 30.5% respectively). Notably, these two types of buildings account for 53.30% of collapses. Three-story unreinforced masonry houses account for 10.12% of buildings, making the overall contribution of this structural system more than six out of ten collapses. The percentage of three-story houses collapsed was smaller than the one from two story houses (which are less vulnerable) because three-story houses are less frequent in the region. (…)'

**5.1. Damage and losses**

(…) This situation stems from the fact that 83% of the houses in the province are constructed using non-ductile structural systems. These houses accounted for more than 90% of collapses in most of the scenarios. The damage results also highlight the importance of discretizing buildings with a same structural system by heights, at least for houses between one and three stories as suggested by Heresi and Miranda (2022) because two- and three-story houses had a significantly higher percentage of collapses compared to one story houses.

8. Table 5 presents the main parameters of the considered fragility curves. As stated by the authors, these fragility curves were selected from different studies after a thorough literature review. Although this is perfectly fine, it has an important drawback that should be commented on: the final set of fragility curves comprise curves developed with very different methods (e.g., analytical vs empirical) which have very different reliabilities (e.g., generally speaking, empirical fragility curves developed after earthquakes have higher uncertainties both in the probability of damage and in the ground-shaking intensity). The authors are encouraged to discuss about the limitations and reliability of the considered fragility curves, taking into account the methods, the data, and the assumptions used to develop them. They address some of these issues in the Caveats and Limitations section, specifically the issue of fragility curves not being developed directly for Colombian structures and not having a uniform description of the damage states, but there are other issues that are missing in this section, as those previously stated in my comment.

R/: Thank you for your comment. We agree with your point. Other reviewers also raised their attention to this fact. Therefore, for consistency in the revised version of the manuscript, we decided to use only analytical fragility functions. Most of them were those calculated by Martins and Silva. (2021) and Villar-Vega et al. (2017). These functions are analytical and all of them use the same modelling approach to avoid incorporating inconsistencies in the damage and losses estimations. We kept the comments about the issue that the fragility functions by Martins and Silva. (2021) were not developed accounting for the particularities of Colombian construction.

Accordingly, we included the following texts in different sections:

**Introduction**

' (…) Regarding the structural vulnerability of the building stock, a database of fragility functions developed for the residential building stock in South America Villar-Vega et al. (2017) and those developed for global seismic risk analysis (Martins and Silva, 2021). (…)'

**3.3 Physical vulnerability of residential building stock to seismic ground shaking**

' (…) The Physical Vulnerability Suite of the GEM Foundation (OpenQuake Platform - Vulnerability, 2021) was considered for the review. The GEM database for the specific case of Colombia has the curves developed by Acevedo et al. (2017) for unreinforced masonry houses constructed in Antioquia, Colombia. There are some curves for reinforced concrete buildings with geographical applicability in Manizales, Colombia by Bonett (2003) and the dataset of Villar-Vega. (2014) for South America. Although the set of curves covers different types of buildings, they are calculated based on different methodologies and different damage states.

Another available dataset of fragility curves are those developed by Martins and Silva (2021), who covered nearly 500 building classes at global level including Colombia. The fragility is calculated from nonlinear dynamic analyses performed on equivalent single-degree-of-freedom (SDOF) oscillators. They considered four damage states that are also intended to study in the present research: slight, moderate, extensive and collapse. The corresponding damage thresholds were defined based on the spectral displacement of the structures. At regional level also are the set of fragility curves for the residential building stock in South America (Villar-Vega et al., 2017), covering 54 common building classes. The methodology used for the derivation of the curves is similar to the one used in Martins and Silva. (2021).

Based on the information collected, the fragility curves available in Martins and Silva. (2021) were used mainly and complemented with those of Villar-Vega et al. (2017). These curves were selected in order to prevent an unbiased comparison of risk between the different municipalities in the region due to the different methodologies used to develop the fragility curves. Therefore, a set of 33 fragility functions was used to represent the probability of exceeding a level of damage conditioned to ground shaking intensity. These functions are comprised of 28 sets of curves reported by Martins and Silva. (2021) and five sets developed by Villar-Vega et al. (2021). The last one are assigned to Non-Ductile confined masonry, 1, 2 and 3 stories and Ductile light wood members, 1 and 2 stories, since in the former these building classes were not included.

9. *Important comment* Results show that a Mw5.95 event at Chía is expected to cause the collapse of more than 17% of the buildings in the region, and some level of damage in about half of the building portfolio. In particular, 6722 out of 14959 (about 45%) of houses made out of non-ductile unreinforced masonry with adobe block walls (1-story) are expected to collapse, according to the authors. Moreover, in Chía, more than 44% of the buildings are expected to collapse due to this Mw5.95 scenario. These numbers seem incredibly high for a Mw5.95 event at a first glance (even more when the authors state, in Line 441, that these estimates should be considered as a lower bound). Note that these are mean (i.e., expected) values, not low-probability values that might represent a somewhat "worst-case scenario" (or, in other words, somehow answer the question "how big may be the consequence if this earthquake occurs tomorrow?"). To put these numbers in perspective, we can compare them with the damage produced by the 2010 Haiti earthquake, Mw7.0:

- According to DesRoches et al. (2011), the 2010 Haiti earthquake damaged nearly half of the structures in the epicentral region.
- Eberhard et al. (2013) performed two field surveys of: (1) 107 structures in Port-au-Prince, where 30 (28%) of them collapsed and other 35 (33%) had enough damage to require repairs; and (2) 52 structures in Léogâne (closest population center to the epicenter), where 32 (62%) of them collapsed and other 16 (31%) had enough damage to require repairs.
- Rathje et al. (2011) performed a field survey of over 400 structures in Port-au-Prince. Of the 414 surveyed structures, 157 (38%) had significant damage (i.e., collapse or very heavy damage, EMS Grade 4).

Considering that the Haiti earthquake was not only 32 times larger in terms of magnitude, but also affected a more socially vulnerable country, it is expected that a Mw 5.95 event in the region of interest would result in considerably less damage and losses, especially if we talk about mean values.

In terms of losses, in Figure 12 we can observe that some of the earthquake scenarios have a 20% probability of producing more than 50% of the total replacement cost as economic losses (about 40% of the GDP of the region!). Considering that these curves were computed neglecting the spatial correlation of ground motion intensities (comment about this below), this probability for such a high loss is extremely large. For perspective, the 2010 Chile earthquake, Mw8.8, produced an economic loss of about 14% of the GDP of the country at the moment of the event.

The previous remarks highlight the importance of comparing risk results from scenario events with previous events to put the numbers in perspective. I suggest the authors include comparisons like the ones proposed above, but also include

other events, such as, for example, the 2020 Puerto Rico earthquake, Mw6.4. Moreover, in the Introduction, the authors mention two historical earthquakes that affected the region of interest, which may also be used to evaluate the reliability of the resulting damage produced by the considered scenario earthquakes. These comparisons would further support the risk results of the article.

DesRoches, R., Comerio, M., Eberhard, M., Mooney, W., & Rix, G. J. (2011). Overview of the 2010 Haiti earthquake. *Earthquake Spectra*, *27*(S1), S1-S21.

Eberhard, M. O., Baldridge, S., Marshall, J., Mooney, W., & Rix, G. J. (2010). The Mw 7.0 Haiti earthquake of January 12, 2010: USGS/EERI advance reconnaissance team report. *US Geological Survey Open-File Report*, *1048*(2013), 64.

Rathje, E. M., Bachhuber, J., Dulberg, R., Cox, B. R., Kottke, A., Wood, C., ... & Rix, G. (2011). Damage patterns in Port-au-Prince during the 2010 Haiti earthquake. *Earthquake Spectra*, *27*(S1), S117-S136.

R/: We highly appreciate this comment and acknowledge the importance of comparing with previous events to put the results in a broader perspective. In this regard, we deemed more appropriate to frame the discussion in terms of the 1999 Armenia earthquake in Colombia. This event had a Mw 6.1 magnitude at 15 km depth. Most of the building stock was comprised of URM buildings, built prior to the 1998 seismic design code of Colombia and like the buildings in Sabana Centro. This earthquake costed 1.6% of the national GDP (roughly five times higher than the earthquake considered in this paper). In terms of damage, the records indicate that 17551 were destroyed, 18421 had severe damage and 43474 had moderate damage. Another earthquake in Colombia was the Mw 5.5 in Popayán in 1983, which occurred at an estimated depth of 12 to 15 km. According to Colombian records, in this earthquake, 12% of buildings suffered complete damage and 34% experienced severe damage.

These two events put the numbers in perspective and support that the results of this paper are reasonable. We included the following text in section 4.1 of the revised manuscript:

**4.1. Damage forecast**

'(…) In the region, 42.95% of the buildings considered in the exposure model are expected to suffer some degree of damage. This result represents 32,598 out of the 75,778 analyzed buildings. Table shows that the type of damage with the highest occurrence is slight (21.31%), followed by collapse (9.85%); moderate (7.36%) and extensive damage (4.44%). Overall, 14.28% might suffer extensive or collapse damage, hence they will not fulfil their life safety functionality. Nearly ten percent of collapse rises concerns from a decision maker perspective, but two Colombian events put the results in perspective: the Mw 6.1 earthquake in Armenia (1999) and the Mw 5.5 earthquake in Popayán (1983). In the former, the records indicate that 17551 buildings were destroyed, 18421 had severe damage and 43474 had moderate damage. In the latter, which occurred at an estimated depth between 12 km and 15 km, 12% of buildings suffered complete damage. In both earthquakes, damage concentrated in unreinforced masonry buildings, constructed prior to the enactment of the Colombian seismic design code in 1998. More than 60% of the building stock in Sabana Centro is comprised of that type of buildings, and what is more, 35% are two- and three-story houses (Table 4), which are more vulnerable than those of one-story houses (Heresi and Miranda., 2022). These buildings are expected to withstand significant damage during an earthquake such as the Chía Mw 5.95 shown here, which is similar in magnitude and depth to the Armenia earthquake and for which the percentage of collapse herein presented is similar to that from the Popayán earthquake. (…)'

Regarding the reviewer comment about the lower bound, the authors have this concern because the field surveys have shown that a significant number of buildings in Sabana Centro are informally constructed, raising the possibility that none of the existing fragility functions can represent this type of construction properly. We address this more throughout in this discussion section:

**5.3. Caveats and limitations**

'(…) One concern held by the authors is that the field observations and surveys show that a good part of the building stock of Sabana Centro is the result of informal construction. Presently, these buildings are constructed using either confined masonry or infilled RC frames due to the influence of the Colombian design code (which holds similarities

with the ACI-318 (Arroyo et al., 2019)), which forbade unreinforced masonry. Research about fragility functions like these buildings in Puerto Rico (Murray et al., 2022) and Villavicencio (Feliciano et al. 2022) show that the collapse probability may be even twice than that of code conforming buildings. The fragility functions by Martins and Silva. (2021) and Villar-Vega. (2017) used in this research do not account for the particularities of these buildings, thus the authors hold the hypothesis that the damage estimates should be considered as a lower bound. (…)'

10. Table 10 presents the resulting SVI for the 11 municipalities of the region. Although the authors previously explain the variables involved in this index (Table 6), I have two comments about this:

I suggest the authors provide more detailed information about how the index of each category is obtained. This explanation would improve the reproducibility of the reported results.

Thank you very much for this suggestion; the selection of the indicators that are part of the composite indicators: population, economy, infrastructure, education, health, and the variables considered for the indicator COVID-19 was already explained in section 3.4. The composite indicators, single indicators and variables considered initially to estimate the SV were listed in Table 5, in the submitted version. However, to provide more detailed information about how the SV index is constructed, we elaborate more on the explanation of the methodology. Please find below the section already revised.

**3.4. Social Vulnerability (SV)**

To determine the level of social vulnerability (SV) of the municipalities of the Sabana Centro province, this paper estimated a social vulnerability index (SVI) based on the methodology proposed by Cutter et al. (2003). The social equivalent to a quantitative physical risk assessment for earthquakes is an SVI. Social vulnerability is the reason for the different experiences of communities regarding the consequence of earthquakes (Burton, C. G., & Silva, V., 2016). The construction of composite indicators based on the mathematical combination of a set of indicators, which consists of a group of variables, is one of the most common methods to objectively assess SV (Freudenberg, 2003). There are several methodological approaches for the construction of composite indicators, but in general, the steps include: (1) the identification of pertinent variables, (2) the aggregation of variables into indicators and composite indicators (3) multivariate analysis (4) weighting (5) convolution or link of variables and (6) visualization and dissemination of results (Burton, C. G., and Silva, V., 2016).
The SVI index aims to identify those municipalities in Sabana Centro whose inhabitants are more vulnerable to an earthquake based on a selection of specific variables, indicators, and composite indicators. The indicators were aggregated into five composite indicators constructed for the SVI of SARA project[2]: population, economy, infrastructure, education, and health. The composite indicator of population considers the indicators that capture the capacity of population to mitigate their risk and recover from earthquakes. In the current research the composite indicator of population accounted initially for the female and native indigenous population, age dependence, population density, number of households and people per household. The composite indicator of economy includes indicators to assess the economic health of the community (Burton, C. G., and Silva, V., 2016) . The single indicators considered for this composite indicator were population unemployed, looking for employment, unsatisfied basic needs (UBN), and impoverished. Poverty is an important aspect to consider because of its direct association with access to resources, which affects coping with the impacts of disasters (Fatemi et al., 2017). The composite indicator of infrastructure considers the access to basic services (Contreras et al., 2020b). The composite indicator of education links the educational level and the socioeconomic status, mitigation, and recovery potential (Burton, C. G., & Silva, V., 2016). It is assumed that lower education level results in lower-income, poor ability to understand emergencies, and low capacity to recover (Cutter et al., 2003). The composite indicator of health includes the indicators related to access to health facilities and health care (Contreras et al., 2020b). The lack of access to healthcare increases people's susceptibility to the potential impact of disasters (Fatemi et al., 2017).

There are many variables used for the SVI that are strongly correlated. For example, in the "Population" category, there are 7 variables, where, for instance, "Female population" and "Total population" are expected to be strongly correlated, unless the percentage of women varies significantly from one municipality to another for some reason. As the authors did not provide too much detail on how the index is computed, I'm not sure if they tested for collinearity between these

[2] https://sara.openquake.org/development_of_indicators_of_social_vulnerability

variables, for example. We can even expect some correlation between different categories. For instance, municipalities with a high index in Economy will probably have also a high index in Infrastructure. These correlations might result in biased SVI's when all the variables are considered.

R/: Thank you very much for this comment. The reviewer is right indicators such as female population and total population are strongly correlated (1.000**). Then, to avoid problems interpreting the model and overfitting, we checked the multicollinearity by looking at the variance inflation factor (VIF) of each variable and indicator. The VIF was identified in a linear regression that included collinearity diagnostics produced in SPSS (Field, 2005). The results are presented on Tables 5, 6 and 7 in the manuscript.

In Table 5, values greater than five and highlighted in red indicate a potentially severe correlation between a given predictor variable and other variables. Table 6 contains excluded variables that do not add significant information to the model. Table 7 contains the selected no collinear predictors of SV.

**3.4. Social Vulnerability (SV)**

[revised manuscript text omitted]

Field, A. (2005). Discovering statistics using SPSS. Chenai: Sage publications.

Freudenberg, M., 2003. Composite Indicators of Country Performance: A Critical Assessment, Organisationm for Economic Co-operation and Development (OECD), Paris.

12. *Important comment* As one of the limitations, the authors state that they did not consider the spatial cross-correlation when modelling the ground motion fields. However, they do not justify this arbitrary exclusion. For example, the OQ-Engine has models of spatial correlation already implemented, and therefore I do not see a good reason for neglecting it. As the authors correctly state, the inclusion of a spatial correlation model would increase the dispersion of the curves presented in Figure 12, making them more "realistic". Thus, I suggest either including a spatial correlation model, or giving a strong justification for its arbitrary exclusion.

R/: The reviewer expresses some concerns about having disregarded a spatial correlation model to model the ground motion fields. It is worth noting that we always consider the ground motion variability through uncorrelated random fields (that allowed us to create Figure 12). It is also worth noting that the sentence in line 451 did not refer to simple spatial correlation models, but rather to spatial inter-period cross-correlation models (IPCCM), i.e., when several intensity measures (IM) are simultaneously required by their set of fragility functions to calculate the physical vulnerability of building stocks to earthquakes. Related to this, it is important to note that the current OpenQuake engine only provides the option to simulate spatially correlated ground motion fields (e.g., Jayaram and Baker, 2009) which was used in our analyses, but it does not provide spatially cross-correlated random fields.

Moreover, it is important to highlight the relation between the spatial extent and density of the building stocks with respect to the decision of including or not IPCCM. For instance, the study by Michel et al. (2017) found that for building portfolios that are spaced a few kilometers apart, the influence of cross-correlation in risk assessment is very small compared to the one imposed by ground motion variability itself. This feature is similar to the one we encounter in the Sabana Centro region where the main urban areas (cascos urbanos) between neighboring municipalities are separated by several kilometers. Conversely, other studies have found that the role of including either simpler spatial correlation models (e.g., Bazzurro and Luco, 2005) or IPCCM (Gomez Zapata et al, 2022a,b) for a dense and spatially aggregated building portfolio is comparatively more relevant, which is not to be the case of our study area. In fact, the aforementioned cited study remarks on this issue by stating that, since the spatial correlation of ground motion IMs decreases rapidly with distance (e.g. Schiappapietra and Douglas, 2020), its effect on loss-estimations is maximized when it is applied to a dense exposure model (i.e. with aggregation areas (~1 x 1km grid) significantly smaller than the correlation distance of the ground motions (~20 km) because buildings within a grid cell are treated as if the inter-station distance was zero. Since our exposure model in Sabana Centro is composed of only 11 geocells where the buildings are therein spatially aggregated and with centroid-to centroid distances of the same order as the ground motion correlation lengths, we can expect that the relevance of including a spatial correlation model would not be high. Of course, this feature is inherent to the decision of the aggregation areas (11 municipalities).

Aligned with the former, and as described by Stafford. (2012) and by Gomez Zapata. (2021), one can suspect that when the dimension of the geo-cells in the exposure model is larger than a typical seismic ground motion correlation length, an artificial bias in the ground motion correlation has to be expected which may be the case in our study. Therefore, more meaningful future studies with higher resolution exposure models are anyway required but are outside our scope. These possible future improvements along with local fragility models (perhaps with more IM) will certainly require the incorporation of spatially correlated or cross-correlated models for which we can then confirm the important relationship and similarity between the correlation of ground motions and the damage correlation of exposed structures as comprehensively presented by Heresi and Miranda. (2022).

***Technical corrections***

13. Line 43: Change "7248 injured" for "7248 injured people" or "7248 injuries".

'(…) In the first case, the earthquake caused 287 deaths, 7248 injuries, and 150 thousand people affected (Cardona et al., 2004; Lomnitz and Hashizume, 1985). (…)'

14. There is an inconsistency in the use of thousand separators. For example, in Line 45 the authors state "… and 35000 buildings that collapsed…", but then in Line 86, they write "resulted in 200,000 deaths". In Table 1, the authors use thousand separators again.

'(…) In the second case, this event left 1185 casualties, 8523 injured people (Naciones Unidas | CEPAL, 1999), and 35,000 buildings that collapsed or experienced severe damage (Chávez et al., 2021). (…)'

15. Line 118: Review the word "gro".

Among the municipalities, Chía, Cajicá, and Zipaquirá had the highest population growth, accounting for 64% of the total population of the region.

16. Line 158: The authors use the Quetame earthquake for defining the rupture geometry of the scenario events. I suggest adding an annotation in Figure 3, showing which one is the Quetame earthquake, for those of us who are not familiar with the historic seismicity of Colombia.

R/: We added a point inside the figure with the date and magnitude of the Quetame earthquake.

[Figure]

**Figure 3.** Geographic distribution of events occurring within 200 km of the study area selected from the Unified Earthquake Catalogue of the SGC (SGC, 2021b). The size of the circle represents the magnitude of the event, and color indicates depth. The event marked with the black circle corresponds to the focal mechanism of the Quetame earthquake of magnitude Mw 5.9.

17. Line 249: There is an incomplete phrase.

R/: We removed the word "with".

R/: Thank you for taking the time to make technical corrections, we have addressed all of them in the manuscript.


**Reviewer 3: Maria Camila Hoyos**

The study deals with the analyses of various seismic risk scenarios for the Sabana Centro region in Colombia, located in the northern region of the capital city Bogotá, which concentrates important industrial facilities, educational facilities. It is an interesting study that follows the state-of-the-art procedures of scenario risk analyses (at least up to the computation of direct losses), for a region that hasn't been studied before, and thus is a good contribution to the scientific literature that communicates the seismic risk in the country.

Having said this, there are some comments about important issues in the study, that hopefully will help improve its clarity, coherence, and thoroughness. It must be said that after doing the review, the reviewer saw that many of the comments and limitations of the study were included in the discussion section as further developments, however there are many that should be included to make the results sound and representative of the region of study, otherwise many of the presented results could be very misleading.

**Dear reviewer**

You kindly spent time delving into our manuscript, and we are grateful. Thank you for acknowledging the value of our study. We will do our best to provide a complete explanation of the limitation of the study to make the results sound and representative, as you request in your comments.

We have used a color code to answer your valuable questions, comments, and suggestion. Please find them already answered in grey and the respective answers in black. The corresponding paragraph in the paper with changes is in blue

- **Section 2**

1. It would be interesting to include why the Sabana Centro region is of particular interest. In previous studies of the major cities, generated GDP or % of the population in comparison with the whole country were presented as reasons for the study of a particular city or region.

R/: Thank you for your comment. This is an important region in the department of Cundinamarca and for the authors is an interesting area for its economic and social growth rate in recent years. We state this in the manuscript, but to improve the statement, we added the following sentence to the updated manuscript:

**2. Description of the study area**

'(…) Sabana Centro is a region of Cundinamarca, Colombia, to the north of Bogotá, the country's capital. Cundinamarca is one of the four most populated regions of the country and Sabana Centro is one of the provinces that contributes the most population (18%) and the department GDP (32%). (…)'

2. Additionally, it is mentioned that it concentrates many economic and industrial activities, but at the end the analysis only deals with the residential building stock.

R/: The previous sentence was mentioned to highlight the fact about the province and its contribution within the region. As the reviewer says, we only deal with buildings for residential use. This is the predominant use in the study area. In addition, there is not enough information in the region to characterize commercial and industrial buildings.

3. Line 105: "The majority of the building stock of the region is comprised of one- and two-story houses" It would be good to show a reference with the numbers based on the 2018 Census for this. It is interesting that this is mentioned and still no two-story houses are considered in the analyses.

R/: Unfortunately, there is no information in the census related to the building height and the information in the national cadastral database does not include Sabana Centro. The phrase was based on the authors' fieldwork observations in the region based on 9000 remote and field surveys.

- **Section 3.1 Seismic hazard**

4. Line 171: "eighteen crustal events were selected from the catalogue to be used in this study" Does this mean that only 'historical' events included in the catalogue were included? No new possible events from the event-based tables from the PSHA model?

R/: We appreciate your comment because our sentence was not clear. We indeed used eighteen events from the PSHA model we developed. Aware that our explanation is unclear, we updated it in the revised manuscript to clearly state our procedure.

**3.1.1. Definition of the earthquake scenarios**

'(…) Based on this disaggregation, eighteen crustal events were selected from the probabilistic seismic hazard catalogue to be used in this study to calculate the expected damages and economic losses (…)'

5. Table 2: Include column with the distance to the population centroid taken as reference point in the disaggregation to be able to compare this selection of scenarios with the disaggregation graphs presented in Figure 4 and their

representativeness in the overall 475-years return period hazard in the region. Given the proximity of the events based on the disaggregation, directivity effects should be considered for some of these events. As further seen in the discussion this was not considered but it would be good to mention it and not leaving it till the discussion as a further development.

R/: We appreciate this comment. Based on your suggestion and other reviewer's comments, we included a figure which complemented Table 2, allowing to visualize the location of the events. The information about faults within Sabana Centro is limited, thus we chose not to consider the directivity in these events. We included a sentence to clarify this in section 4.

[Figure]

 Figure 5. Location of the 18 selected events, the color range varies according to the magnitude of the timing of the events. The gray region represents the zone where shallow events have the largest contribution.

**4. Results**

(….) These eighteen scenarios are defined based on the earthquake events presented in Table 2. These did not include directivity effects because there was insufficient information available for a reliable model (…)

- **Section 3.2 Exposure model for the residential building stock**

6. Was the replacement cost updated to 2021? 2022? In which way was this done if indeed it has been updated? If not, it should be done and explained. Were the new inflated exposure building numbers (based on population as proxy) in any way compared to the dwellings or building numbers reported in the 2018 Census for these regions?

R/: Thank you for your questions and recommendations. The replacement cost was not updated in the original manuscript. Based on your comment, we decided to update the replacement costs using cadastral information available in the Territorial Statistical Systems (TerriData) of the country. We compared the number of dwellings with those reported by the 2018 census and the difference is 2.8%. This difference is mainly because the census considered dwellings whose wall material is poured concrete and we did not include it since it was not possible to assign a percentage of construction. Clarifications in this regard are included in the manuscript and we decided to update the exposure model and use the information of the dwellings in the 2018 census instead of the one based on population. Nonetheless, we think if worth highlighting that in the absence of census information, inferring based on population can provide a result with a moderate degree of approximation.

**Introduction**

'(…) Information available from the national census was used to create the exposure model. The methodology used in (Yepes-Estrada et al., 2017)  was followed to assign the number of buildings per municipality.  (…)'

**3.2. Exposure model for the residential building stock**

' (…) The building exposure model for the region has information about the building classes, the number of buildings, inhabitants, and the buildings' replacement costs. To develop this model for Sabana Centro, the methodology used by the South America Risk Assessment (SARA) project to develop exposure models in South America (Yepes-Estrada et al., 2017) was taken as a basis. The source of information to assign the number of buildings was the 2018 national census (DANE, 2018). The census allowed having information on the number of dwellings and typical wall and roof materials, which were used to infer the different classes of buildings by municipality. A total of 156,628 dwellings were calculated; this number differs from that reported by the national census by 2.8%, since it did not consider dwellings whose wall material is poured concrete. This material was not included since there was no information available to relate it to any type of building class. The set of dwellings were related to the same building classes and same relationships ('mapping schemes') used in Yepes-Estrada et al. (2017). As the census information is reported in terms of dwellings the procedure used in Yepes-Estrada et al. (2017) to calculate the number of buildings was also followed. Then, this data was complemented using information collected during remote surveys carried out by students from the Universidad de La Sabana in the municipality of Chía. The building replacement cost refers to the cost of structural and non-structural components of a building and it is a value associated with the building's rehabilitation. This study has only considered the structural cost per building calculated based on cadastral information available in the Territorial Statistical Systems (TerriData[3]) of the country. This replacement cost was computer per building, expressed in USD. As the currency in Colombia is in Colombian pesos, the exchange to U.S. dollars was made for an average exchange rate of 4080 pesos. (..)'

7. Table 4 only considers unreinforced masonry of 1 storey, which is known to be less vulnerable than the unreinforced masonry of 2 stories, which is actually more common in many urban areas. This typology should be included (assuming something probably based on census data or the surveys), as in the region it is very common to find 2-storey, in some cases more than single storey houses (as previously mentioned in the study also). In the current version, the study may be underestimating the losses.

R/: Thank you for this important comment. Other reviewers also raised their attention to the need of updating the exposure model, further discretizing between building heights. We agree specially that the difference between one-and two-story houses is significant. Based on this comment and the other reviewers' suggestions, we decided to update the exposure model using information of the 2018 national census, with further differentiation between building heights. To discretize the building height, we used information collected using remote and field surveys for Chía presented in the article, conducted by students of Universidad de La Sabana. We found that for houses between 1 and 3 stories, 34% are one-story houses, 48% are two-story houses and 18% are three-story houses. Accordingly, we updated our exposure model, as mentioned in the answered to your comment 6. Also, we added the following sentence and Table 4.

**3.2. Exposure model for the residential building stock**

'(…) For such a purpose, the method proposed in Pittore et al. (2018) was used to evaluate the level of compatibility between the observed building attributes and each predefined building typology. Details of this process can be consulted in Arroyo et al. (2022). This procedure served to compare the percentages of the building classes calculated based on the SARA methodology and classify the buildings by height. (…)'

Also, we have updated Table 4 as follows:

**Table 4.** Summary of the building typologies in the exposure model defined for the study area. The building classes are defined based on the GEM v.2.0.

| Building class | Description | Number of buildings | Proportion (%) | Replacement cost (M. USD) |
|---|---|---|---|---|
| CR/LDUAL/DUC/H:4,7 | Ductile reinforced concrete dual frame-wall system, 4 to 7 stories | 14 | 0.02 | 34.94 |
| CR/LFINF/DUC/H:1 | Non-Ductile reinforced concrete infilled frames, 1, 2 and 3 stories | 855 | 1.13 | 557.05 |
| CR/LFINF/DUC/H:2 | | 1207 | 1.59 | 786.42 |
| CR/LFINF/DUC/H:3 | | 453 | 0.60 | 294.91 |
| CR/LFINF/DUC/H:4,7 | Ductile reinforced concrete infilled frames, 4 to 7 stories | 464 | 0.61 | 1131.06 |
* * *
[3] https://terridata.dnp.gov.co/index-app.html#/

| | | | | |
|---|---|---|---|---|
| CR/LFM/DNO/H:1 | Non-Ductile reinforced concrete moment frames, 1, 2 and 3 stories | 855 | 1.13 | 557.05 |
| CR/LFM/DNO/H:2 | | 1207 | 1.59 | 786.42 |
| CR/LFM/DNO/H:3 | | 453 | 0.60 | 294.91 |
| CR/LFM/DUC/H:4,7 | Ductile reinforced concrete moment frames, 4 to 7 stories | 464 | 0.61 | 1131.06 |
| CR/LWAL/DUC/H:4,7 | Ductile reinforced concrete walls, 4 to 7 stories | 292 | 0.39 | 715.24 |
| CR/LWAL/DUC/H:1 | Ductile reinforced concrete walls, 1, 2 and 3 stories | 163 | 0.22 | 170.02 |
| CR/LWAL/DUC/H:2 | | 230 | 0.30 | 240.03 |
| CR/LWAL/DUC/H:3 | | 86 | 0.11 | 90.01 |
| MCF/LWAL/DNO/H:1 | Non-Ductile confined masonry, 1, 2 and 3 stories | 6154 | 8.12 | 1388.94 |
| MCF/LWAL/DNO/H:2 | | 7055 | 9.31 | 1713.00 |
| MCF/LWAL/DNO/H:3 | | 2646 | 3.49 | 642.37 |
| MCF/LWAL/DUC/H:1 | Ductile confined masonry walls, 1, 2 and 3 stories | 1230 | 1.62 | 796.62 |
| MCF/LWAL/DUC/H:2 | | 1736 | 2.29 | 1124.65 |
| MCF/LWAL/DUC/H:3 | | 651 | 0.86 | 421.74 |
| MR/LWAL/DUC/H:1 | Ductile reinforced masonry walls, 1, 2 and 3 stories | 473 | 0.62 | 384.56 |
| MR/LWAL/DUC/H:2 | | 668 | 0.88 | 542.91 |
| MR/LWAL/DUC/H:3 | | 251 | 0.33 | 203.59 |
| MUR/LWAL/DNO/H:1 | Non-Ductile unreinforced masonry walls, 1, 2 and 3 stories | 14738 | 19.45 | 3516.93 |
| MUR/LWAL/DNO/H:2 | | 19682 | 25.97 | 4753.33 |
| MUR/LWAL/DNO/H:3 | | 7384 | 9.74 | 1783.33 |
| MUR-ADO/LWAL/DNO/H:1 | Non-Ductile unreinforced masonry with adobe blocks walls, 1 and 2 stories | 231 | 0.30 | 48.17 |
| MUR-ADO/LWAL/DNO/H:2 | | 254 | 0.34 | 54.71 |
| MUR-STDRE/LWAL/DNO/H:1 | Non-Ductile unreinforced masonry with dressed stone walls, 1 and 2 stories | 675 | 0.89 | 134.67 |
| MUR-STDRE/LWAL/DNO/H:2 | | 941 | 1.24 | 187.74 |
| MUR-STRUB/LWAL/DNO/H:1 | Non-Ductile Unreinforced masonry with semi-Dressed stone, 1 and 2 stories | 210 | 0.28 | 40.19 |
| MUR-STRUB/LWAL/DNO/H:2 | | 292 | 0.39 | 56.02 |
| W/WLI/DUC/H:1 | Ductile light wood members, 1 and 2 stories | 1515 | 2.00 | 357.22 |
| W/WLI/DUC/H:2 | | 2246 | 2.96 | 554.13 |

- **Section 3.3 Physical vulnerability of residential building stock to seismic ground shaking**

8. Chilean wood structures are known to be in better shape than those in Colombia, and they consider a different type of construction technique. The same goes to the curves used in HAZUS, which are not as representative of the local conditions and may be underestimating the risk. If they are going to be used a more thorough explanation of the limitation of using them should be included.

R/: Regarding the selection of fragility functions, we acknowledge the reviewer is correct about the representativeness of fragility functions.

After updating the exposure model accounting for differences between one- and two-story houses, for consistency in the revised version of the manuscript, we decided to use only analytical fragility functions. Most of them were those calculated by Martins and Silva. (2021) and Villar-Vega et al. (2017). These functions are analytical and all of them use the same modelling approach.

**Introduction**

'(…) Regarding the structural vulnerability of the building stock, a database of fragility functions developed for the residential building stock in South America Villar-Vega et al. (2017) and those developed for global seismic risk analysis (Martins and Silva, 2021). (…)'

**3.3 Physical vulnerability of residential building stock to seismic ground shaking**

'(…)The Physical Vulnerability Suite of the GEM Foundation (OpenQuake Platform - Vulnerability, 2021) was considered for the review. The GEM database for the specific case of Colombia has the curves developed by Acevedo et al. (2017) for unreinforced masonry houses constructed in Antioquia, Colombia. There are some curves for

reinforced concrete buildings with geographical applicability in Manizales, Colombia by Bonett (2003) and the dataset of Villar-Vega. (2014) for South America. Although the set of curves covers different types of buildings, they are calculated based on different methodologies and different damage states.

Another available dataset of fragility curves are those developed by Martins and Silva (2021), who covered nearly 500 building classes at global level including Colombia. The fragility is calculated from nonlinear dynamic analyses performed on equivalent single-degree-of-freedom (SDOF) oscillators. They considered four damage states that are also intended to study in the present research: slight, moderate, extensive and collapse. The corresponding damage thresholds were defined based on the spectral displacement of the structures. At regional level also are the set of fragility curves for the residential building stock in South America (Villar-Vega et al., 2017), covering 54 common building classes. The methodology used for the derivation of the curves is similar to the one used in Martins and Silva. (2021).

Based on the information collected, the fragility curves available in Martins and Silva. (2021) were used mainly and complemented with those of Villar-Vega et al. (2017). These curves were selected in order to prevent an unbiased comparison of risk between the different municipalities in the region due to the different methodologies used to develop the fragility curves. Therefore, a set of 33 fragility functions was used to represent the probability of exceeding a level of damage conditioned to ground shaking intensity. These functions are comprised of 28 sets of curves reported by Martins and Silva. (2021) and five sets developed by Villar-Vega et al. (2021). The last one are assigned to Non-Ductile confined masonry, 1, 2 and 3 stories and Ductile light wood members, 1 and 2 stories, since in the former these building classes were not included.

9. Table 5 needs a clarification of what each curve considers in each of the damage states. If the vulnerability model is considering a unique consequence model, there may be incompatibility between the loss ratios of the derived vulnerabilities, as each one considers each damage state in a specific different way. This is one of the main issues when combining vulnerability functions from different sources. This is particularly true given the damage results of the studies are shown considering these categories of the damage states. The reviewer saw this mentioned in the further developments of the discussion, but is should be included in the computation of the vulnerability curves here in some way, for the results to be coherent.

R/: We appreciate this important comment. Indeed, we agree with your argument and as we mentioned in our previous response, we decided to use a set of fragility functions from Martins and Silva (2021) and Villar Vega et al. (2017) that were calculated using the same methodology and with a consistent definition of the limits damage states.

10. Given the exposure is not considering separately the 2-storey housing, there are no vulnerabilities for 2-storeys considered, even when it is more common in the urban environment than the single storey houses. This should be included.

R/: Again, thank you for pointing us to this fact. As we mentioned in response to comment # 7, we updated our exposure model accounting for one, two and three-story houses.

- **Section 3.4. Social Vulnerability (SV)**

11. Major comment: One of the main criticisms of the paper is the consideration of the social vulnerability index as a percentage increase using the expression (1+SVI). As stated in the study "The min-max normalization was used to standardize the SV indicators from zero to one to estimate the SVI per municipality. Higher score indicate more socially vulnerable municipalities, and lower scores reflects less vulnerable ones. Then, the indicators were integrated by summing them with equal weight, as followed in Contreras et al. (2020c). The resulting SVI index is therefore used to adjust the percentage of economic losses with respect to the cost presented by the building inventory, i.e. multiplying them by (1+SVI) (Carrenio et al., 2007).". The problem with this is that there is no analysis done on the significance of the variables included within the study and no way to know if there are variables that shouldn't be included and if anything is counted double.

R/: Thank you for your comment. The reviewer is correct; in the submitted manuscript version, there was no analysis done on the significance of the variables included within the study and no way to know if there are variables that shouldn't be included and if anything is counted double. Therefore, considering this observation, which is the same written by Reviewer 1, to avoid problems interpreting the model and overfitting, we checked the multicollinearity by

[revised manuscript text omitted]

a. Dependent Variable: SV

The following table groups the variables by composite indicators (This table is not included in the manuscript).

| Composite indicators | Indicators |
|---|---|
| Population | Native Indigenous population |
| | Population density (inhabitants/km2) |
| | Number of people per household |
| Economy | Population unemployed |
| | Population with unsatisfied basic needs (UBN) |
| | Total population in poverty |
| Infrastructure | Households with no electric energy access |
| | No sewage system |
| Education | Illiteracy rate |
| Health | Deceased due to COVID-19 |

12. Additionally, considering this index as a "percentage increase" is extremely misleading. If there was a way to correlate the SVI of each variable in economic terms to the direct economic loss, then this could be done. But this is not done and there is no parametric study or anything else to validate any of the assumptions.

R/: Thank you very much to the reviewer for raising this interesting question. Little research has tested the correlation between social vulnerability (SV) and losses. To our best knowledge, the relationship between SV and modeled losses has been so far informative rather than indicating that total losses (measured as dollar losses or debris generated) increase with SV (Schmidtlein et al, 2011). However, it was found that only relative losses (dollar losses per average family income) tend to increase with SV. Case study areas with a low SV tend to have more material goods with significant monetary value (dollar) exposed to risk, than areas with high SV. Therefore, we should expect a negative correlation between property losses and SV (Cutter and Finch, 2008). It is important to understand that while total loss (dollar) in case study areas with high SV is lower, the impact of those losses in their communities is high (Schmidtlein et al., 2011).

'(…)

**5.2. Effects of social vulnerability (SV)**
Very few research studies have tested the correlation between social vulnerability (SV) and losses. To our best knowledge, the relationship between SV and modeled losses has been so far informative rather than indicating that total losses (measured as dollar losses or debris generated) increase with SV (Schmidtlein, Shafer, Berry, & Cutter, 2011). However, it was found that only relative losses (dollar losses per average family income) tend to increase with SV. Case study areas with a low SV tend to have more material goods with significant monetary value (dollar) exposed to risk, than areas with high SV. Therefore, we should expect a negative correlation between property losses and SV (Cutter & Finch, 2008). It is important to understand that while total loss (dollar) in case study areas with high SV is lower, the impact of those losses in their communities is high (Schmidtlein et al., 2011). (…)'

13. This SVI cannot be considered a percentage unless there is backup data validating this. This has been done also in fatality modelling where the models that are presented in any publication are previously calibrated and validated with data from historic events. Moreover, considering previous events reporting post-loss amplification that include costs from the response and recovery stages in some disasters, it has been shown that numbers over 30-40% are almost non-existent (what is demand surge? Olsend and Porter 2010), while this study mention cases with increases

of up to 60%. There may be a problem with the explanation of the methodology, but as it is right now it is very difficult understand how it can relate to economic losses, especially direct physical losses after an event.

R/: Thanks again for this question. Social vulnerability assessment considers variables and indicators with different units, e.g., number, percentages, number of people/m2, etc. This is the main reason to use normalization to construct the social vulnerability index (SVI), which is why there is no unit, and the levels of SV are expressed in ordinal categories, e.g., high, medium, and low, according to ranges defined by authors. The reason to use percentage is that this value is integrated with the percentage of economic losses with respect to the cost presented by the building inventory.

14. (These limitations are also afterwards mentioned by the authors in the discussion, but it is a MAJOR limitation of the inclusion of the SVI methodology in the results in this study, as there is no validation or calibration of any kind for the methodology)

R/: Data related to the damages caused by natural hazards are usually low quality. It is difficult to compare the severity of the event's characteristics in different zones of any case study area (Schmidtlein et al, 2011), because it is wrong to assume that losses are uniformly distributed (Cutter and Finch, 2008). Unfortunately, SV is estimated at the national or regional scale, making it difficult to calibrate or validate any SV model based on damages after earthquakes, which will be an ideal procedure, as the reviewer suggested.

15. Table 9 numbers are misleading as a direct non-weighted average of the 18 scenarios is not probabilistically and statistically sound. It should consider the contribution of each event, otherwise the less probable events are counted in the same way as the more probable ones. In this way, as when computing AAL from a probabilistic analysis, the contributions should consider the probability of occurrence of each scenario. After saying this, it is advised not to present this table and instead present one with the analysis of each scenario done separately as in a deterministic approach, unless it is possible to demonstrate that the 18 scenarios included account for the 100% of the 475 years return period loss and a weighted average is calculated based on the contribution of each.

R/: We appreciate this comment. We agree that 18 scenarios are not enough to represent 100% of the hazard. In the light of the comment, we decided to remove table 9 and instead include a table that shows the percentage of buildings for each damage state in the eighteen scenarios. We also added a Figure that illustrates the variability of these damage estimates.

One point to note here is that Sabana Centro has an uneven distribution of the building stock, therefore, similar earthquakes with the same magnitude and similar depth such as the Mw 6.25 Sopó, Mw 6.25 Tabio and Mw 6.25 Nemocón have different consequences in terms of damage and economic losses.

The following was included in section 4.1

**4.1. Damage forecast**

'(…) The damage calculations were conducted for all the seismic events presented in Table 2. Table 10 shows the resulting percentage of buildings that suffer damage for each of the eighteen seismic risk scenarios. The results show that the worst scenario for the region is the Mw 6.95 Cota, which has 21.37% collapsed buildings, and the highest percentages of severe and moderate damage. Interestingly, the Mw 6.85 Tocancipá had 8.12% of collapsed buildings, roughly 2.5 times less than the Mw 6.95 Cota. This difference is a consequence of the uneven distribution of the buildings stock in the region, as nearly 40% is in Chía and Cajicá, which are close to Cota. A similar situation occurs for the earthquakes of Mw 6.25 in Sopó, Tabio and Cajicá.

**Table 10.** Expected damage for each of the eighteen scenarios presented in Table 2. The name of the scenarios has the magnitude of the events and the municipality where they are located.

| Scenario | Percentage of buildings for each damage state | | | | |
|---|---|---|---|---|---|
| | No damage | Slight | Moderate | Extensive | Collapse |
| 5.35 Tenjo | 76.99 | 14.41 | 3.86 | 1.95 | 2.79 |
| 5.55 Cota | 74.44 | 15.17 | 4.35 | 2.32 | 3.72 |
| 5.65 Zipaquirá | 62.77 | 22.12 | 6.48 | 3.41 | 5.23 |
| 5.95 Chía | 57.05 | 21.31 | 7.36 | 4.44 | 9.85 |

| | | | | | |
|---|---|---|---|---|---|
| 5.95 Gachancipá | 66.78 | 19.60 | 5.66 | 3.01 | 4.94 |
| 5.95 Tenjo | 73.69 | 15.72 | 4.41 | 2.35 | 3.83 |
| 6.15 Tocancipá | 63.26 | 20.44 | 6.25 | 3.49 | 6.56 |
| 6.25 Nemocón | 65.26 | 19.73 | 5.81 | 3.19 | 6.01 |
| 6.25 Sopó | 44.90 | 25.57 | 9.48 | 5.90 | 14.14 |
| 6.25 Tabio | 54.11 | 24.11 | 8.02 | 4.64 | 9.12 |
| 6.35 Cajicá | 38.73 | 26.89 | 10.63 | 6.83 | 16.92 |
| 6.35 Cogua | 47.20 | 26.56 | 9.49 | 5.66 | 11.09 |
| 6.45 Cogua | 39.74 | 27.14 | 10.61 | 6.74 | 15.77 |
| 6.55 Sopó | 35.51 | 26.97 | 11.08 | 7.29 | 19.15 |
| 6.65 Nemocón | 51.80 | 24.04 | 8.29 | 4.95 | 10.92 |
| 6.65 Tabio | 42.97 | 25.86 | 9.84 | 6.23 | 15.10 |
| 6.85 Tocancipá | 53.10 | 25.37 | 8.57 | 4.84 | 8.12 |
| 6.95 Cota | 32.49 | 26.86 | 11.51 | 7.78 | 21.37 |

Figure 13 shows the variability of the results for each damage state of Table. The results illustrate the notable variability of the damage estimates between the different seismic scenarios. This variability is higher in for the No damage state, which ranges between 32.49% and 76.99%, corresponding to the Mw 6.95 Cota and the Mw 5.35 Tenjo scenarios. These two also had the highest and smallest percent of collapse, respectively. The median results of the eighteen scenarios for the damage states are 53.6%, 24.1%, 8.1%, 4.7% and 9.5%, values that are close to those of the Mw 6.25 Tabio.

[Figure]

**Figure 13.** Percent of buildings expected to experience either no damage ('None'), slight, moderate, extensive damage, or collapse as a result of the 18 earthquakes scenarios presented in Table 2.

**Section 4.4**

16. As stated previously, to present these analyses, it would be important to show the contribution of the 18 scenarios to the total hazard in the region (based on the disaggregation results). If not Figure 13 is misleading, considering that it says: "Within the municipalities, the mean percentage of losses is presented with respect to the total expected losses in the region".

"The economic losses experienced by a province due to an earthquake depends on the event's epicenter as it is depicted in Figure 12." Figure 12 does not show in any way anything regarding the epicenter. It may not be only the epicenter but also the Mw for each event the cause of the differences, so this statement is not provable from the Figure. Delete it.

R/: We understand the reviewer comment and agree that each scenario has a different contribution to the hazard. Accordingly, we decided to remove Figure 13 and included Table 14. We kept figure 12 but we improve our analysis of this figure. The main point here is that Sabana Centro has an uneven distribution of the building stock, therefore, similar earthquakes with the same magnitude and similar depth such as the Mw 6.25 Sopó, Mw 6.25 Tabio and Mw 6.25 Nemocón have different consequences in terms of economic losses. We included the information for economic losses of each scenario in Table 9 of the previous comment. We deleted the statement you suggested.

**4.4. Mean direct economic losses for the earthquake scenarios**

The seismic ground motion fields expected for each earthquake scenario listed in Table 2 were simulated 1,000 times to account for their aleatoric uncertainty as advised by Silva (2016), making use of the OQ Engine. The physical vulnerability was calculated in a similar manner as for the Mw 5.95 earthquake scenario. Figure 15 shows the loss exceedance curves that describe the probability of exceeding a given percent of economic losses for each earthquake scenario.

[Figure]

**Figure 15.** Loss-exceedance curves (LEC) as a function of percentage of economic losses in the region for the 18 earthquake scenarios considered (Table 2) whose ground motion fields were simulated 1000 times. The legend is sorted according to the line position in the figure from left to right.

Among the simulated scenarios, the most significant economic losses might occur with the scenario that considers an earthquake of Mw 6.95 in Cota and the smallest with the simulation of the earthquake of Mw 5.35 in Tenjo. For example, the probability that economic losses exceed 20% in the "6.95 Cota" scenario is 61%, while for the scenario "5.35 Tenjo", the probability is 3%. Besides magnitude, the uneven distribution of the building stock in the region is an important factor that exerts an influence on economic losses, as shown by the differences between the three Mw 6.25 events in Figure 15. The highest economic losses are in the municipality of Cota with 24.29% of losses (1517.56 US$ million). On average, economic losses when including social vulnerability increase by 37% as shown in Table 14.

**Table 14.** Expected Economic Losses in the region for each of the eighteen scenarios considered. Economic losses when including SV consider the percentage of losses with respect to the total losses per municipality and are adjusted with the SVI.

| Scenario | Direct economic losses | | Economic losses with SV | |
| | Losses (M.USD) | Percentage of Losses (%) | Losses after considering the SVI (M. USD) | Losses + SVI (%) |
| --- | --- | --- | --- | --- |
| 5.35 Tenjo | 322.99 | 5.17 | 437.28 | 7.00 |
| 5.55 Cota | 434.35 | 6.95 | 588.35 | 9.42 |
| 5.65 Zipaquirá | 426.26 | 6.82 | 592.19 | 9.48 |
| 5.95 Chía | 900.49 | 14.41 | 1240.72 | 19.86 |
| 5.95 Gachancipá | 438.62 | 7.02 | 599.55 | 9.60 |
| 5.95 Tenjo | 461.39 | 7.38 | 615.75 | 9.85 |
| 6.15 Tocancipá | 571.58 | 9.15 | 778.38 | 12.46 |
| 6.25 Nemocón | 484.05 | 7.75 | 660.98 | 10.58 |
| 6.25 Sopó | 1181.64 | 18.91 | 1628.70 | 26.07 |
| 6.25 Tabio | 824.01 | 13.19 | 1120.51 | 17.93 |
| 6.35 Cajicá | 1295.51 | 20.73 | 1790.78 | 28.66 |
| 6.35 Cogua | 794.91 | 12.72 | 1101.68 | 17.63 |
| 6.45 Cogua | 1154.40 | 18.47 | 1596.73 | 25.55 |
| 6.55 Sopó | 1450.55 | 23.21 | 2002.18 | 32.04 |
| 6.65 Nemocón | 888.24 | 14.22 | 1211.52 | 19.39 |

| | | | | |
|---|---|---|---|---|
| 6.65 Tabio | 1343.27 | 21.50 | 1824.25 | 29.19 |
| 6.85 Tocancipá | 695.64 | 11.13 | 949.69 | 15.20 |
| 6.95 Cota | 1517.56 | 24.29 | 2093.23 | 33.50 |

- **Discussion**

17. Just until this section this is stated: "The simulations of eighteen seismic scenarios with a return period of 475 years show that half of the building stock will experience some degree of damage". How was this '475 years' return period calculated? Even when the disaggregation was done for the '475 years' return period, how can it be confirmed that the 18 scenarios add up to the 100% contribution for the hazard for this return period? Either way this statement should be included in some way in previous sections and not only until the discussion.

R/: Thank you for this observation. It is not possible to ascertain that eighteen scenarios add 100% of the hazard for any given period. For Sabana Centro, crustal sources account for more than 80% percent of the hazard. The eighteen scenarios were selected from the PSHA because they were representative of crustal source earthquakes.

We clarified this in section 3.1.1.

**3.1.1. Definition of the earthquake scenarios**

'(…) Based on this disaggregation, eighteen crustal events were selected from the probabilistic seismic hazard catalogue to be used in this study to calculate the expected damages and economic losses. The magnitude, location and geometry of ruptures are shown in Table 2. The epicenter of each event is located within the municipality mentioned in the first column of the Table 2 and as shown in Figure 5.

[Figure]

Figure 5. Location of the 18 selected events, the color range varies according to the magnitude of the timing of the events. The gray region represents the zone where shallow events have the largest contribution.

- **Effects of SV**

18. Major comment: The main criticism for this approach is also stated by the authors in this sentence: "First, as not all social aspects exert equel effect after an earthquake, it is necessary to develop a weighted approach to best estimate a more realistice SVI for earthquake events.

R/: Thank you for bringing this statement to our attention. We have decided to eliminate it because the allocation of a weighted approach not only will not capture all social aspects after an earthquake but allocating weights to the composite indicators will add subjectivity to the analysis.

19. A second improvement required is to devise a better way of estimating the economic impact of social vulnerability. One potential approach is to generate a database of past earthquakes with different consequences that

include the economic costs". There are not needed future improvements but major limitations of the proposed approach. In this kind analyses, as when performing a linear regression, it is important to avoid counting double and establish the significance of each variable within the analysis, if not this could be overestimating the vulnerability and losses in the region considerably.

R/: Thank you again for this observation. We agree with the reviewer that to avoid problems interpreting the model and overfitting, we checked the multicollinearity by looking at the variance inflation factor (VIF) of each variable and indicator. The VIF was identified in a linear regression that included collinearity diagnostics produced in SPSS (Field, 2005). We excluded those variables and indicators that were potentially correlated with others and those that did not add significant information according to the collinearity diagnostics (Table 6). Eventually, the model included only independent and relevant indicators to estimate the SV in the case study area (Tables 7 shown above in the answer to comment 11).

However, in this sentence, we were referring to the economic impact of the SV after an earthquake, not the method for the vulnerability assessment. Considering that this sentence can be misunderstood, we also decided to eliminate it.

20. Also, using the min and max approach is very subjective as many variables as unemployment and poverty are tempered with by local organism. This kind of indicators are good to compare and prioritize actions withing regions but cannot be used in the way they are presented in this study to increase direct physical losses.

Thanks for this comment. However, it is necessary to differentiate the method, the indicators, and data sources in the assessment of SV, the operationalization of the analysis, and the actions. Regarding the method, as we explained before, SV assessment considers variables and indicators with different units. This is the main reason to use normalization to construct the social vulnerability index (SVI). With respect to the indicators, we agree with the reviewer that governments can temper data regarding unemployment and poverty, actually, in the most vulnerable countries, this data does not exist, or it is not accessible, but any analysis has an uncertainty level, and we decided to accept it using official numbers. As we explained before based on the literature review, integrating the level of SV to the physical losses will not produce a significant increase in the last one, considering that they are negatively correlated (Cutter & Finch, 2008). Integrating the SV analysis makes the risk assessment more holistic and useful to prioritize actions at the scale of the assessment because it can be national, regional, or municipal.

**5.3. Effects of social vulnerability (SV)**

'(…) Integrating the level of SV to the physical losses will not produce a significant increase in the last ones, considering that they are negatively correlated (Cutter & Finch, 2008), but the result will be a more holistic risk assessment, also useful to prioritize actions at the regional level. (…)'